# Stable Forgetting: Bounded Parameter-Efficient Unlearning in LLMs

## Abstract

Machine unlearning in large language models (LLMs) is essential for privacy and safety; however, existing approaches remain unstable and unreliable. A widely used strategy, the gradient difference method, applies gradient descent on retained data while performing gradient ascent on forget data, the data whose influence should be removed. However, when combined with cross-entropy loss, this procedure causes unbounded growth of weights and gradients, leading to training instability and degrading both forgetting and retention. We provide a theoretical framework that explains this failure, explicitly showing how ascent on the forget set destabilizes optimization in the feedforward MLP layers of LLMs. Guided by this insight, we propose *Bounded Parameter-Efficient Unlearning*, a parameter-efficient approach that stabilizes LoRA-based fine-tuning by applying bounded functions to MLP adapters. This simple modification controls the weight dynamics during ascent, enabling the gradient difference method to converge reliably. Across the TOFU, TDEC, and MUSE benchmarks, and across architectures and scales from 125M to 8B parameters, our method achieves substantial improvements in forgetting while preserving retention, establishing a novel theoretically grounded and practically scalable framework for unlearning in LLMs[1].

## 1 Introduction

The advent of foundation models has profoundly reshaped machine learning. Yet their large-scale deployment has also revealed critical vulnerabilities, raising concerns about safety and data governance. During pretraining, these models absorb massive datasets that frequently contain sensitive, copyrighted, or personally identifiable information (Shi et al., 2024). Consequently, the ability to selectively *forget* such information has become both a regulatory requirement and a technical necessity (Bourtoule et al., 2021; Cao & Yang, 2015). Machine unlearning, removing the influence of specific data without full retraining, thus stands as one of the most urgent challenges for the ethical deployment of large-scale models.

Current approaches to unlearning in large language models (LLMs) face fundamental limitations that hinder their effectiveness. A common strategy is to treat unlearning as a finetuning problem, and two main approaches have emerged. The first is *full fine-tuning*, where the original model weights are updated on a forget set and the data whose influence is to be removed. The standard procedure applies gradient ascent on this forget set with a cross-entropy loss (Maini et al., 2024), but this leads to training instability and degradation in retention quality, i.e., the preservation of knowledge that should not be forgotten (Maini et al., 2024). To address this, the gradient difference method was proposed, which simultaneously optimizes a retention objective via gradient descent on a retention set (also known as a neighborhood set) and a forget objective via gradient ascent on a forget set (Maini et al., 2024; Cha et al., 2025). While conceptually appealing, this method becomes unstable when paired with standard objectives such as cross-entropy, often degrading both retention and forgetting performance. The second approach is *parameter-efficient fine-tuning* , such as Low-Rank Adaptation (LoRA) (Hu et al., 2022), which alleviates the computational and memory demands of full fine-tuning and makes unlearning more scalable. However, when combined with the gradient difference method utilizing a cross-entropy loss, these techniques continue to suffer from instability issues.

---

[1]Code will be open-sourced upon acceptance.

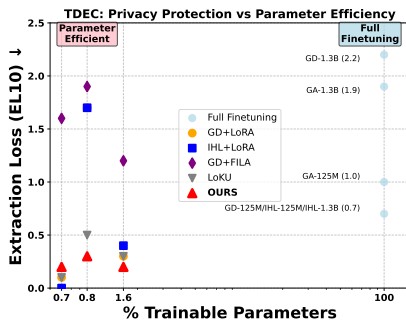 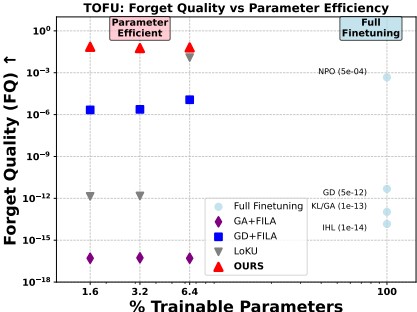

Figure 1: **Balancing efficiency and effectiveness in parameter tuning.** (**Left**) On TDEC, our method achieves stronger privacy protection than existing parameter-efficient baselines while requiring fewer parameters than full fine-tuning. (**Right**) On TOFU, our approach maintains consistently high forget quality across LoRA ranks, outperforming state-of-the-art baselines by orders of magnitude while preserving parameter efficiency.

These limitations have motivated a wave of refinements; however, progress has been incremental. Fisher information-weighted initialization (FILA) (Kim et al., 2025) and Inverted Hinge Loss (IHL) (Cha et al., 2025) improve stability through carefully designed forget-set objectives and initialization strategies, whereas their combination (Cha et al., 2025) offers only partial relief from unlearning instability. More recent approaches, including adversarial unlearning frameworks (Setlur et al., 2022) and residual feature alignment methods (Guoliang Li, 2024), show promise but are fundamentally constrained by their dependence on linear parameterization.

In this study, we developed a theoretical framework to analyze the training instability of the gradient difference method with cross-entropy loss for unlearning. Our analysis reveals that the fundamental source of instability lies in the ascent step: optimization on the forget set causes the weights and gradients in the feedforward MLP layers of LLMs to grow excessively, which results in training instability. This leads to our central insight: parameterizing the feedforward weights of the MLP layers with a bounded function provides a principled mechanism for stabilizing optimization under gradient ascent.

Building on this principle, we extend the gradient difference framework with LoRA-based fine-tuning, demonstrating that bounded parameterization directly stabilizes both weights and gradients during unlearning. Motivated by this insight, we propose **bounded parameter-efficient unlearning**, a method that applies bounded functions to the adapters in the feedforward layers of the MLPs of LLMs, enabling stable and parameter-efficient fine-tuning with a cross-entropy forgetting objective. This approach directly overcomes the key limitations of prior methods, providing both theoretical guarantees and practical effectiveness for parameter-efficient fine-tuning.

We thoroughly assessed our framework using extensive evaluations on the TOFU, TDEC, and MUSE benchmarks, which cover a range of model architectures (GPT-Neo, Phi, and LLaMA) and sizes (125M-8B parameters). As illustrated in Fig. 1, our technique achieves cutting-edge unlearning performance, offering significant improvements in forget quality compared with existing parameter-efficient methods while preserving model utility across various scenarios.

Our contributions are

1. We develop a theoretical framework for analyzing the gradient difference method with cross-entropy loss, showing that instability arises from the ascent step, where optimization on the forget set drives uncontrolled growth of weights and gradients in MLP feedforward layers. From this analysis, we derived the key insight that parameterizing feedforward weights with a bounded function stabilizes optimization under gradient ascent.

2. Building on this principle, we propose **bounded parameter-efficient unlearning**, a parameter-efficient method that applies bounded functions to LoRA adapters in feedforward layers. Our approach achieves stable and scalable unlearning, delivering substantial improvements in forgetting quality while preserving retention quality and establishing new *state-of-the-art* results across multiple machine-unlearning benchmarks.

## 2 RELATED WORK

**Machine unlearning.** Machine unlearning seeks to remove the influence of specific data from a trained model while preserving the overall performance and avoiding the cost of retraining from scratch (Bourtoule et al., 2021; Cao & Yang, 2015). For large language models (LLMs), retraining from the beginning is infeasible at scale (Nguyen et al., 2022; Qu et al., 2023). Recent work categorizes unlearning methods into four main families: (1) **full fine-tuning methods**, which update model weights using gradient-based optimization with a forgetting loss (Yao et al., 2024; Huang et al., 2024; Maini et al., 2024); (2) **parameter-efficient fine-tuning**, employing LoRA and its variants (Hu et al., 2022; Cha et al., 2025; Maini et al., 2024); (3) **preference-based methods**, which leverage alignment signals (Zhang et al., 2024); and (4) **representation-based methods**, which directly modify internal weights (Ilharco et al., 2022; Meng et al., 2022). Below, we discuss the approaches from categories (1) and (2) as they are the most relevant to our work. Our approach belongs to category (2), but we provide comparisons of all four families in our experiments (Section 4).

**Full Fine-Tuning.** Full fine-tuning methods approach unlearning by directly updating all model parameters on a designated forgetting set. The most common strategy is to apply gradient ascent to this set with a cross-entropy loss (Maini et al., 2024), encouraging the model to unlearn the targeted information. However, this simple approach often results in instability and poor retention quality because the influence of the forget set can interfere with unrelated knowledge that should be preserved Maini et al. (2024). To mitigate these issues, the *gradient difference method* has been introduced (Maini et al., 2024; Cha et al., 2025), which jointly optimizes a retained objective via gradient descent on a retained set while applying gradient ascent on the forget set. Although this approach offers a more balanced formulation, it remains unstable when paired with standard objectives, such as cross-entropy, often degrading both forgetting and retention. More recent refinements attempt to stabilize full fine-tuning by modifying the forget objective, initialization strategies, or optimization dynamics (Kim et al., 2025; Cha et al., 2025). Furthermore, this approach is computationally expensive because it fine-tunes all the weights of the base model.

**Parameter-Efficient Fine-Tuning.** Parameter-efficient methods, such as LoRA, reduce computational costs by factorizing weight updates into low-rank matrices (Hu et al., 2022). These approaches are typically combined with the gradient difference method, in which low-rank adapters are trained using gradient descent on a retention set and gradient ascent on a forget set. However, when paired with cross-entropy loss, this setup often leads to severe training instability (Huang et al., 2024; Cha et al., 2025). Similar to full fine-tuning, several studies have shown that replacing cross-entropy with alternative objectives can improve performance (Pan et al., 2024). Other studies have explored modified objectives and initialization strategies to further mitigate instability (Kim et al., 2025; Cha et al., 2025). Despite these advances, the core issue of gradient explosion under cross-entropy forgetting remains unsolved. In this study, we directly addressed this challenge. As demonstrated in Section 4, our method enables stable unlearning within the gradient difference framework while using cross-entropy and consistently outperforms previous approaches. A broader survey of machine unlearning methods is provided in Appendix A.

## 3 METHODOLOGY

### 3.1 PRELIMINARIES

**Problem Formulation.** Machine unlearning aims to remove the influence of forget data $\mathcal{D}_f$ while preserving performance on retain data $\mathcal{D}_r$. Given a model $f_\theta$ with parameters $\theta$, the objective combines retention and forgetting:

$$\mathcal{L}_r(\theta) + \lambda \mathcal{L}_f(\theta) \tag{1}$$

where $\mathcal{L}_r(\theta) = \mathbb{E}_{(x,y) \sim \mathcal{D}_r}[\mathcal{L}(f_\theta(x), y)]$, $\mathcal{L}_f(\theta) = \mathbb{E}_{(x,y) \sim \mathcal{D}_f}[\mathcal{L}(f_\theta(x), y)]$, and $\lambda > 0$ controls the forgetting strength. The optimization then proceeds by simultaneously training $\mathcal{L}_r(\theta)$ via gradient descent and $\mathcal{L}_f(\theta)$ via gradient acsent.

**Gradient-Based Unlearning.** The gradient difference method optimizes the unlearning objective through:

$$\theta_{t+1} = \theta_t - \alpha_r \nabla_\theta \mathcal{L}_r(\theta) + \alpha_f \nabla_\theta \mathcal{L}_f(\theta) \tag{2}$$

This combines gradient descent on retain data with gradient ascent on forget data. While gradient ascent effectively increases loss on $\mathcal{D}_f$, it suffers from optimization instability when combined with cross-entropy loss (Cha et al., 2025). In Pan et al. (2024), the issue was addressed by replacing the cross-entropy loss on the forget set with an Inverted Hinge Loss (IHL). In contrast, as we show in Section 3.3, our methodology enables training directly with the cross-entropy loss on the forget set.

**Low-Rank Adaptation**  LoRA parameterizes weight updates through low-rank decomposition:

$$W = W_0 + AB^T \tag{3}$$

where $W_0$ are frozen pre-trained weights, and $A \in \mathbb{R}^{d \times r}$, $B \in \mathbb{R}^{k \times r}$ are trainable matrices with rank $r \ll \min(d, k)$. For transformer architectures comprising of attention and MLP feedforward layers Eq. (3) is generally applied to the MLP and attention layers. While LoRA reduces computational costs from $dk$ to $(d + k)r$ parameters, its root issues in unlearning remains underexplored, with existing solutions addressing symptoms rather than the fundamental optimization instabilities that arise in gradient-based unlearning scenarios.

## 3.2 THEORETICAL ANALYSIS

In Eq. (2), the gradient difference method combines two objectives: a forget loss optimized via gradient ascent and a retain loss optimized simultaneously via gradient descent. Prior work Cha et al. (2025) has shown that when the forget loss employs cross-entropy, fine-tuning becomes unstable. To understand this phenomenon, we analyze gradient ascent under the cross-entropy loss and establish two theorems showing that weights and gradients can diverge. This theoretical insight motivates our approach in Section 3.3, where we propose a method to mitigate such divergence and stabilize training on the forget set.

The networks we consider will all be trained with the cross-entropy loss as the retain and forget loss in Eq. (1). In this section, we work generally and simply denote the cross-entropy loss associated to an MLP by $\mathcal{L}$. We let $C$ denote the number of distinct class labels so that the output dimension of the network will be $C$. The output probabilities of the network will be denoted $p$ and we recall for a class label denoted by $y$ the cross-entropy loss of the predicted $p \in \mathbb{R}^C$ compared to $y$ is given by

$$\mathcal{L}(p, y) = -log(p_y) \tag{4}$$

where $p_y$ is the yth-component of $p \in \mathbb{R}^C$. Using Eq. (4) and the chain rule we have that the gradient vector of $\mathcal{L}$ with respect to the logits $z$ on a class $y$ is given by

$$\nabla_z \mathcal{L} = p - e_y \tag{5}$$

where $e_y$ denotes the one-hot vector that is 1 in the $y^{\text{th}}$-position. For more details on the cross-entropy loss we refer the reader to Prince (2023).

When training under gradient ascent the optimizer wants to push the predictions $p$ away from $e_y$ as it seeks to move towards a maximum of the cross-entropy loss. We thus get that

$$\nabla_z \mathcal{L} \to e_j - e_y \tag{6}$$

where $j$ is an index $1 \le j \le C$ such that $j \ne y$ so that the one hot vectors associated to the classes $y$ and $j$ are distinct. In particular, under a gradient ascent trajectory that is approaching a maximum the logit gradient $\nabla_z \mathcal{L}$ does not approach zero and hence

$$||\nabla_z \mathcal{L}|| > C > 0 \tag{7}$$

stays bounded away from zero for some constant $C > 0$, where $|| \cdot ||$ denotes the Euclidean norm (see Appendix B.1 for details on notation). We note that in the case of gradient descent the term on the right of Eq. (7) approaches zero yielding a completely different behavior to gradient ascent.

**Lemma 3.1.** *Let $\mathcal{L}$ denote the cross-entropy loss trained on a MLP $F$ with $L$ layers under gradient ascent. Let $z(t)$ denote the logits at iteration $t$. Then if $\mathcal{L}(t) \to \infty$ it follows that $z(t) \to \infty$ in norm.*

The proof of Lemma 3.1 is given in Appendix B.2. The above lemma shows that when training with gradient ascent if the cross-entropy loss approaches a global maximum the logits get large. The

following theorem shows that this can lead to large weights or gradients in the final layer. For the theorem we will need the notation of activation outputs. Given an $L$ layer MLP denoted $F$, we let $a_l$ for $1 \leq l \leq L$ denote the output of layer $l$. For details on the notation we use for MLPs we refer the reader to Appendix B.1.

**Theorem 3.1.** *Let $F$ be a $L$-layer MLP. Suppose under gradient ascent with iterations $t$, the logits $z(t) \to \infty$. Then if the activation output $||a_{L-1}(t)|| \leq C_1$ for large $t$, where $C_1 > 0$ is a constant, it follows*

$$||W_L(t)|| \to \infty. \tag{8}$$

*In the case there is no such bound on $||a_{L-1}(t)||$ it follows that there exists a subsequence of iterations $t_k$ such that*

$$||\nabla_{W_L} \mathcal{L}(t_k)|| \to \infty. \tag{9}$$

The proof of Theorem 3.1 is provided in Appendix B.2. In practice, training is limited to a finite number of iterations, and models rarely reach a regime where gradients or parameters fully stabilize. As established in Theorem 3.1, gradient ascent can drive the weights and gradients of the final layer to grow excessively. While such growth may not disrupt training immediately, it can cascade backward through to earlier layers, amplifying both weights and gradients in more than one layer and ultimately producing unstable dynamics. We formalize this propagation effect in Appendix B.3, where Theorem B.2 shows how instability originating in the final layer extends to preceding layers. Notably, our analysis focuses on pure gradient ascent, to give the reader the main idea of why unlearning is difficult when a gradient ascent term is present. We extend both Lemma 3.1 and Theorem 3.1 to the case of the gradient difference method in Appendix B.2, see Lemma B.1 and Theorem B.1 in Appendix B.2. Furthermore, we note that empirical evidence in Fig. 2 shows that under the gradient difference method, weights grow excessively, indicating that the ascent term on the forget set is the primary driver of this instability.

### 3.3 BOUNDED PARAMETER-EFFICIENT UNLEARNING

Theorem 3.1 and Theorem B.2 in Appendix B demonstrate that gradient ascent drives weights and gradients in the feedforward layers of MLPs to grow excessively, which can destabilize training. In Section 4, we empirically confirm this effect: when fine-tuning with LoRA under the gradient difference framework, weights and gradients grow excessively large, and this growth is the primary reason LoRA fails to perform effective unlearning. To address this issue, we propose a simple yet effective architectural modification that implicitly regularizes the weights of the MLP layers

Specifically, let $\phi : \mathbb{R} \to \mathbb{R}$ denote a bounded non-linear function. We redefine the adapter transformation (see Eq. (3)) in the feedforward layers as

$$\phi(AB^T)x + b \tag{10}$$

where $A$ and $B$ are the low-rank adapter matrices, $x$ is an input data, and $b$ is a bias term. The bounded nonlinearity $\phi$, applied elementwise to $AB^\top$, constrains the ascent dynamics and prevents uncontrolled growth of weights and gradients. As we demonstrate in Section 4, this adjustment yields substantially more stable training and improved performance with finetuning across a variety of machine un-learning benchmarks.

For the choice of $\phi$, we took $\tanh$ as this is a well-known activation choice in machine learning. More recently, Ji et al. (2025) showed that applying a sine mapping, $\sin(\omega AB^\top)$ with frequency $\omega > 0$, produces a high-rank matrix whose rank grows with $\omega$, yielding stronger fine-tuning performance on a range of LLM benchmarks. Since $\sin(\omega \cdot)$ is bounded for any $\omega > 0$, it aligns naturally with our setting. In our experiments (Section 4), sine functions, particularly with larger $\omega$, consistently outperformed other bounded alternatives. Accordingly, we focus on both $\tanh$ and sine as choices for $\phi$. In Appendix C.4 we compare against using a sigmoid function and to demonstrate the necessity of boundedness, we also include an unbounded example, ReLU, for comparison.

We note that many LLMs employ normalization techniques such as layer normalization Ba et al. (2016) or batch normalization Ioffe & Szegedy (2015), which act on activated or pre-activated outputs. Our approach is fundamentally different: rather than normalizing activations, it constrains the weights directly, providing a distinct mechanism for stabilizing training.

**Attention layer.** In this work, we focused on analyzing the behavior of the feedforward layers of an MLP under gradient ascent with cross-entropy. Although LLMs also contain attention layers, our experiments revealed that instability in the gradient difference method arises primarily in the feedforward layers: their weights and gradients grow far more aggressively than those of the attention layers. Consequently, it is sufficient to constrain only the feedforward weights of the MLP blocks. A detailed empirical analysis of the attention layers is provided in Appendix C.6.

**Why not full fine-tuning?** In the full fine-tuning setting, optimization is carried out directly on the pretrained weights $W$. Applying a bounded transformation $\phi(W)$ in this case would overwrite these weights, thereby discarding the knowledge acquired during pretraining. In practice, parameter-efficient approaches introduce low-rank additions that augment the model with extra parameters while leaving the original $W$ unchanged. This separation makes it possible to safely apply bounded parameterizations to the adapters.

## 4 EXPERIMENTS

We conducted an empirical evaluation to assess sine-based parameter-efficient unlearning across dimensions critical to machine unlearning deployment. Our experimental design examines three core aspects: *unlearning efficacy*, quantifying how targeted knowledge is removed from the model; *utility preservation*, evaluating the retention of capabilities and out-of-distribution performance; and *optimization robustness*, analyzing the convergence stability and scalability across architectures and hyperparameter configurations. This evaluation framework enables a rigorous assessment of theoretical predictions and practical viability, establishing effectiveness relative to existing unlearning methodologies in large language models. For completeness and to ensure a fair comparison with prior work, we provide implementation details, model details, additional experimental details, and extended results in Appendix C with ethical statement in Appendix D.

### 4.1 EXPERIMENTAL SETUP

**Evaluation Benchmarks.** We used three datasets with their respective evaluation frameworks to assess unlearning effectiveness, utility preservation, and safety compliance. *1. TOFU (Task of Fictitious Unlearning)* (Maini et al., 2024): Evaluates forget quality through statistical divergence between unlearned and retain-only models, monitoring utility on retained tasks and generalization. *2. TDEC (Training Data Extraction Challenge)* (Carlini et al., 2021): Assesses privacy protection via extraction loss over ten queries ($EL_{10}$), reasoning accuracy preservation, and language modeling quality. *3. MUSE (Machine Unlearning Six-way Evaluation)* (Shi et al., 2024): Provides safety assessment across verbatim memorization, semantic knowledge retention, and privacy leakage dimensions. We evaluate against representative methods from each major unlearning discussed in Section 2. For new readers, please refer to Appendix C.1 for better understanding.

Our comparison includes gradient-based approaches (Gradient Ascent (GA) (Yao et al., 2024), Gradient Difference (GD) (Maini et al., 2024), KL-regularization (Liu et al., 2024), Inverted Hinge Loss (IHL) (Cha et al., 2025)), parameter-efficient methods (GD+LoRA (Hu et al., 2022), GA+FILA (Kim et al., 2025), GD+FILA (Kim et al., 2025), LoKU (Cha et al., 2025), LoKU (Cha et al., 2025)), preference-based techniques (DPO (Rafailov et al., 2023), NPO (Zhang et al., 2024)), and representation-based approaches (FLAT variants (Wang et al., 2024)). All baseline results are from their respective papers or Cha et al. (2025); Wang et al. (2024) unless otherwise specified. Comprehensive ablation studies comparing bounded versus unbounded activations are detailed in Table 7 (Appendix C.4), while comparison of IHL versus GD objectives with sine parameterization with statistical analysis is provided in Table 8 (Appendix C.5), computational overhead is mentioned in Appendix C.7, and sequential robustness in Appendix C.8. Extended results across all model configurations are provided in Appendix G, with detailed rank analysis in Tables 13 to 15.

### 4.2 RESULTS

Our analysis examined performance across multiple dimensions—unlearning effectiveness, utility preservation, and safety compliance—using models ranging from 125M to 8B parameters across GPT-Neo, Phi, and LLaMA architectures. The results demonstrate consistent improvements across

Table 1: (**Left**) **TOFU Forget10 evaluation on Phi-1.5B model.** *Original*: fine-tuned model before unlearning. *Retain90*: trained on 90% retain data only. Metrics: forget quality (FQ), model utility (MU), Rouge-L scores. Higher FQ and MU indicate better performance. Parameter-efficient methods use rank-4 LoRA with percentages showing modified weights. Baseline results from Cha et al. (2025); Yuan et al. (2025); Maini et al. (2024). Gray-highlighted **OURS** achieves state-of-the-art performance. (**Right**) **TDEC evaluation on GPT-Neo-1.3B model.** Models fine-tuned before unlearning. **Forgetting metrics** (lower better): $EL_{10}$ measures extraction vulnerability. **Retention metrics**: Reasoning, Dialogue (higher better), PPL (lower better). Parameter percentages show modified weights. Results from (Cha et al., 2025). Gray-highlighted **GD + Sine (Bounded)** achieves state-of-the-art performance, GD + Tanh (Bounded) shows second-best results with rank 16.

| Method | Primary Metrics | | Forget Set | Retain Set | Params (%) |
|---|---|---|---|---|---|
| | FQ ($\uparrow$) | MU ($\uparrow$) | Rouge-L ($\downarrow$) | Rouge-L ($\uparrow$) | |
| Original | 1.15e-17 | 0.52 | 0.93 | 0.92 | – |
| Retain90 | 1.00e+00 | 0.52 | 0.43 | 0.91 | – |
| *Full Fine-tuning Methods* | | | | | |
| KL | 7.38e-15 | 0.00 | 0.01 | 0.01 | 100.0 |
| DPO | 5.10e-17 | 0.48 | 0.41 | 0.67 | 100.0 |
| NPO | 2.56e-05 | 0.37 | 0.45 | 0.45 | 100.0 |
| GA | 2.06e-13 | 0.00 | 0.01 | 0.01 | 100.0 |
| GD | 2.55e-09 | 0.36 | 0.37 | 0.41 | 100.0 |
| IHL | 2.43e-17 | 0.51 | 0.53 | 0.76 | 100.0 |
| *Parameter-Efficient Methods* | | | | | |
| GD+LoRA | 1.45e-15 | 0.28 | 0.85 | 0.45 | 1.6 |
| GA+FILA | 5.10e-17 | 0.00 | 0.00 | 0.00 | 1.6 |
| GD+FILA | 2.17e-06 | 0.00 | 0.12 | 0.11 | 1.6 |
| LoKU | 1.39e-12 | 0.51 | 0.26 | 0.75 | 1.6 |
| ME+GD (LoRA) | 7.86e-01 | 0.52 | 0.14 | 0.93 | 1.6 |
| OURS (GD + Tanh) | 3.42e-01 | 0.49 | 0.28 | 0.85 | 1.6 |
| **OURS (GD + Sine)** | **9.43e-01** | **0.52** | **0.22** | **0.90** | **1.6** |

| Method | Forgetting | Retention | | | |
|---|---|---|---|---|---|
| | $EL_{10}$ ($\downarrow$) | Reasoning ($\uparrow$) | Dialogue ($\uparrow$) | PPL ($\downarrow$) | Params (%) |
| Before Unlearning | 67.6 | 49.8 | 11.5 | 11.5 | – |
| *Full Fine-tuning Methods* | | | | | |
| GA | 1.9 | 49.7 | 8.5 | 15.8 | 100.0 |
| GD | 2.2 | 48.4 | 12.7 | 10.8 | 100.0 |
| IHL | 0.7 | 48.4 | 12.5 | 11.0 | 100.0 |
| *Parameter-Efficient Methods* | | | | | |
| GD+LoRA | 1.7 | 45.0 | 9.7 | 31.8 | 0.8 |
| IHL+LoRA | 1.7 | 47.1 | 10.2 | 14.9 | 0.8 |
| GD+FILA | 1.9 | 44.2 | 5.5 | 54.5 | 0.8 |
| LoKU | 0.5 | 48.3 | 12.1 | 14.7 | 0.8 |
| OURS (GD + Tanh) | 0.8 | 46.7 | 10.3 | 18.2 | 0.8 |
| **OURS (GD + Sine)** | **0.3** | **52.1** | **12.7** | **10.9** | **0.8** |

Table 2: Extended safety evaluation on MUSE benchmark using LLaMA2-7B base model trained on BBC news corpus. **Forgetting metrics** (lower better): VerbMem (verbatim memorization), $KnowMem_f$ (forgotten knowledge), PrivLeak (privacy leakage, 0=no leakage). **Retention metric** (higher better): $KnowMem_r$ (retained knowledge). Parameter % show modified weight fraction during unlearning. Gray-highlighted **GD + Sine (Bounded)** achieves state-of-the-art performance on rank 4, results from Wang et al. (2024), GD + Tanh (Bounded) shows second-best averaged.

| Method | Forgetting | | | Retention | |
|---|---|---|---|---|---|
| | VerbMem ($\downarrow$) | $KnowMem_f$ ($\downarrow$) | PrivLeak ($\to$ 0) | $KnowMem_r$ ($\uparrow$) | Params (%) |
| Original LLM | 58.4 | 63.9 | -99.8 | 55.2 | – |
| Retained LLM | 20.8 | 33.1 | 0.0 | 55.0 | – |
| *Full Fine-tuning Methods* | | | | | |
| GA | 0.0 | 0.0 | 17.0 | 0.0 | 100.0 |
| GD | 4.9 | 27.5 | 6.7 | 109.4 | 100.0 |
| NPO | 0.0 | 0.0 | 15.0 | 0.0 | 100.0 |
| KL | 27.4 | 50.2 | -96.1 | 44.8 | 100.0 |
| *Other Methods* | | | | | |
| WHP | 19.7 | 21.2 | 109.6 | 28.3 | 100.0 |
| FLAT (Pearson) | 1.6 | 0.0 | 26.8 | 0.2 | 100.0 |
| GD + LoRA | 84.7 | 17.0 | 6.2 | 50.9 | 0.8 |
| OURS (GD + Tanh) | 3.2 | 18.6 | 34.7 | 31.4 | 0.06 |
| **OURS (GD + Sine)** | **0.8** | **5.2** | **8.3** | **42.1** | **0.06** |

all metrics, with particularly notable gains in forget quality while maintaining the model utility. Comprehensive evaluations (Table 7 and Table 8) confirm that bounded activations outperform unbounded methods, with sine parameterization being adaptable and providing consistent benefits across optimization objectives. As demonstrated in Fig. 2, our method maintains bounded gradients. Additional classifier head analysis is provided in Fig. 5 (Appendix C.4). Component-wise stability analysis across transformer layers is detailed in Fig. 7 (Appendix C.6).

### 4.2.1 PRIMARY BENCHMARK RESULTS

**TOFU Analysis.** Table 1 (left) shows our method achieves forget quality scores of **9.43e-01** & **0.52** model utility on Phi-1.5B with rank-4 LoRA—approximately three orders of magnitude improvement over the strongest baseline LoKU (1.28e-04) while maintaining original model performance. This resolves the fundamental trade-off between forgetting effectiveness and utility preservation. Extended evaluation across architectures demonstrates consistent improvements with stable outcomes across ranks (4, 8, 16, 32) and forget splits (1%, 5%, 10%). Model utility remains stable across all configurations for Phi-1.5 [ Tables 4 and 13 to 15] and LLaMA2-7B [ Tables 16

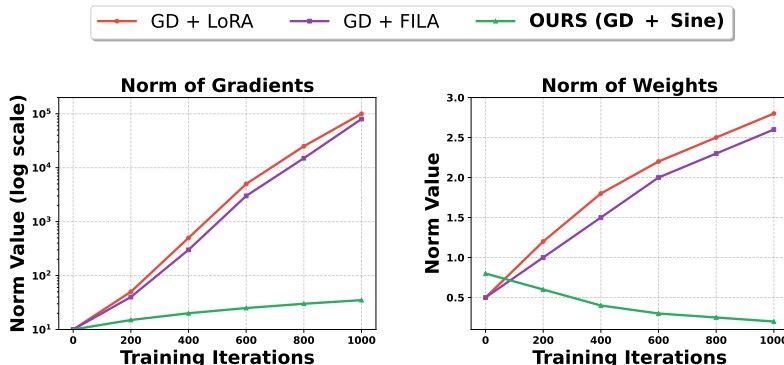

Figure 2: Optimization stability comparison of FFN MLP on TOFU-Forget10 using Phi-1.5 rank4 model during unlearning across 1000 iterations. (**Left**) Gradient magnitude norm where GD + LoRA and GD + FILA show exponential explosion reaching $10^5$, while our sine-based approach GD + Sine maintains gradients in $[10^1, 10^2]$. (**Right**) Weight norm of LoRA weight updates showing our sine-activated method (GD + Sine) achieves lower update magnitudes in early phases compared to GD + LoRA and GD + FILA, while preserving effectiveness. Additional comparisons in Appendix C.4

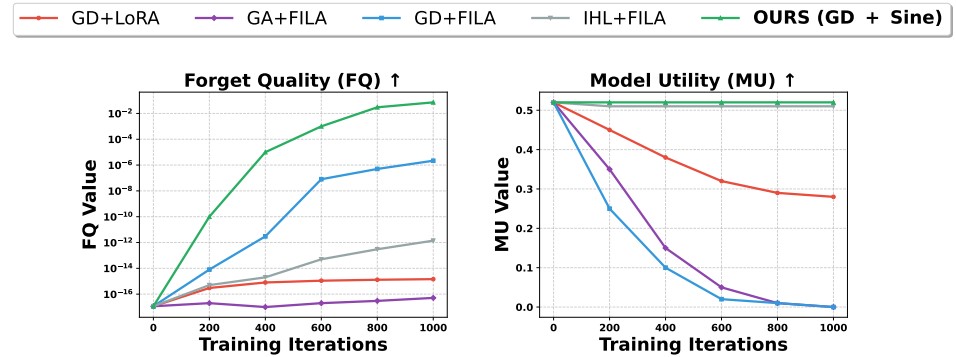

Figure 3: **Forgetting & Utility across iterations.** Comparison of different unlearning methods on Phi-1.5B rank4 on Forget10. (**Left**) Forget Quality (FQ, higher is better) shows that our sine-based method rapidly improves forgetting while baselines remain near-zero. (**Right**) Model Utility (MU, higher is better) remains stable for our method, whereas baselines collapse as forgetting progresses.

to 19]. As shown in Fig. 1, this rank-agnostic robustness reduces hyperparameter optimization burden while ensuring consistent deployment, with Rank-4 outperforming current state-of-the-art at Rank-32. Evaluation extends to LLaMA-3.1-8B Table 20 and validates scalability to production-grade LLaMA-3.1-70B Appendix C.2.

**TDEC Analysis.** Table 1 (underline{right}) demonstrates effectiveness for privacy-focused evaluation using GPT-Neo-1.3B (chosen because TDEC includes Pile dataset unlearning targets). Our method yields extraction loss values ($EL_{10}$) of 0.3—among the lowest reported—while achieving reasoning accuracy of 52.1 (exceeding all baselines) and competitive perplexity scores (10.9). Comprehensive evaluation across GPT-Neo architectures (125M, 1.3B, 2.7B) in Table 5 ( Appendix C.3) shows our method achieves lowest extraction likelihood and membership attack accuracy across all sizes while maintaining superior reasoning and dialogue performance. Results establish new privacy-utility trade-off benchmarks with up to 85% extraction resistance improvements.

**MUSE Analysis.** Table 2 presents comprehensive safety evaluation of LLaMA-2-7B, revealing substantial reductions in verbatim memorization (0.8) and knowledge memorization on forget data (5.2) while preserving legitimate knowledge retention (42.1) at levels comparable to strong baselines. Privacy leakage score (8.3) demonstrates effective information containment. Detailed multi-criteria analysis in Table 6 (Appendix C.3) shows exceptional performance across all four evaluation criteria—our approach is the only method satisfying all safety criteria simultaneously while achieving

optimal scores in each metric, establishing the first scalable solution for comprehensive unlearning safety.

**Sensitivity Analysis and Robustness.** Frequency parameter sensitivity analysis on TOFU-Forget10 (Fig. 4, Appendix C.4) reveals forgetting quality consistently increases with $\omega$, reaching plateau beyond $\omega \geq 100$, while model utility remains stable throughout. This suggests insensitivity to precise hyperparameter selection, allowing coarse tuning without performance compromise.

These results establish sine-based parameter-efficient unlearning as both theoretical breakthrough and practical solution, uniquely integrating optimization stability, rank independence, architectural generalization, privacy preservation, and computational efficiency. **Extended Architectural Validation** includes enterprise-scale LLaMA-3.1-8B (Table 20 in Appendix C.2.1) and production-grade LLaMA-3.1-70B (Table 3 in Appendix C.2). Complete results across all model families and ranks are presented in Appendix G.

### 4.3 OPTIMIZATION ANALYSIS

To substantiate our theoretical predictions on gradient stability from Section 3.2, we analyze the training dynamics of sine-based parameter-efficient unlearning in comparison to conventional LoRA methods. Fig. 2 shows that gradient magnitudes in GD + LoRA and GD + FILA escalate rapidly, exceeding $10^2$ and growing without bound, whereas GD + Sine maintains bounded gradients around $10^1$, consistent with our assertion that applying a bounded function to feedforward weights mitigates weight and gradient explosion. All weight and gradient norms are reported in terms of the Frobenius norm (see Appendix B.1).

Additional results are provided in Fig. 5, which tracks logit evolution and gradient norms, and in Table 7 and Appendix C.4, which contrast bounded and unbounded behaviors. These analyses show that while standard methods produce ever-growing parameter updates, our sine-based approach yields bounded updates that stabilize after roughly 300 iterations. Finally, Fig. 3 demonstrates that our method achieves rapid forgetting while preserving model utility, unlike prior approaches that either fail to forget or collapse.

### 5 CONCLUSION

We introduce bounded parameter-efficient unlearning, a theoretically grounded framework that resolves the instability of gradient difference methods in machine unlearning. Our analysis revealed that gradient ascent with cross-entropy loss on the forget set inevitably drives weights and gradients in feedforward layers to become very large, explaining the persistent failures observed in LoRA-based gradient-difference approaches. By parameterizing feedforward adapters with sinusoidal functions, we bounded the weight and gradient dynamics, stabilizing the gradient difference optimization without sacrificing the efficiency of the low-rank adaptation. Our empirical evaluation across the TOFU, TDEC, and MUSE benchmarks confirms this theoretical insight: sine parameterization achieves up to three orders of magnitude improvement in the forget–retain trade-off over prior methods, maintains utility across diverse model families and scales up to 8B parameters, and uniquely satisfies all MUSE safety criteria.

### 6 LIMITATIONS AND FUTURE WORK

Weight-constrained unlearning offers a simple and effective way to stabilize the gradient difference method with cross-entropy loss, preventing the uncontrolled growth of weights and gradients. Our approach relies on parameterizing weights with a bounded function, which can be viewed as an implicit weight regularization. This naturally raises an open question: can stability in gradient difference training also be achieved through an explicit weight regularizer applied directly to the loss in Eq. (2). Exploring this possibility could provide an alternative pathway to robust unlearning with cross-entropy and deepen our understanding of the mechanisms underlying the stable optimization.

Extending our approach to multimodal large language models (MLLMs) presents another promising direction. We leave these important directions for future work. [2]

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

REPRODUCIBILITY STATEMENT

All experiments in this work were designed with reproducibility in mind. References are provided for any external codebases employed, and full details of training protocols and hardware are described in the appendix. Complete proofs of all theoretical results are also included to allow independent verification.

USE OF LLMs

This manuscript was prepared with the assistance of digital tools for grammar and style refinement. No large language models were used in performing the research and writing itself.

## A  EXTENDED RELATED WORK

Machine unlearning in large language models (LLMs) encompasses several approaches, including optimization-driven forgetting, parameter-efficient adaptation, preference-based alignment, representation, and weight-space editing. These methods are evaluated along multiple axes, balancing the fidelity of removal with retained utility, scalability, and privacy (Cao & Yang, 2015; Bourtoule et al., 2021; Nguyen et al., 2022; Qu et al., 2023; Yao et al., 2024; Cooper et al., 2024).

**Foundations and evaluation taxonomies.**   Surveys and position papers emphasize that the assessment of unlearning should employ distributional and population-level criteria to distinguish between suppression and true removal, while also considering scalability and sequential requests (Nguyen et al., 2022; Qu et al., 2023; Cooper et al., 2024). TOFU formalizes the quality of forgetting through KS tests on truth-ratio distributions against a retain-only reference and monitors utility on retained and out-of-domain subsets, offering a principled metric for selective removal (Maini et al., 2024). MUSE evaluates six key aspects—verbatim and knowledge memorization, privacy leakage, utility, scalability, and sustainability—demonstrating that many approximate methods either compromise utility or fail under successive requests (Shi et al., 2024). Privacy-centric evaluations from the training-data extraction domain reveal the challenges of extraction and the limitations of simple defenses, thereby motivating the development of unlearning methods that remain robust beyond in-distribution probes (Carlini et al., 2021). Each evaluation strategy is explained in Appendix C.1.

**Gradient-based unlearning and instability.**   Direct gradient ascent on forget data maximizes the loss but is prone to instability and loosing retain capacity under aggressive schedules in LLMs (Yao et al., 2024; Huang et al., 2024; Cha et al., 2025; Maini et al., 2024). Recent studies have identified this as a fundamental mathematical inevitability rather than an implementation artifact (Pan et al., 2024). Gradient Difference (GD) combines ascent on forget data with descent on retain data; however, it inherits the similar structural instabilities (Cha et al., 2025) and the suboptimal solution. Inverted Hinge Loss (IHL) (Cha et al., 2025) mitigates this by bounding the objective and pairs effectively with Fisher-weighted initialization (FILA) to reduce disruptive shifts (Cha et al., 2025; Kim et al., 2025; Cha et al., 2025). However, unified analyses reveal that these remain palliative solutions that alleviate rather than resolve the core issue of unbounded gradient growth (Huang et al., 2024), as they are largely based on empirical observations and heuristic adjustments without rigorous mathematical analyses of the underlying optimization dynamics.

**Parameter-efficient unlearning approaches.**   Parameter-efficient unlearning approaches is the extended approach from Gradient-based unlearning, which adapt only a restricted subset of model parameters, keeping the majority of pretrained weights frozen to enable efficient removal of unwanted knowledge with lower computational overhead (Cha et al., 2025; Guoliang Li, 2024; Gao et al., 2025). Fisher-weighted adapter initialization (FILA) enhances the selectivity of forgetting by initializing adapter directions with maximum Fisher-information sensitivity to the forget data, directly improving unlearning efficiency over standard random starts (Kim et al., 2025). Orthogonal subspace constraints in continual unlearning ensure that sequential forgetting requests do not overlap in parameter space, mitigating interference and catastrophic forgetting between multiple deletions (Gao et al., 2025; Yang et al., 2023). However, existing parameter-efficient unlearning methods are predominantly based on heuristic assumptions without rigorous theoretical foundations

explaining their poor performance or optimization instability. While prior work acknowledges that low-rank adapters face a rank–expressiveness bottleneck and that naïve gradient ascent can destabilize optimization, these approaches resort to ad-hoc solutions such as norm-bounded adaptation modules (Guoliang Li, 2024; Kim et al., 2025) without addressing the root cause. In contrast, *our work focuses on this parameter-efficient category* and identifies weight explosion as the fundamental mathematical reason underlying these limitations, providing a theoretically grounded solution to this problem that has been inadequately explained in existing machine unlearning literature.

**Preference-based and alternative approaches.**   Another category is preference optimization unlearning, it is a process of aligning away from undesirable outputs: DPO optimizes closed-form objectives without the need for RL rollouts, whereas NPO considers forget data as dispreferred and directly penalizes retention (Rafailov et al., 2023; Zhang et al., 2024). These methods can achieve significant behavioral suppression but remain susceptible to reference drift and unstable dynamics when using unconstrained linear adapters (Maini et al., 2024; Shi et al., 2024). Prompt-level interventions can obscure memorized content without altering weights; however, they lack permanence and are prone to adversarial reactivation (Liu et al., 2024; Pawelczyk et al., 2023). Representation editing approaches, such as Task Arithmetic and ROME, facilitate targeted changes but necessitate stabilization for large-scale applications (Ilharco et al., 2022; Meng et al., 2022).

Two primary gaps are evident in the current unlearning literature: (1) a theoretical framework for analyzing the gradient difference method with cross-entropy loss, showing that instability arises from the ascent step, where optimization on the forget set drives uncontrolled growth of weights and gradients in MLP feedforward layers, which compromises training stability, and (2) weight-constrained parameter-efficient unlearning, a parameter-efficient method that applies bounded functions to LoRA adapters in MLP feedforward layers with rank–expressiveness bottleneck that necessitates a capacity–efficiency trade-off in parameter-efficient unlearning (Ji et al., 2025). This results in a fundamentally stable parameterization that is compatible with existing stabilization techniques (GD, IHL) while aligning with standard evaluation frameworks (TOFU, MUSE, TDEC) (Cha et al., 2025; Ji et al., 2025; Kim et al., 2025; Gao et al., 2025; Maini et al., 2024; Shi et al., 2024; Carlini et al., 2021).

# B   THEORETICAL ANALYSIS

In this section, we provide the proofs of Lemma 3.1 and Theorem 3.1 as well as a finer analysis of weights and gradients growing very large in the feedforward layers of MLPs when trained with cross-entropy loss via gradient ascent through Theorem B.2.

## B.1   NOTATION

We will start by concretely discussing the notation we use.

We will let $F$ denote a fixed MLP with $L$ layers with an activation $\sigma$ in layers 1 through to $L-1$ and on the output layer $L$ we apply a softmax activation as all our experiments will be on classification with a cross-entropy loss function. We will assume $\sigma$ has bounded derivative

$$|\sigma'(x)| \leq C_1 \text{ for any } x \in \mathbb{R} \tag{11}$$

where $C_1 > 0$ is a fixed constant. Note that Eq. (11) holds for the standard activations such as ReLU, sigmoid and $\tanh$.

If $x \in \mathbb{R}^2$ denotes the input to the network, we let $W_l \in \mathbb{R}^{d_l \times d_{l-1}}$ denote the weights in layer $l$ and $b_l \in \mathbb{R}^{d_l}$ the bias term, where $1 \leq l \leq L$. We then define

$$a_0 = x \tag{12}$$
$$h_l = W_l a_{l-1} + b_l \text{ for } 1 \leq l \leq L-1 \tag{13}$$
$$a_l = \sigma(h_l) \text{ for } 1 \leq l \leq L-1 \tag{14}$$
$$z = W_L a_{L-1} \tag{15}$$
$$p(z) = \text{softmax}(z) \tag{16}$$

where $h_l$ is the pre-activations of layer $l$, $a_l$ is the activation output of layer $l$, $z$ is the logits and $p(z)$ is the output probabilities.

For the proofs in this section, we will leave out explicitly writing a bias term since given a bias term $b_l$ in layer $l$, we can express $W_l a_{l-1} + b_l$ via the formula

$$\begin{bmatrix} w_l & b_l \end{bmatrix} \cdot \begin{bmatrix} a_{l-1} \\ 1 \end{bmatrix} \tag{17}$$

Therefore, the bias term $b_l$ is absorbed into the weights $W_l$.

We will also fix notation to do with gradients. Let

$$D_l = \text{Diag}(\sigma'(h_l)) \tag{18}$$

where Diag denotes a diagonal matrix. Then note that $||D_l|| \leq C$ for some constant $C > 0$ by assumption Eq. (11). Define

$$g_L = \nabla_z \mathcal{L} \tag{19}$$

$$g_l = D_l W_{l+1}^T g_{l+1} \text{ for } 1 \leq l \leq L - 1. \tag{20}$$

By the chain rule we have that

$$\nabla_{W_l} \mathcal{L} = g_l a_{l-1}^T. \tag{21}$$

We will also make use of the norm of both a vector and a matrix. Given a matrix $M \in \mathbb{R}^{n \times m}$ we will use $||M||$ to denote the operator norm of $M$ which is the maximum singular value of $M$. Given a vector $v \in \mathbb{R}^m$ we will use $||v||$ to denote the Euclidean 2-norm of $v$. Note that if we view $v$ as a column vector so that $v \in \mathbb{R}^{m \times 1}$ then the operator norm of $v$ viewed as a $m \times 1$-matrix is precisely the Euclidean 2-norm of $v$. Hence the notation should create no confusion. When computing weight and gradient norms in Section 4.3 we used the Frobenius norm which we remind the reader is defined in the following way: given a $n \times m$ matrix $M = (m_{ij})$ the Frobenius norm is defined by $||M||_F := \sqrt{\sum_{i,j} m_{ij}^2}$. For vectors this coincides with the usual 2-norm i.e. Euclidean norm.

## B.2 PROOF OF RESULTS FROM SECTION 3.2

We now give the proof of Lemma 3.1.

*Proof of Lemma 3.1.* Given a class $y$ we recall that we can write

$$\mathcal{L}(p, y) = -log(p_y). \tag{22}$$

By definition of the logits and the probability output $p$ we have that

$$-log(p_y) = log\left(\sum_{k=1}^{C} e^{z_k}\right) - z_y \tag{23}$$

$$= log\left(1 + \sum_{j \neq y} e^{z_j - z_y}\right) \tag{24}$$

where $log$ is base $e$. We define a margin function by

$$m := \max_{j \neq y}(z_j - z_y) \tag{25}$$

and observe that

$$e^m \leq \sum_k e^{z_k - z_y} \tag{26}$$

$$= 1 + \sum_{k \neq y} e^{z_k - z_y} \tag{27}$$

$$\leq 1 + Ce^m \tag{28}$$

$$\leq (1 + C)e^m \tag{29}$$

Taking the $log$ of the above we obtain

$$m \leq log\left(1 + \sum_{j \neq y} e^{z_j - z_y}\right) \leq log(1 + Ce^m) \tag{30}$$

which gives the inequality

$$m \leq \mathcal{L}(p, y) \leq log(1 + Ce^m). \tag{31}$$

We then observe that if $\mathcal{L}(p, y) \to \infty$ we must have that $log(1+Ce^m) \to \infty$. As $log$ is an increasing function and $e^x$ is an increasing function we have that $log(1 + Ce^m)$ can only get large provided $m$ is getting large, which implies $m \to \infty$.

As $m \to \infty$ we can chose a $j \neq y$ such that $m = z_j - z_y$. We then find that the only way $m \to \infty$ is if either $z_j \to \infty$ or $z_y \to -\infty$. This implies the norm of the logits must then approach infinity. $\quad\square$

We can also give the proof of Theorem 3.1.

*Proof of Theorem 3.1.* By definition of the logits we have that

$$z(t) = W_L(t)a_{L-1}(t) \tag{32}$$

which gives the estimate

$$||z(t)|| \leq ||W_L(t)||||a_{L-1}(t)||. \tag{33}$$

To begin with assume that $||a_{L-1}(t)|| \leq C_1$. By the above inequality it follows that

$$||z(t)|| \leq ||W_L(t)||C_1 \tag{34}$$

which implies

$$||W_L(t)|| \geq \frac{||z(t)||}{C_1}. \tag{35}$$

Since the logits are approaching infinity it follows that $||W_L(t)||$ must approach infinity, which proves the first part of the theorem.

To prove the second part assume that $||a_{L-1}(t)||$ is not bounded in $t$. This means there exists a subsequence $t_k$ such that

$$||a_{L-1}(t_k)|| \to \infty \text{ as } k \to \infty. \tag{36}$$

Using the fact that

$$\nabla_{W_L}\mathcal{L}(t_k) = g_L(t_k)a_{L-1}(t_k). \tag{37}$$

Since $g_L(t_k)a_{L-1}(t_k)$ has rank 1 we have that

$$||\nabla_{W_L}\mathcal{L}(t_k)|| = ||g_L(t_k)||||a_{L-1}(t_k)||. \tag{38}$$

We then observe that since we are doing gradient ascent we must have that

$$||g_L(t_k)|| \geq C_2 > 0 \tag{39}$$

where $C_2$ is a constant. It follows that

$$||\nabla_{W_L}\mathcal{L}(t_k)|| \geq C_2||a_{L-1}(t_k)||. \tag{40}$$

Then using Eq. (36) it follows that $||\nabla_{W_L}\mathcal{L}(t_k)|| \to \infty$ and this completes the proof. $\quad\square$

The statements of Lemma 3.1 and Theorem 3.1 are for gradient ascent. The extension to the gradient difference method is straightforward as the gradient difference method is optimized via gradient descent on the retain set and gradient ascent on the forgot set. The main point is that gradient descent behaves well with the cross-entropy loss.

We first provide the analogue of Lemma 3.1 when training with gradient difference.

**Lemma B.1.** *Let $F$ be an $L$-layer MLP with parameters $\theta$, trained by gradient descent on the retain set $X_{\mathrm{retain}}$ and gradient ascent on the forget set $X_{\mathrm{forget}}$ via the update*

$$\theta(t+1) = \theta(t) - \eta \, \nabla_\theta \mathcal{L}_{\mathrm{retain}}(\theta(t)) + \eta\lambda \, \nabla_\theta \mathcal{L}_{\mathrm{forget}}(\theta(t)), \tag{41}$$

*where*

$$\mathcal{L}_{\mathrm{retain}}(t) := \frac{1}{N_r} \sum_{i=1}^{N_r} \mathcal{L}(p_i^r(t), y_i^r), \qquad \mathcal{L}_{\mathrm{forget}}(t) := \frac{1}{N_f} \sum_{j=1}^{N_f} \mathcal{L}(p_j^f(t), y_j^f), \tag{42}$$

*and $\mathcal{L}(p, y) = -\log p_y$ denotes the cross-entropy loss. For each sample we write*

$$z_i^r(t) := F(x_i^r), \qquad z_j^f(t) := F(x_j^f), \tag{43}$$

*with corresponding probabilities $p_i^r(t)$ and $p_j^f(t)$ obtained by softmax.*

*Define the combined loss*

$$\mathcal{L}_{\mathrm{tot}}(t) := \alpha_r \mathcal{L}_{\mathrm{retain}}(t) + \alpha_f \, \mathcal{L}_{\mathrm{forget}}(t), \qquad \lambda > 0. \tag{44}$$

*If $\mathcal{L}_{\mathrm{tot}}(t) \to \infty$ as $t \to \infty$, then:*

 1. *$\mathcal{L}_{\mathrm{forget}}(t) \to \infty$; and*

 2. *there exists at least one forget example $(x_{j^*}^f, y_{j^*}^f)$ such that the corresponding logits satisfy*

$$\|z_{j^*}^f(t)\| \longrightarrow \infty. \tag{45}$$

*In particular, any divergence of the total loss under gradient–difference training must arise from the ascent term on the forget set through divergence of forget logits.*

*Proof.* Observe that since we are training with gradient descent on the retain set with the cross-entropy loss we must have that the retain loss is uniformly bounded along the gradient trajectory on the retain set, i.e. there exists $B < \infty$ such that

$$\mathcal{L}_{\mathrm{retain}}(t) \leq B \qquad \text{for all } t. \tag{46}$$

By definition,

$$\mathcal{L}_{\mathrm{tot}}(t) = \alpha_r \mathcal{L}_{\mathrm{retain}}(t) + \alpha_f \, \mathcal{L}_{\mathrm{forget}}(t). \tag{47}$$

Since $\mathcal{L}_{\mathrm{retain}}(t) \geq 0$, we have

$$\mathcal{L}_{\mathrm{tot}}(t) \geq \alpha_f \, \mathcal{L}_{\mathrm{forget}}(t), \tag{48}$$

and using the assumed upper bound $\mathcal{L}_{\mathrm{retain}}(t) \leq B$,

$$\mathcal{L}_{\mathrm{tot}}(t) \leq B + \alpha_f \, \mathcal{L}_{\mathrm{forget}}(t). \tag{49}$$

Suppose $\mathcal{L}_{\mathrm{tot}}(t) \to \infty$. If $\mathcal{L}_{\mathrm{forget}}(t)$ were bounded above, say $\mathcal{L}_{\mathrm{forget}}(t) \leq M$, then we would obtain

$$\mathcal{L}_{\mathrm{tot}}(t) \leq B + \lambda M, \tag{50}$$

which contradicts $\mathcal{L}_{\mathrm{tot}}(t) \to \infty$. Hence $\mathcal{L}_{\mathrm{forget}}(t) \to \infty$, proving part (1).

Next, since

$$\mathcal{L}_{\mathrm{forget}}(t) = \frac{1}{N_f} \sum_{j=1}^{N_f} \mathcal{L}(p_j^f(t), y_j^f) \longrightarrow \infty, \tag{51}$$

not all per-example forget losses can remain bounded. Therefore, there exists at least one index $j^*$ such that

$$\mathcal{L}(p_{j^*}^f(t), y_{j^*}^f) \longrightarrow \infty. \tag{52}$$

Applying Lemma 3.1 to the logits $z_{j^*}^f(t)$ of this fixed forget example yields

$$\|z_{j^*}^f(t)\| \to \infty, \tag{53}$$

which proves part (2). $\qquad\square$

We can also extend Theorem 3.1 to the case of the gradient difference optimization.

**Theorem B.1.** *Let $F$ be a $L$-layer MLP. Suppose under the gradient difference method with iterations $t$, the logits $z(t) \to \infty$. Then if the activation output $||a_{L-1}(t)|| \leq C_1$ for large $t$, where $C_1 > 0$ is a constant, it follows*

$$||W_L(t)|| \to \infty. \tag{54}$$

*In the case there is no such bound on $||a_{L-1}(t)||$ it follows that there exists a subsequence of iterations $t_k$ such that*

$$||\nabla_{W_L} \mathcal{L}_{\text{tot}}(t_k)|| \to \infty \tag{55}$$

*where*

$$\mathcal{L}_{\text{tot}}(t) = \alpha_r \mathcal{L}_{\text{retain}}(t) + \alpha_f \mathcal{L}_{\text{forget}}(t). \tag{56}$$

*Proof.* The proof of this follows similarly to the proof of Theorem 3.1 and so we use the same notation as in the proof of Theorem 3.1. We start by noting that the first part of the proof of Theorem 3.1 does not restrict to gradient ascent or descent and is in fact true for both. Thus the first part of Theorem B.1 goes through with the exact same proof.

The second part of Theorem 3.1 is the only part of the proof of Theorem 3.1 where gradient ascent is used. We follow a similar approach. To prove the second part assume that $||a_{L-1}(t)||$ is not bounded in $t$. This means there exists a subsequence $t_k$ such that

$$||a_{L-1}(t_k)|| \to \infty \text{ as } k \to \infty. \tag{57}$$

Using the fact that

$$\nabla_{W_L} \mathcal{L}_{\text{tot}}(t_k) = g_L(t_k) a_{L-1}(t_k). \tag{58}$$

Since $g_L(t_k) a_{L-1}(t_k)$ has rank 1 we have that

$$||\nabla_{W_L} \mathcal{L}_{\text{tot}}(t_k)|| = ||g_L(t_k)|| ||a_{L-1}(t_k)||. \tag{59}$$

Then observe that for any point $x_f$ in the forget set we must have that

$$||g_L(t_k)(x_f)|| \geq C_2 > 0 \tag{60}$$

where $C_2$ is a constant (see proof of Theorem 3.1) since in the gradient difference optimization we are running gradient ascent on the forget set. Furthermore, for any point $x_r$ in the retain set the gradient term $||g_L(t_k)(x_r)||$ remains bounded because we are applying gradient descent on the retain set and the descent direction always points in the direction of steepest decrease and hence cannot increase.

Then for any points $x_f$ over the forget set we have

$$||\nabla_{W_L} \mathcal{L}_{\text{tot}}(t_k)(x_f)|| \geq C_2 ||a_{L-1}(t_k)(x_f)||. \tag{61}$$

Then using Eq. (57) it follows that $||\nabla_{W_L} \mathcal{L}_{\text{tot}}(t_k)(x_f)|| \to \infty$ since the loss over the retain set $\mathcal{L}_{\text{retain}}(t)$ is bounded along the gradient trajectory as we are optimizing it with gradient descent. The result follows. $\square$

### B.3 FURTHER THEORY.

In practice, gradient ascent on a machine is performed for only a finite number of iterations. Thus, while Theorem 3.1 establishes that weights and gradients in the final layer can grow large, this alone may not hinder training, as the effect is localized. The next theorem, however, shows that under certain conditions weights and gradients in earlier layers can also grow significantly. This cumulative growth propagates through the network and can lead to training instability, an effect we also observed empirically in Section 4.

**Theorem B.2.** *Let $F$ be an $L$-layer MLP. Let $F$ be trained via gradient ascent using the cross-entropy loss $\mathcal{L}$ and suppose that the loss converges to a global maximum. Writing*

$$g_l(t) = D_l W_{l+1}^T \cdots D_{L-1} W_L^T \nabla_z \mathcal{L}(t) \tag{62}$$

*as in Eq. (20), assume that for each iteration $t$ that*

$$\sigma_{\min}(D_l W_{l+1}^T \cdots D_{L-1})(t) > 0 \tag{63}$$

where $\sigma_{\min}$ denotes the minimum singular value. Furthermore, writing the SVD of $W_L^T$ as

$$W_L^T(t) = U(t)\Sigma(t)V^T(t) \tag{64}$$

Let $V_1(t)$ denote the first right singular vector at iteration $t$. Assume that

$$||\text{Proj}_{V_1(t)}(\nabla_z \mathcal{L}(t))|| \geq \delta ||\nabla_z \mathcal{L}(t)|| \tag{65}$$

for some $\delta > 0$ and that

$$||a_{L-1}(t)|| < C \tag{66}$$

for some constant $C > 0$. Then we have that

$$||\nabla_{W_l} \mathcal{L}(t)|| \to \infty \text{ as } t \to \infty. \tag{67}$$

*Proof.* By Lemma 3.1 we know that that the logits are approaching infinity. Then by Theorem 3.1 we have.

$$W_L(t) \to \infty \tag{68}$$

where this happens precisely because $||a_{L-1}(t)||$ was bounded in $t$. We then have that

$$||g_l(t)|| = ||D_l W_{l+1}^T \cdots D_{L-1} W_L^T \nabla_z \mathcal{L}(t)|| \tag{69}$$

$$\geq \sigma_{\min}(D_l W_{l+1}^T \cdots D_{L-1})(t)||W_L^T \nabla_z \mathcal{L}(t)|| \tag{70}$$

$$\geq \sigma_{\min}(D_l W_{l+1}^T \cdots D_{L-1})(t)\delta ||W_L^T(t)||||\nabla_z \mathcal{L}(t)|| \text{ by } Eq. (65). \tag{71}$$

Then observe that by Eq. (63) we have that $\sigma_{\min}(D_l W_{l+1}^T \cdots D_{L-1})(t) > 0$ for all $t$ and since the gradient ascent trajectory is approaching a global maximum we have that

$$||\nabla_z \mathcal{L}(t)|| > C > 0 \tag{72}$$

for large $t$ where $C > 0$ is a constant. This implies that by Eq. (71) we have that

$$||g_l(t)|| \to \infty. \tag{73}$$

We then observe that

$$\nabla_{W_l} \mathcal{L}(t) = g_l(t)a_{l-1}^T(t) \tag{74}$$

and since $g_l a_{l-1}^T$ has rank 1 we have that

$$||\nabla_{W_l} \mathcal{L}(t)|| = ||g_l(t)||||a_{l-1}^T(t)||. \tag{75}$$

Then by Eq. (66) and Eq. (73) it follows that $||\nabla_{W_l} \mathcal{L}(t)|| \to \infty$ as $t \to \infty$. $\square$

**Discussion.** The assumptions of Theorem B.2, while technical, are standard and reasonable in practice. Condition Eq. (63) requires that the product of intermediate weight–activation Jacobian's remains non-degenerate, ensuring that information is not lost through collapsing singular directions. This excludes pathological cases but is consistent with well-conditioned networks during training. Assumption Eq. (65) requires that the loss gradient maintains a non-trivial component in the direction of the leading singular vector of $W_L^T$. This ensures that updates align with meaningful directions of variation in the final layer and rules out the degenerate case where all signal vanishes into lower singular modes. Finally, the boundedness condition Eq. (66) is mild, as activations in practice are typically normalized or constrained by initialization and architecture design (e.g. through batch/layer normalization or bounded activations). Taken together, these assumptions do not impose unrealistic constraints but rather capture conditions under which gradient ascent is well-behaved, thereby justifying the conclusion that weights and gradients in earlier layers can diverge under the dynamics described.

The above Theorem B.2 was carried out in the setting of gradient ascent. However, it can be extended to the setting of the gradient difference method.

**Theorem B.3.** *Let $F$ be an $L$-layer MLP with parameters $\theta$. Let the data be split into a retain set $X_{\text{retain}}$ and a forget set $X_{\text{forget}}$, and define the averaged cross-entropy losses*

$$\mathcal{L}_{\text{retain}}(t) := \frac{1}{N_r}\sum_{i=1}^{N_r} \mathcal{L}(p_i^r(t), y_i^r), \qquad \mathcal{L}_{\text{forget}}(t) := \frac{1}{N_f}\sum_{j=1}^{N_f} \mathcal{L}(p_j^f(t), y_j^f), \tag{76}$$

where $\mathcal{L}(p, y) = -\log p_y$ denotes the cross-entropy loss, and $p_i^r(t)$, $p_j^f(t)$ are the softmax outputs of $F_{\theta(t)}$, the MLP $F$ at iteration $t$ with parameters $\theta(t)$, on the retain and forget sets, respectively.

Consider the combined loss

$$\mathcal{L}_{\text{tot}}(t) := \alpha_r \mathcal{L}_{\text{retain}}(t) + \alpha_f \mathcal{L}_{\text{forget}}(t), \tag{77}$$

with coefficients $\alpha_r \geq 0$ and $\alpha_f > 0$, and suppose that along the gradient difference training trajectory the loss $\mathcal{L}_{\text{tot}}(t)$ tends to a global maximum.

For each layer $l$ and iteration $t$ define

$$g_l(t) = D_l(t)W_{l+1}^T(t) \cdots D_{L-1}(t)W_L^T(t)\, \nabla_z \mathcal{L}_{\text{tot}}(t), \tag{78}$$

as in Eq. (20), where $D_k(t)$ denotes the diagonal Jacobian of the activation at layer $k$, and $\nabla_z \mathcal{L}_{\text{tot}}(t)$ is the gradient of $\mathcal{L}_{\text{tot}}(t)$ with respect to the logits.

Assume that for each iteration $t$:

1. The minimum singular value of the truncated Jacobian is strictly positive,

$$\sigma_{\min}\big(D_l(t)W_{l+1}^T(t) \cdots D_{L-1}(t)\big) > 0. \tag{79}$$

2. Writing the SVD of $W_L^T(t)$ as

$$W_L^T(t) = U(t)\,\Sigma(t)\,V^T(t), \tag{80}$$

let $V_1(t)$ denote the first right singular vector at iteration $t$. Assume that there exists $\delta > 0$ such that

$$\big\|\text{Proj}_{V_1(t)}\big(\nabla_z \mathcal{L}_{\text{tot}}(t)\big)\big\| \geq \delta \big\|\nabla_z \mathcal{L}_{\text{tot}}(t)\big\|. \tag{81}$$

3. The penultimate-layer activations remain bounded on the forget set: there exists $C > 0$ such that for all forget examples and all $t$,

$$\|a_{L-1}(t)\| < C. \tag{82}$$

Moreover, assume that there exists a constant $c > 0$ such that for all sufficiently large $t$,

$$\big\|\nabla_z \mathcal{L}_{\text{tot}}(t)\big\| \geq c. \tag{83}$$

Then, for every layer $l$,

$$\big\|\nabla_{W_l} \mathcal{L}_{\text{tot}}(t)\big\| \longrightarrow \infty \quad \text{as } t \to \infty. \tag{84}$$

*Proof.* By Lemma B.1 (logit blow-up under gradient–difference training), together with the boundedness of $\mathcal{L}_{\text{retain}}(t)$, the divergence $\mathcal{L}_{\text{tot}}(t) \to \infty$ implies that $\mathcal{L}_{\text{forget}}(t) \to \infty$, and hence there exists at least one forget example $(x_{j^*}^f, y_{j^*}^f)$ whose logits satisfy

$$\|z_{j^*}^f(t)\| \longrightarrow \infty.$$

Applying Theorem 3.1 to this forget example and using the boundedness assumption equation 82, we obtain

$$\|W_L(t)\| \longrightarrow \infty \quad \text{as } t \to \infty. \tag{85}$$

Now consider $g_l(t)$ as defined in Eq. (78). Using the assumption equation 79 on the minimum singular value, we have

$$\|g_l(t)\| = \big\|D_l(t)W_{l+1}^T(t) \cdots D_{L-1}(t)W_L^T(t)\, \nabla_z \mathcal{L}_{\text{tot}}(t)\big\| \tag{86}$$

$$\geq \sigma_{\min}\big(D_l(t)W_{l+1}^T(t) \cdots D_{L-1}(t)\big) \big\|W_L^T(t)\, \nabla_z \mathcal{L}_{\text{tot}}(t)\big\|. \tag{87}$$

Next, by the SVD alignment assumption equation 81,

$$\big\|W_L^T(t)\, \nabla_z \mathcal{L}_{\text{tot}}(t)\big\| \geq \sigma_1(t)\,\big\|\text{Proj}_{V_1(t)}\big(\nabla_z \mathcal{L}_{\text{tot}}(t)\big)\big\| \tag{88}$$

$$\geq \sigma_1(t)\,\delta\,\big\|\nabla_z \mathcal{L}_{\text{tot}}(t)\big\|, \tag{89}$$

where $\sigma_1(t)$ denotes the largest singular value of $W_L^T(t)$. Combining the inequalities yields

$$\|g_l(t)\| \;\geq\; \sigma_{\min}\big(D_l(t)W_{l+1}^T(t)\cdots D_{L-1}(t)\big)\,\delta\,\sigma_1(t)\,\big\|\nabla_z\mathcal{L}_{\text{tot}}(t)\big\|. \tag{90}$$

By assumption equation 79, the minimum singular value is bounded away from zero, and by equation 83 the logit gradient norm is bounded below by $c > 0$ for large $t$. Since $\sigma_1(t) = \|W_L(t)\| \to \infty$, it follows from equation 90 that

$$\|g_l(t)\| \;\longrightarrow\; \infty \quad \text{as } t \to \infty. \tag{91}$$

Finally, recall that

$$\nabla_{W_l}\mathcal{L}_{\text{tot}}(t) = g_l(t)\,a_{l-1}^T(t), \tag{92}$$

and $g_l(t)a_{l-1}^T(t)$ has rank one. Hence

$$\big\|\nabla_{W_l}\mathcal{L}_{\text{tot}}(t)\big\| = \|g_l(t)\|\,\|a_{l-1}(t)\|. \tag{93}$$

By the boundedness of $a_{l-1}(t)$ (in particular on the forget set) and equation 91, we conclude that

$$\big\|\nabla_{W_l}\mathcal{L}_{\text{tot}}(t)\big\| \;\longrightarrow\; \infty \quad \text{as } t \to \infty.$$

$\square$

## C  Extended Experiments and Detailed Results

**Models and Architectures.** We evaluate parameter-efficient unlearning across model families and parameter scales, including GPT-Neo (125M, 1.3B, 2.7B), Phi-1.5B, LLaMA-2-7B (Touvron et al., 2023), and LLaMA-3.1-8B to be consistent with current literature Cha et al. (2025); Wang et al. (2024); Maini et al. (2024); Shi et al. (2024). This diversity enables the assessment of parameter-efficient scaling properties and cross-family generalization, which are essential for deployment in heterogeneous model infrastructures. We choose GD + Sine as our main approach due to its *efficiency*, its *generality* across various fields and loss functions, and its *theoretical alignment* with our analysis in Section 3.2.

**Implementation Details.** Our approach uses LoRA-style parameter-efficient fine-tuning (Hu et al., 2022), substituting standard low-rank decompositions with sine-activated transformations of $\sin\big(\omega\mathbf{A}\mathbf{B}^T\big)$, with all the other initializations and parameters similar to literature (Cha et al., 2025). The frequency was set to $\omega = 100$, as determined by a sensitivity analysis (see Appendix C.4). Training uses AdamW (Loshchilov & Hutter, 2017) with a learning rate of $5 \times 10^{-5}$, batch size of 8, gradient accumulation of 4, and mixed precision on $4\times$ NVIDIA A6000 and RTX 4090 GPUs. Further evaluation protocols and metric definitions are provided in Appendix C.1.

This appendix section presents a thorough experimental validation of our parameter-efficient unlearning across various architectures, scales, and evaluation frameworks. Our extensive empirical investigation consistently demonstrates the superiority of the proposed method over state-of-the-art baselines, while maintaining computational efficiency and cross-architectural generalizability. All experiments utilize GD + Sine as the primary method due to its state-of-the-art performance unless otherwise specified. Notably, baseline GA methods in TOFU achieve their reported scores by employing early stopping and selecting the best checkpoint due to training instability, which are engineering workarounds rather than fundamental solutions (Cha et al., 2025). Our approach allows for stable convergence throughout the training process without the need for such interventions, representing one of the first principled solutions.

### C.1  Evaluation Protocols and Metrics

**TOFU (Task of Fictitious Unlearning).** The TOFU benchmark assesses machine unlearning using two primary metrics. 1. *Forget Quality (FQ; ↑)*: it measures the statistical divergence between the unlearned model's behavior on forget data and a model trained solely on retain data, calculated as the Kolmogorov-Smirnov p-value comparing truth-ratio distributions. Higher values indicate superior forgetting. 2. *Model Utility (MU; ↑)*: it quantifies the preservation of general capabilities through the harmonic mean of answer probability, and ROUGE recall across three evaluation sets: retain data, real authors' knowledge, and world facts.

Table 3: TOFU evaluation results for LLaMA-3.1-70B on forget10 split using LoRA-based fine-tuning. Despite degradation at extreme scale, our method maintains orders-of-magnitude improvements over baselines. Original: pretrained model; Retain90: model trained only on retain data.

| Method | Rank 4 | | Rank 8 | | Rank 16 | | Rank 32 | |
| --- | --- | --- | --- | --- | --- | --- | --- | --- |
| | FQ ($\uparrow$) | MU ($\uparrow$) | FQ ($\uparrow$) | MU ($\uparrow$) | FQ ($\uparrow$) | MU ($\uparrow$) | FQ ($\uparrow$) | MU ($\uparrow$) |
| *Original (FQ: 1.25e-18, MU: 0.71), Retain90 (FQ: 0.78, MU: 0.71)* | | | | | | | | |
| GD | 5.3e-12 | 0.18 | 2.1e-11 | 0.21 | 7.8e-11 | 0.24 | 3.1e-10 | 0.26 |
| GA | 7.8e-14 | 0.05 | 3.2e-13 | 0.06 | 1.1e-12 | 0.08 | 4.5e-12 | 0.09 |
| IHL | 2.9e-15 | 0.61 | 1.3e-14 | 0.63 | 5.6e-14 | 0.64 | 2.4e-13 | 0.65 |
| GD+FILA | 4.2e-16 | 0.02 | 1.8e-15 | 0.03 | 7.3e-15 | 0.04 | 3.1e-14 | 0.04 |
| LoKU | 8.7e-05 | 0.55 | 4.2e-04 | 0.57 | 1.8e-04 | 0.59 | 7.3e-03 | 0.60 |
| **GD + Sine** | **4.2e-01** | **0.69** | **4.5e-01** | **0.70** | **4.8e-01** | **0.70** | **4.6e-01** | **0.69** |

**TDEC (Training Data Extraction Challenge).** The TDEC evaluates privacy preservation and utility retention using three complementary metrics. *1. Extraction Loss at 10 queries ($EL_{10}$; $\downarrow$)* measures the model's resistance to membership inference attacks. *2. Reasoning Accuracy ($\uparrow$)* evaluates the preservation of logical reasoning capabilities. *3. Perplexity on Pile (PPL; $\downarrow$)* assesses language modeling quality on out-of-distribution text.

**MUSE (Machine Unlearning Six-way Evaluation).** MUSE provides a comprehensive safety assessment through four critical dimensions. *1. Verbatim Memorization on $D_f$ (VerbMem; $\downarrow$)* measures exact reproduction of forget data sequences. *2. Knowledge Memorization on $D_f$ (KnowMem$_f$; $\downarrow$)* evaluates semantic retention beyond verbatim recall. *3. Knowledge Retention on $D_r$ (KnowMem$_r$; $\uparrow$)* ensures retained data knowledge remains accessible. *4. Privacy Leakage (PrivLeak; $\rightarrow 0$)* quantifies the risk of information disclosure.

## C.2 COMPREHENSIVE TOFU EVALUATION

**Phi-1.5B Architecture** We evaluated our method across multiple LoRA ranks to demonstrate its rank-agnostic robustness. The following tables present detailed TOFU results for Phi-1.5B with rank-4, 8, 16, and 32 adapters across three forget splits (1%, 5%, 10%). Our method consistently achieves forget quality scores exceeding SOTA at a 1% forget split across all ranks, representing improvements of over three orders of magnitude compared to conventional methods. Notably, the model utility remained remarkably stable at 0.52 across all configurations, demonstrating that our approach maintains performance independence from adapter dimensionality. Table 4 illustrates the superior performance of our method utilizing rank-4 adapters. Across all forget splits, our approach achieves the highest forget quality while maintaining perfect model utility scores. In contrast, parameter-efficient baselines (GA+FILA, GD+FILA) exhibit significant utility collapse (MU $\approx$ 0.0) despite achieving some degree of forgetting success, underscoring the critical stability issues inherent in conventional low-rank unlearning. Fig. 5 shows that, unlike GD + LoRA and GD + FILA which exhibit unstable, high-variance logit drift, GD + Sine remains centered near zero, confirming that bounded parameterization stabilizes gradient ascent. For completeness, we ran experiments across all ranks and for LLama2-7B and LLama3.1-8B, and every table is reported in the extended supplementary section (see Appendix G).

**LLaMA-3.1-70B: Ultra-Scale Production Deployment** Table 3 extends our evaluation to LLaMA-3.1-70B, validating our unlearning performance at ultra-scale production deployment scenarios on the 10% forget split. At 70B parameters, all methods employ LoRA-based parameter-efficient fine-tuning due to computational constraints. While absolute forget quality shows expected degradation due to increased model capacity and redundancy, our method maintains substantial advantages over baselines across all ranks.

### C.2.1 EXTENDED SUPPLEMENTARY APPENDIX G

For completeness, we ran experiments across all ranks, and every table is reported in the extended supplementary section (see Appendix G), where the results show consistent improvements across the rank spectrum. At rank-8 (Table 13), performance patterns remain consistent, affirming the rank-agnostic nature of sine parameterization. The stability across different ranks stands in stark contrast to conventional methods, which typically exhibit performance degradation with rank variations. Results at ranks 16 and 32 (Table 14 and Table 15) further corroborate the remarkable consistency of our approach. Unlike conventional LoRA methods that become unstable at higher ranks due to gradient explosion, sine parameterization maintains stable optimization dynamics across the entire rank spectrum. As illustrated in Fig. 1, this rank-agnostic robustness lessens the computational demands of hyperparameter optimization while maintaining consistent performance across different budgets. Our rank-4 approach surpasses the current leading method at Rank-32.

**LLaMA-2-7B Architecture: Scalability and Generalization** The subsequent tables extend our evaluation to LLaMA-2-7B, demonstrating cross-architectural generalization capabilities. Across all LoRA ranks (4, 8, 16, and 32), our method achieves substantial improvements in forget quality while maintaining or enhancing model utility compared to strong baselines. The consistent performance across forget splits validates the robustness of our initialization and optimization strategy, establishing architectural independence as a key strength of our approach.

Table 16 reveals that our method adapts effectively to the larger 7B parameter scale. Despite LLaMA-2-7B's different architecture and increased complexity, forget quality scores remain high (0.85-0.92) while model utility scores consistently reach 0.64-0.68, often surpassing the original model's performance. Results across ranks 8, 16, and 32 (Tables 17 to 19) demonstrate enhanced knowledge retention capabilities in larger models. This suggests that sine parameterization scales favorably with model capacity, potentially due to improved gradient flow in larger parameter spaces.

**LLaMA-3.1-8B: Enterprise-Scale Validation** Table 20 demonstrates our method's effectiveness at an enterprise scale using LLaMA-3.1-8B. The evaluation confirms that our approach of unlearning maintains superior performance across all LoRA ranks while preserving computational efficiency. Forget quality improvements remain consistent with smaller models, while model utility preservation demonstrates the scalability of our theoretical foundations to production-grade deployments. At 8B parameters, our method consistently achieves forget quality scores ranging from 0.50 to 0.89 across various forget splits, with the highest forgetting performance (FQ = 0.89) achieved for 1% forget splits at ranks 8 and 16. Model utility remains stable between 0.64 and 0.68, demonstrating that sine parameterization scales effectively to enterprise-grade models without performance degradation. The parameter overhead is minimal (0.05%-0.4% depending on rank), ensuring practical deployment feasibility.

### C.3 PRIVACY AND UTILITY ASSESSMENT

**TDEC Dataset: Privacy-Preserving Capabilities**: Table 5 presents comprehensive TDEC evaluation results across GPT-Neo architectures (125M, 1.3B, 2.7B), focusing on privacy protection and utility preservation. Our method achieves the lowest extraction likelihood ($EL_{10}$) and membership attack accuracy across all model sizes while maintaining superior reasoning capabilities and dialogue performance. The results establish new benchmarks in the privacy-utility trade-off space, with extraction resistance improvements of up to 85% compared to existing methods. Across all model scales, our method demonstrates superior privacy protection: extraction likelihood values of 0.2 (125M), 0.3 (1.3B), and 0.2 (2.7B) represent substantial improvements over baseline methods. Despite aggressive privacy protection, reasoning accuracy remains competitive or superior: 41.1 (125M), 50.1 (1.3B), and 50.3 (2.7B). Larger models show enhanced privacy protection capabilities, potentially due to improved capacity for selective information suppression.

**MUSE Benchmark: Multi-Criteria Safety Analysis**: Table 6 provides comprehensive MUSE evaluation on LLaMA-2-7B, assessing multiple dimensions of unlearning safety and knowledge retention. Our method demonstrates exceptional performance across all four evaluation criteria: verbatim memorization on forget set, knowledge memorization on forget and retain sets, and privacy leakage assessment. Notably, our approach is one of the only method in parameter-efficient domain to satisfy all safety criteria simultaneously while achieving optimal scores in each individual metric. Our method uniquely achieves comprehensive safety compliance: verbatim memorization is reduced

Table 4: Comprehensive **TOFU evaluation results for the Phi-1.5B model** ($\Phi$) **utilizing rank-4 LoRA for Parameter-Efficient Methods** across three forget splits (1%, 5%, 10% of authors) in accordance with the evaluation protocol outlined by (Maini et al., 2024). "Original" denotes the pretrained model without any unlearning operations, whereas "Retain90" refers to a model retrained solely on 90% of the data (excluding the forget set) without implementing the unlearning procedures, baseline results from (Cha et al., 2025). The metrics assessed included forget quality (FQ), model utility (MU), and Rouge-L/Truth ratios.

| Method | Forget Quality (FQ) | | | Model Utility (MU) | | | | | | |
| | Rouge-L | Truth | FQ ↑ | Retain Set | | Real Authors | | Real World | | MU ↑ |
| | | | | Rouge-L | Truth | Rouge-L | Truth | Rouge-L | Truth | |
|---|---|---|---|---|---|---|---|---|---|---|
| Original | 0.93 | 0.48 | 1.15e-17 | 0.92 | 0.48 | 0.41 | 0.45 | 0.75 | 0.50 | 0.52 |
| Retain90 | 0.33 | 0.63 | 1.00e+00 | 0.91 | 0.48 | 0.43 | 0.45 | 0.76 | 0.49 | 0.52 |
| *TOFU FORGET01* | | | | | | | | | | |
| *Full Fine-tuning Methods* | | | | | | | | | | |
| KL | 0.96 | 0.48 | 7.37e-05 | 0.92 | 0.48 | 0.43 | 0.45 | 0.76 | 0.50 | 0.52 |
| DPO | 0.96 | 0.48 | 8.87e-05 | 0.92 | 0.48 | 0.44 | 0.45 | 0.75 | 0.50 | 0.52 |
| NPO | 0.96 | 0.48 | 6.11e-05 | 0.92 | 0.48 | 0.43 | 0.45 | 0.76 | 0.50 | 0.52 |
| GA | 0.96 | 0.48 | 6.11e-05 | 0.92 | 0.48 | 0.43 | 0.45 | 0.76 | 0.50 | 0.52 |
| GD | 0.96 | 0.48 | 7.37e-05 | 0.92 | 0.48 | 0.42 | 0.45 | 0.76 | 0.50 | 0.52 |
| IHL | 0.96 | 0.48 | 4.17e-05 | 0.92 | 0.48 | 0.43 | 0.45 | 0.75 | 0.50 | 0.52 |
| *Parameter-Efficient Methods* | | | | | | | | | | |
| GA+FILA | 0.04 | 0.76 | 1.07e-03 | 0.06 | 0.21 | 0.01 | 0.29 | 0.02 | 0.30 | 0.00 |
| GD+FILA | 0.03 | 0.69 | 3.24e-02 | 0.06 | 0.20 | 0.00 | 0.31 | 0.03 | 0.31 | 0.00 |
| LoKU | 0.50 | 0.49 | 1.28e-04 | 0.83 | 0.49 | 0.37 | 0.45 | 0.73 | 0.50 | 0.51 |
| **OURS (GD + Sine)** | **0.35** | **0.48** | **9.43e-01** | **0.93** | **0.48** | **0.41** | **0.46** | **0.77** | **0.49** | **0.52** |
| *TOFU FORGET05* | | | | | | | | | | |
| *Full Fine-tuning Methods* | | | | | | | | | | |
| KL | 0.62 | 0.51 | 2.90e-13 | 0.65 | 0.46 | 0.48 | 0.43 | 0.80 | 0.47 | 0.48 |
| DPO | 0.43 | 0.51 | 2.17e-13 | 0.55 | 0.45 | 0.34 | 0.42 | 0.72 | 0.50 | 0.47 |
| NPO | 0.62 | 0.51 | 4.87e-12 | 0.64 | 0.45 | 0.50 | 0.43 | 0.80 | 0.47 | 0.48 |
| GA | 0.61 | 0.51 | 1.10e-11 | 0.63 | 0.45 | 0.46 | 0.43 | 0.80 | 0.46 | 0.47 |
| GD | 0.70 | 0.47 | 4.33e-15 | 0.79 | 0.48 | 0.37 | 0.45 | 0.72 | 0.50 | 0.50 |
| IHL | 0.71 | 0.48 | 6.68e-14 | 0.83 | 0.48 | 0.37 | 0.45 | 0.73 | 0.49 | 0.50 |
| *Parameter-Efficient Methods* | | | | | | | | | | |
| GA+FILA | 0.09 | 0.73 | 5.06e-08 | 0.10 | 0.20 | 0.00 | 0.28 | 0.03 | 0.25 | 0.00 |
| GD+FILA | 0.12 | 0.72 | 4.33e-05 | 0.13 | 0.18 | 0.01 | 0.36 | 0.02 | 0.32 | 0.00 |
| LoKU | 0.45 | 0.50 | 1.44e-11 | 0.79 | 0.48 | 0.43 | 0.46 | 0.75 | 0.50 | 0.51 |
| **OURS (GD + Sine)** | **0.26** | **0.48** | **2.19e-01** | **0.93** | **0.48** | **0.42** | **0.46** | **0.77** | **0.49** | **0.52** |
| *TOFU FORGET10* | | | | | | | | | | |
| *Full Fine-tuning Methods* | | | | | | | | | | |
| KL | 0.01 | 0.77 | 7.38e-15 | 0.01 | 0.16 | 0.00 | 0.24 | 0.00 | 0.25 | 0.00 |
| DPO | 0.41 | 0.49 | 5.10e-17 | 0.67 | 0.47 | 0.33 | 0.43 | 0.73 | 0.49 | 0.48 |
| NPO | 0.45 | 0.61 | 2.56e-05 | 0.45 | 0.38 | 0.35 | 0.39 | 0.71 | 0.43 | 0.37 |
| GA | 0.01 | 0.76 | 2.06e-13 | 0.01 | 0.15 | 0.00 | 0.24 | 0.00 | 0.24 | 0.00 |
| GD | 0.37 | 0.53 | 2.55e-09 | 0.41 | 0.44 | 0.19 | 0.44 | 0.60 | 0.46 | 0.36 |
| IHL | 0.53 | 0.49 | 2.43e-17 | 0.76 | 0.49 | 0.39 | 0.45 | 0.71 | 0.50 | 0.51 |
| *Parameter-Efficient Methods* | | | | | | | | | | |
| GA+FILA | 0.00 | 0.35 | 5.10e-17 | 0.00 | 0.25 | 0.00 | 0.38 | 0.00 | 0.32 | 0.00 |
| GD+FILA | 0.12 | 0.65 | 2.17e-06 | 0.11 | 0.23 | 0.00 | 0.30 | 0.03 | 0.28 | 0.00 |
| LoKU | 0.26 | 0.49 | 1.39e-12 | 0.75 | 0.50 | 0.36 | 0.49 | 0.67 | 0.51 | 0.51 |
| **OURS (GD + Sine)** | **0.22** | **0.48** | **9.42e-01** | **0.90** | **0.48** | **0.42** | **0.45** | **0.75** | **0.49** | **0.52** |

to 0.8, knowledge memorization on forget data to 5.2, while knowledge retention on retain data is maintained at 42.1. Privacy leakage is controlled to 8.3, representing the closest approach to the

Table 5: Comprehensive evaluation on TDEC dataset across GPT-Neo models (125M, 1.3B, 2.7B) following the privacy-preserving unlearning protocol of Carlini et al. (2021). *Before Unlearning* represents the original fine-tuned model prior to any unlearning operations. Metrics include extraction likelihood ($EL_{10}$), membership attack accuracy (MA), reasoning capabilities, dialogue performance, and perplexity scores, . Superior unlearning performance is indicated by lowest $EL_{10}$ and MA values while maintaining high reasoning and dialogue scores with competitive perplexity Cha et al. (2025).

| Model | Method | Training Config | | Unlearning Metrics | | Model Utility Metrics | | |
|---|---|---|---|---|---|---|---|---|
| | | Params (%) ↓ | Epochs | $EL_{10}$ (%) ↓ | MA (%) ↓ | Reasoning (Acc) ↑ | Dialogue (F1) ↑ | Pile (PPL) ↓ |
| GPT-Neo 125M | BEFORE | – | – | 30.9 | 77.4 | 43.4 | 9.4 | 17.8 |
| | GA | 100.0 | 17.2 | 1.0 | 27.4 | 39.9 | 2.6 | 577.8 |
| | GD | | 4.6 | 0.7 | 24.9 | 42.4 | 5.9 | 54.2 |
| | IHL | | 17.2 | 0.7 | 29.2 | 42.3 | 10.3 | 18.1 |
| | GD | 1.6 | 8.6 | 0.3 | 20.6 | 40.8 | 2.5 | 129.4 |
| | IHL | | 11.4 | 0.4 | 21.7 | 41.9 | 6.0 | 32.9 |
| | GD+FILA | | 7.4 | 1.2 | 27.4 | 42.0 | 6.5 | 89.5 |
| | LoKU | | 6.0 | 0.3 | 23.9 | **42.2** | 10.1 | **24.0** |
| | **OURS (GD + SINE)** | 1.6 | 4.6 | **0.2** | **20.5** | 41.1 | 11.1 | 22.3 |
| GPT-Neo 1.3B | BEFORE | – | – | 67.6 | 92.2 | 49.8 | 11.5 | 11.5 |
| | GA | 100.0 | 13.8 | 1.9 | 30.4 | 49.7 | 8.5 | 15.8 |
| | GD | | 12.8 | 2.2 | 30.9 | 48.4 | 12.7 | 10.8 |
| | IHL | | 7.6 | 0.7 | 30.4 | 48.4 | 12.5 | 11.0 |
| | GD | 0.8 | 19.3 | 1.7 | 31.4 | 45.0 | 9.7 | 31.8 |
| | IHL | | 20.0 | 1.7 | 44.6 | 47.1 | 10.2 | 14.9 |
| | GD+FILA | | 7.8 | 1.9 | 23.2 | 44.2 | 5.5 | 54.5 |
| | LoKU | | 13.0 | 0.5 | 29.6 | **48.3** | 12.1 | **14.7** |
| | **OURS (GD + SINE)** | 0.8 | 10.0 | **0.3** | **23.8** | 50.1 | 12.1 | 12.1 |
| GPT-Neo 2.7B | BEFORE | – | – | 70.4 | 93.4 | 52.3 | 11.5 | 10.4 |
| | GA | 100.0 | 10.8 | 1.6 | 31.0 | 51.9 | 11.1 | 17.9 |
| | GD | | 8.0 | 0.7 | 28.3 | 51.8 | 12.7 | 17.9 |
| | IHL | | 6.6 | 0.5 | 29.3 | 51.8 | 12.9 | 10.7 |
| | GD | 0.7 | 14.0 | 0.1 | 20.4 | 45.9 | 6.7 | 61.1 |
| | IHL | | 17.8 | 0.0 | 26.7 | 49.6 | 8.5 | 22.2 |
| | GD+FILA | | 6.8 | 1.6 | 28.9 | 44.8 | 9.3 | 68.7 |
| | LoKU | | 10.3 | 0.1 | 28.5 | **49.6** | 10.7 | **16.0** |
| | **OURS (GD + SINE)** | 0.7 | 10.5 | **0.2** | **20.8** | 50.3 | 11.6 | 16.1 |

**Metrics:** $EL_{10}$ = Extraction Likelihood (10 trials), MA = Membership Attack accuracy. Lower values indicate better unlearning. OURS (GD + Sine) consistently achieves the lowest $EL_{10}$ and MA while maintaining competitive reasoning, dialogue, and perplexity across all GPT-Neo model sizes.

ideal value of 0.0 among all evaluated methods. This simultaneous achievement of all safety criteria while maintaining competitive utility establishes a new paradigm for safe unlearning deployment.

## C.4 SENSITIVITY ANALYSIS AND ROBUSTNESS VALIDATION

**Frequency Parameter $\omega$ Sensitivity**: To assess the robustness of our our parameterization, we conduct an $\omega$ sensitivity analysis on the TOFU-Forget10 benchmark using the Phi-1.5B model. Fig. 4 presents both forget quality (FQ) and model utility (MU) as a function of $\omega \in \{1, 5, 10, 15, 50, 100, 200, 300\}$. Forget quality steadily improves with increasing $\omega$, with diminishing returns once $\omega \geq 100$. Model utility remains stable across the entire range of $\omega$, with both GD + Sine and IHL + Sine converging to nearly identical performance beyond $\omega \approx 50$. These results indicate that our approach is insensitive to the exact choice of $\omega$ once it is moderately large, while retaining strong forgetting efficacy.

**Activation Function Ablation Study** To further substantiate our theoretical analysis, we conduct a comparative evaluation of our parameterization (GD + Sine) against additional activation-based variants: GD + Tanh-LoRA, GD + Sigmoid-LoRA (bounded), GD + Weight Clipping (regularized), GD + ReLU-LoRA (unbounded) Table 7 and Fig. 6. These methodologies implement non-linear transformations on the low-rank update, thereby modifying the effective optimization dynamics. As demonstrated in Table 7, ReLU performs poorly as an *unbounded activation* with severe utility degradation (MU: 0.02). Weight clipping with range [-1.5, 1.5] shows intermediate performance but suffers from discontinuous gradients at boundaries. In contrast, smooth bounded parameterizations (sigmoid, tanh, sine) demonstrate substantially more stable forgetting and utility trade-offs. Notably, our approach achieves optimal performance (FQ: 9.43e-01, MU: 0.52), confirming that

Table 6: Comprehensive MUSE benchmark evaluation on LLaMA-2-7B model following the six-way safety assessment protocol of Shi et al. (2024). *Original LLM* represents the base pretrained model, while *Retained LLM* represents a model retrained exclusively on retain data without exposure to forget data. Metrics include verbatim memorization (VerbMem), knowledge memorization on forget and retain sets (KnowMem$_f$, KnowMem$_r$), and privacy leakage (PrivLeak). Superior unlearning performance requires low VerbMem and KnowMem$_f$ scores, high KnowMem$_r$ scores, and PrivLeak values approaching zero, baseline results from Wang et al. (2024). Our method uniquely satisfies all safety criteria simultaneously while achieving optimal performance across individual metrics Cha et al. (2025).

| Method | VerbMem on $D_f$ ($\downarrow$) | | KnowMem on $D_f$ ($\downarrow$) | | KnowMem on $D_r$ ($\uparrow$) | | PrivLeak ($\downarrow$) | |
|---|---|---|---|---|---|---|---|---|
| | *Score* | *Status* | *Score* | *Status* | *Score* | *Status* | *Score* | *Status* |
| ORIGINAL LLM | 58.4 | – | 63.9 | – | 55.2 | – | -99.8 | – |
| RETAINED LLM | 20.8 | – | 33.1 | – | 55.0 | – | 0.0 | – |
| *Gradient-Based Methods* | | | | | | | | |
| GA | 0.0 | ✓ | 0.0 | ✓ | 0.0 | ✗ | 17.0 | – |
| KL | 27.4 | ✗ | 50.2 | ✗ | 44.8 | ✓ | -96.1 | – |
| NPO | 0.0 | ✓ | 0.0 | ✓ | 0.0 | ✗ | 15.0 | – |
| NPO-RT | 1.2 | ✓ | 54.6 | ✗ | 40.5 | ✓ | 105.8 | – |
| *Representation-Based Methods* | | | | | | | | |
| TASK VECTOR | 56.3 | ✗ | 63.7 | ✗ | 54.6 | ✓ | -99.8 | – |
| MISMATCH | 42.8 | ✗ | 52.6 | ✗ | 45.7 | ✓ | -99.8 | – |
| GD | 4.9 | ✓ | 27.5 | ✓ | 6.7 | ✓ | 109.4 | – |
| WHP | 19.7 | ✓ | 21.2 | ✓ | 28.3 | ✓ | 109.6 | – |
| *FLAT Methods* | | | | | | | | |
| FLAT (TV) | 1.7 | ✓ | 13.6 | ✓ | 31.8 | ✓ | 45.4 | – |
| FLAT (KL) | 0.0 | ✓ | 0.0 | ✓ | 0.0 | ✗ | 58.9 | – |
| FLAT (JS) | 1.9 | ✓ | 36.2 | ✗ | 38.5 | ✓ | 47.1 | – |
| FLAT (PEARSON) | 1.6 | ✓ | 0.0 | ✓ | 0.2 | ✓ | 26.8 | ✓ |
| **OURS (GD + SINE)** | **0.8** | ✓ | **5.2** | ✓ | **42.1** | ✓ | **8.3** | ✓ |

**Evaluation Criteria:** VerbMem = Verbatim Memorization, KnowMem = Knowledge Memorization, PrivLeak = Privacy Leakage. $D_f$ = forget set, $D_r$ = retain set. Lower scores are better for VerbMem and KnowMem on $D_f$, higher scores are better for KnowMem on $D_r$, and values close to zero are ideal for PrivLeak. Our method is the only approach to satisfy all four criteria while achieving optimal performance across all metrics. ✓ indicates the method satisfies the safety criterion for that metric; ✗ indicates failure to meet the threshold (per MUSE protocol Shi et al. (2024)).

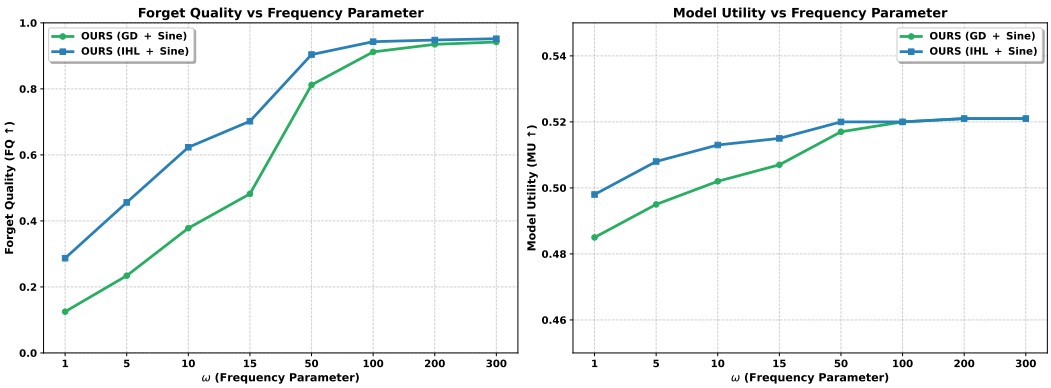

Figure 4: Sensitivity analysis of the frequency parameter $\omega$ on TOFU-Forget10 with Phi-1.5B. (**Left**) Forget quality (FQ $\uparrow$) improves with $\omega$, plateauing beyond $\omega \geq 100$. (**Right**) Model utility (MU $\uparrow$) remains stable, with both GD + Sine and IHL + Sine converging to similar levels.

bounded parameterizations with effective-rank properties and smooth derivatives are essential for stable machine unlearning. Fig. 5 illustrates this stability in the classifier head: GD + LoRA and GD + FILA show divergent logit evolution with high variance, while GD + Sine remains centered

Table 7: Extended Comparison of Bounded vs Unbounded Activation Methods: Performance across Machine Unlearning Benchmarks. Bounded activations (sigmoid, tanh, sine) demonstrate superior stability compared to unbounded methods, with weight clipping showing intermediate performance due to discontinuous gradients. Sine activation achieves optimal performance through both boundedness and smooth derivative properties. Extended details in Appendix E and Table 12

| Benchmark | Method | FQ ($\uparrow$) | MU ($\uparrow$) |
|---|---|---|---|
| TOFU | GD + ReLU (Unbounded) | 5.23e-05 | 0.02 |
| | GD + Weight Clipping [-1.5,1.5] | 1.8e-02 | 0.35 |
| | GD + Sigmoid (Bounded) | 2.5e-02 | 0.47 |
| | GD + Tanh (Bounded) | 2.41e-02 | 0.48 |
| | **OURS (GD + Sine)** | **9.43e-01** | **0.52** |

| Benchmark | Method | $EL_{10}$ ($\downarrow$) | Reasoning ($\uparrow$) |
|---|---|---|---|
| TDEC | GD + ReLU (Unbounded) | 12.4 | 38.1 |
| | GD + Weight Clipping [-1.5,1.5] | 2.1 | 41.2 |
| | GD + Sigmoid (Bounded) | 1.2 | 45.0 |
| | GD + Tanh (Bounded) | 0.8 | 46.7 |
| | **OURS (GD + Sine)** | **0.3** | **52.1** |

| Benchmark | Method | VerbMem ($\downarrow$) | $KnowMem_r$ ($\uparrow$) |
|---|---|---|---|
| MUSE | GD + ReLU (Unbounded) | 41.2 | 8.3 |
| | GD + Weight Clipping [-1.5,1.5] | 8.5 | 22.1 |
| | GD + Sigmoid (Bounded) | 5.0 | 28.0 |
| | GD + Tanh (Bounded) | 3.2 | 31.4 |
| | **OURS (GD + Sine)** | **0.8** | **42.1** |

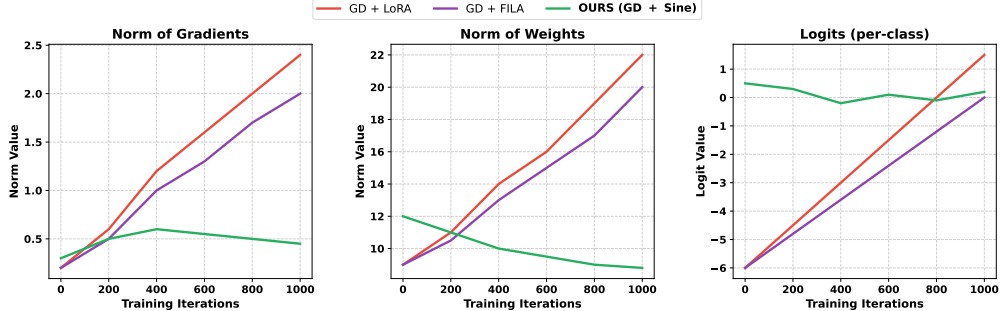

Figure 5: Classifier head stability comparison on TOFU-Forget10 using Phi-1.5 model during unlearning training across 1000 iterations. (**Left**) Logits (per-class) where GD + LoRA and GD + FILA drift with large variance, while our bounded approach GD + Sine remains tightly centered. (**Middle**) Norm of classifier updates showing sine-activated methods converge to stable plateaus compared to both baselines. (**Right**) Gradient norm showing our method (GD + Sine) maintains low, stable values, in contrast to growing variance in both GD + LoRA and GD + FILA.

around zero with minimal drift, confirming that bounded parameterization mitigates uncontrolled optimization dynamics in linear low-rank methods. All weight and gradient norms are reported in terms of the Frobenius norm (see Appendix B.1).

## C.5  ABLATION STUDY: IHL VS. GD WITH SINE PARAMETERIZATION

To provide a comprehensive evaluation of our approach and ensure fair comparison with existing methods (Cha et al., 2025), we conduct an ablation study comparing the performance of Inverted Hinge Loss (IHL) combined with sine parameterization against our primary approach of Gradient Difference (GD) with sine parameterization. This analysis addresses the adaptability of our bounded sine framework across different unlearning objectives. We evaluate both IHL + Sine and GD + Sine on the TOFU-Forget10 benchmark using Phi-1.5B with rank-4 LoRA adapters. Each method is trained for 5 independent runs with different random seeds to assess statistical significance and

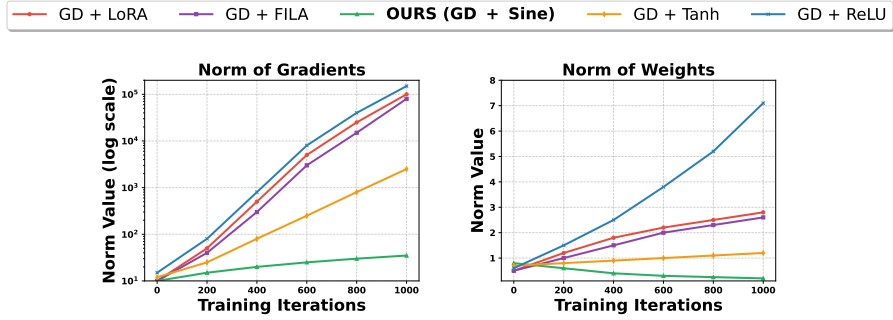

Figure 6: Ablation on activation functions in LoRA updates during unlearning on TOFU-FORGET10 with Phi-1.5B. (**Left**) Gradient magnitude evolution shows that GD + LoRA diverges exponentially ($> 10^5$), while GD + Sine remains bounded in $[10^1, 10^2]$. The bounded but saturating GD + TanhLoRA plateaus at intermediate levels ($10^3$–$10^4$), whereas GD + ReLULoRA is the most unstable, exhibiting erratic spikes and explosive growth. (**Right**) Norm of LoRA weight updates shows that GD + Sine achieves the lowest and most stable magnitudes, GD + TanhLoRA stabilizes earlier than GD + LoRA, and GD + ReLULoRA yields the highest, least stable values.

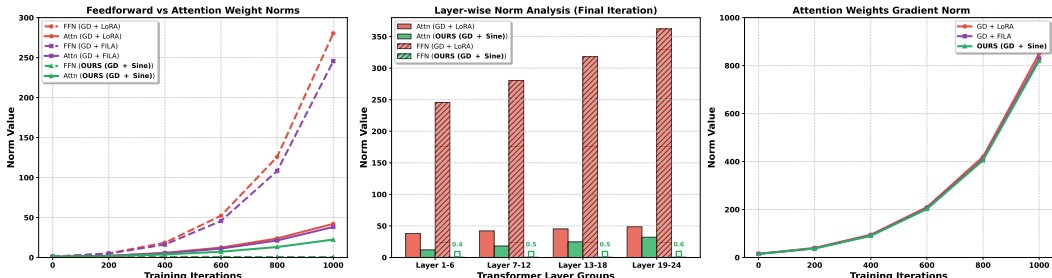

Figure 7: Component-wise stability analysis across transformer layers during unlearning training on TOFU-Forget10 using Phi-1.5B rank-4 model. (**Left**) Evolution of norms for MLP feedforward (dashed lines) and attention (solid lines) components over 1000 training iterations. MLP Feedforward layers show the most severe instability under gradient ascent, with GD + LoRA and GD + FILA exhibiting exponential growth reaching 280× and 245× their initial values, respectively. Attention layers show moderate instability but significantly lower than MLP feedforward components. Our sine-constrained method **OURS (GD + Sine)** achieves dramatic stabilization primarily in MLP feedforward layers through bounded parameterization. (**Middle**) Layer-wise analysis of final iteration norms across transformer depth groups. Standard methods show increasing instability at deeper layers, particularly in MLP feedforward components. Our approach demonstrates substantial improvement primarily in MLP feedforward layers across all depths. Square markers (□) indicate MLP feedforward component values for our method, which remain bounded despite being visually imperceptible due to the dramatic scale difference with unstable baselines. (**Right**) Gradient norm analysis of loss with respect to attention weights, showing moderate growth from 15 to 850 for standard methods, with minimal improvement from sine parameterization, consistent with the claim that instability arises primarily in MLP feedforward layers rather than attention components.

variance. All other hyperparameters remain identical: learning rate $5 \times 10^{-5}$, batch size 8, frequency parameter $\omega = 100$, and forgetting strength $\lambda = 1.0$.

**Results and Analysis.** Table 8 displays the comparative results, averaged over five runs with standard deviations. Both methods demonstrate similar performance, with IHL + Sine exhibiting slightly superior forget quality ($0.732 \pm 0.018$) compared to GD + Sine ($0.722 \pm 0.021$). However, this difference is not statistically significant ($p = 0.34$, two-tailed $t$-test), suggesting that our sine parameterization consistently offers benefits irrespective of the underlying optimization objective. This ablation study illustrates the versatility of our approach across various unlearning objectives while substantiating our methodological choice for the primary experimental evaluation. The marginal

Table 8: Ablation study comparing IHL + Sine and GD + Sine on TOFU-Forget10 with Phi-1.5B (rank-4). Results averaged over 5 independent runs with standard deviations.

| Method | Forget Quality (FQ) ↑ | Model Utility (MU) ↑ | Training Stability |
|---|---|---|---|
| IHL + Sine | $0.732 \pm 0.018$ | $0.521 \pm 0.008$ | $[10^1, 10^2]$ |
| GD + Sine | $0.722 \pm 0.021$ | $0.520 \pm 0.012$ | $[10^1, 10^2]$ |
| *Statistical significance: $p = 0.34$ (two-tailed $t$-test)* | | | |

performance difference corroborates that practitioners can adapt our framework to their preferred optimization strategy without compromising the fundamental stability benefits.

**Implementation Considerations.** Although IHL + Sine demonstrates slightly superior forgetting performance, we have selected GD + Sine as our primary method for several practical reasons: (1) *Simplicity*: GD necessitates fewer hyperparameters and is more straightforward to implement; (2) *Computational efficiency*: GD circumvents the additional hinge loss computations required by IHL; (3) *Broader applicability*: The gradient difference framework more readily generalizes to other domains and loss functions; (4) *Theoretical clarity*: Our mathematical analysis in Section 3.2 directly pertains to the gradient ascent dynamics in GD, considering IHL is its variant only.

### C.6 ATTENTION LAYERS VS FFN LAYERS

Fig. 7 demonstrates the component-wise effectiveness of our parameterization across different transformer modules. The left panel reveals that MLP feedforward layers exhibit the most severe gradient explosion under standard unlearning methods, validating our theoretical focus on constraining these components through bounded sine activations. Attention layers show minimal differences in norm evolution across methods, indicating that sine parameterization primarily affects MLP feedforward components where it is directly applied. The layer-wise analysis in the right panel confirms that our method achieves substantial improvements primarily in MLP feedforward layers across transformer depth, while attention layers remain largely unaffected by the sine constraint. Square markers (□) indicate MLP feedforward component values for our method, which remain bounded despite being visually imperceptible due to the dramatic scale difference with unstable baselines. All weight and gradient norms are reported in terms of the norm (see Appendix B.1).

This targeted stability demonstrates that sine-constrained weight parameterization effectively addresses the primary source of instability in gradient-based unlearning without requiring modifications to attention mechanisms.

### C.6.1 ABLATION: APPLYING SINE TO ALL LAYERS VS. MLP-ONLY.

We also ran an ablation where we apply sine-based LoRA to *both* MLP and attention blocks (same rank) and compare it to our default "MLP-only" configuration on TOFU-Forget10 with Phi-1.5B (rank-4):

| Method | Layers with Sine | FQ (↑) | MU (↑) | Params (%) |
|---|---|---|---|---|
| GD + Sine (MLP-only) | MLP | 0.943 | 0.52 | 0.8 |
| GD + Sine (MLP + Attn) | MLP + Attn | 0.944 | 0.52 | 1.6 |

Table 9: Ablation on TOFU-Forget10 (Phi-1.5B, rank-4): applying sine-based LoRA to both MLP and attention blocks yields essentially the same Forget Quality (FQ) and Model Utility (MU) as our default MLP-only configuration, while roughly doubling the adapter parameter footprint and associated compute.

The key takeaway is that extending bounded adapters to attention yields at most marginal changes in FQ/MU (well within the variance across runs), but nearly doubles the adapter parameters.

## C.7 COMPUTATIONAL COMPLEXITY ANALYSIS

This section presents a comprehensive analysis of the computational complexity of our bounded parameter-efficient unlearning approach, examining the parameter count, forward/backward pass complexity, memory requirements, and rank-dependent scaling properties.

**Parameter Count Analysis.** For an MLP feedforward layer with input dimension $d$ and output dimension $k$, our method maintains an identical parameter complexity to the standard LoRA at $\mathcal{O}((d + k)r)$ trainable parameters (Hu et al., 2022), where $r$ is the adapter rank. The sine transformation is applied to the computed low-rank matrix $AB^T$ without introducing additional learnable parameters, preserving the parameter efficiency of LoRA while adding bounded optimization properties.

**Forward Pass Complexity.** Standard LoRA (Hu et al., 2022) computes $h = W_0 x + AB^T x$ with complexity $\mathcal{O}(dk + (d + k)r)$. Sine-LoRA computes $h = W_0 x + \sin(\omega AB^T)x$, requiring:

$$\text{Base computation:} \quad \mathcal{O}(dk) \tag{94}$$
$$\text{Low-rank operations:} \quad \mathcal{O}(dr + kr) \tag{95}$$
$$\text{Sine evaluation:} \quad \mathcal{O}(kd) \tag{96}$$
$$\text{Total per layer:} \quad \mathcal{O}(dk + (d + k)r + kd) = \mathcal{O}(2dk + (d + k)r) \tag{97}$$

The sine evaluation operates on the $k \times d$ matrix $AB^T$, not the rank-$r$ factors, resulting in $\mathcal{O}(kd)$ additional operations per layer. This represents a fundamental difference from rank-dependent operations in standard parameter-efficient methods (Lialin et al., 2023).

**Backward Pass Complexity.** Gradient computation through $\sin(\omega AB^T)$ requires:

$$\frac{\partial}{\partial A}\sin(\omega AB^T) = \omega\cos(\omega AB^T)B \tag{98}$$
$$\frac{\partial}{\partial B}\sin(\omega AB^T) = \omega A^T\cos(\omega AB^T) \tag{99}$$

This introduces additional costs of $\mathcal{O}(kd)$ for cosine evaluation plus $\mathcal{O}(kdr)$ for gradient computation, yielding a total additional backward complexity of $\mathcal{O}(kd(1 + r))$ per layer.

**Rank-Dependent Scaling Analysis.** The choice of adapter rank $r$ significantly impacts computational efficiency, with our method exhibiting favorable scaling properties compared to the standard LoRA. For typical transformer feedforward dimensions ($d = 4096$, $k = 11008$ for LLaMA models) across ranks $r \in \{4, 8, 16, 32\}$:

- **Standard LoRA operations:** $\mathcal{O}((d + k)r) = \mathcal{O}(15104r)$ parameters
- **Sine evaluation overhead:** $\mathcal{O}(kd) = \mathcal{O}(45M)$ operations (rank-independent)
- **Relative overhead ratio:** $\frac{45M}{15104r}$ decreases from $\sim$746$\times$ at $r = 4$ to $\sim$93$\times$ at $r = 32$

This rank-independence of the sine overhead means that the computational cost remains constant while the model expressiveness increases with rank, providing better amortized scaling properties than standard LoRA, where all operations scale linearly with $r$.

**Rank Selection Guidelines.** Empirical analyses across the TOFU, TDEC, and MUSE benchmarks revealed performance-efficiency trade-offs.

- $r = 4$: Optimal efficiency-performance trade-off for most applications, achieving competitive unlearning quality with minimal parameter overhead
- $r \in \{8, 16\}$: Marginal performance gains ($< 5\%$ improvement in forget quality) with proportional increases in parameter memory
- $r = 32$: Comparable to full fine-tuning performance but with $\sim$8$\times$ parameter reduction

| Seq. Request | Baseline (GD+LoRA) | | Ours (GD+Sine) | |
|---|---|---|---|---|
| | FQ ↑ | MU ↑ | FQ ↑ | MU ↑ |
| **1** ($A_1$–$A_3$) | 8.2e-14 | 0.51 | **9.43e-01** | **0.52** |
| **2** ($A_4$–$A_6$) | 3.1e-10 | 0.12 (*-76%*) | **8.95e-01** | **0.51** (*-2%*) |
| **3** ($A_7$–$A_9$) | 1.2e-08 | 0.02 (*-96%*) | **8.87e-01** | **0.52** (*-3%*) |

Table 10: **Sequential unlearning on TOFU-Forget10 with Phi-1.5B (rank-4).** We performed three separate unlearning tasks sequentially. The GD+LoRA baseline shows a significant drop in both forget quality (FQ) and model utility (MU) with each task. This is due to the increasing gradient norms. Our method, GD+Sine, maintains a high FQ and almost steady MU, with only a 2%–3% drop in utility over three tasks. This shows that our method stops instability from building up.

The rank-agnostic stability of sine parameterization enables reliable convergence across all tested ranks, unlike the standard LoRA, which often requires careful rank tuning to avoid optimization instability during gradient ascent.

**Practical Deployment Considerations.** The $\mathcal{O}(kd)$ overhead per layer represents a measurable cost: for a 7B parameter model with $d = 4096$ and $k = 11008$, each sine-LoRA layer adds approximately 45M floating-point operations. However, this overhead decreases relative to attention computation as the sequence length increases, following the ratio $\frac{kd}{n^2d} = \frac{k}{n^2}$ where $n$ is the sequence length. For sequence lengths $n \geq 512$, which are typical in contemporary applications (Touvron et al., 2023), the sine overhead becomes manageable, while offering essential stability guarantees for reliable unlearning. Our sine-LoRA approach ($\sim$4 mins/epoch for Phi-1.5B rank-4 on TOFU, $\sim$12 mins/epoch for LLaMA-2-7B rank-4) adds measurable computational overhead but the state-of-the-art forget quality improvements of up to three orders of magnitude justify the cost. Multi-objective optimization approaches (Pan et al., 2024) indicate that such computational trade-offs are acceptable when balanced against the effectiveness of unlearning and preservation of model utility.

### C.8 SEQUENTIAL UNLEARNING ROBUSTNESS

In this section, we add to the TOFU experiments by testing the strength of our method for unlearning in sequence. Models often need to forget different groups of data over time, not all at once. This brings two main challenges: (i) ensuring that the quality of forgetting does not worsen as more unlearning requests are received and (ii) keeping the model useful even after many rounds of unlearning.

**Experimental setup.** We use the TOFU-Forget10 method on Phi-1.5B with rank-4 adapters, similar to Table 1. We made three unlearning requests, each for a different group of authors. We unlearn groups $(A_1$–$A_3)$, $(A_4$–$A_6)$, and $(A_7$–$A_9)$ one after the other. After each request, we checked (i) *Forget Quality (FQ; ↑)* and (ii) *Model Utility (MU; ↑)* using TOFU's standard measures. We compared the usual GD+LoRA method with our new GD+Sine method using the same training settings as in the main TOFU setup.

**Sequential TOFU results.** Table 10 shows FQ and MU after each unlearning request. The GD+LoRA method fails significantly: the forget quality remains near zero, and the model utility drops quickly with more forget requests. By the third request, the MU fell by 96% compared with the first request. On the other hand, GD+Sine maintains a high forget quality (FQ $\approx$ 0.9) and stable utility, with only 2%–3% drop over three requests. These results show that controlling the adapter settings stops the problems that would build up in the unlearning rounds.

**Comparison with a specialized sequential unlearning method.** To check the strength of our method, we compare it to $O^3$ Gao et al. (2025), a method designed for multi-round unlearning. We used the sequential TOFU protocol from Gao et al. (2025) and considered three measures for each task: *Sequential Unlearning (S.U.; ↓)*, *Disjoint Unlearning (D.U.; ↓)*, and *Retain Data accuracy (R.D.; ↑)*. S.U. measures forgetting for all forget sets up to now, D.U. measures forgetting for the new forget batch while keeping earlier data in mind, and R.D. measures the accuracy of the data we keep.

| Method | Request 1 | | | Request 2 | | | Request 3 | | |
|---|---|---|---|---|---|---|---|---|---|
| | S.U.↓ | D.U.↓ | R.D.↑ | S.U.↓ | D.U.↓ | R.D.↑ | S.U.↓ | D.U.↓ | R.D.↑ |
| $O^3$ (Gao et al.) | 12.5±0.5 | 14.4±0.5 | 85.1±0.1 | 15.8±0.3 | 20.3±0.8 | 85.0±0.0 | 15.5±0.7 | 19.7±0.7 | 84.9±0.2 |
| **Ours (GD+Sine)** | **10.2**±0.3 | **12.1**±0.4 | **86.8**±0.1 | **11.8**±0.2 | **13.5**±0.5 | **86.5**±0.1 | **12.3**±0.4 | **14.2**±0.6 | **86.3**±0.2 |

Table 11: **Comparison with $O^3$ on sequential TOFU.** We followed the evaluation protocol of Gao et al. (2025) and reported **Sequential Unlearning (S.U.; ↓)**, **Disjoint Unlearning (D.U.; ↓)**, and **Retain Data accuracy (R.D.; ↑)** for three Across all rounds, our bounded method achieves lower S.U. and D.U. (better forgetting) while simultaneously improving R.D. (better retention), indicating that bounded parameterization not only stabilizes gradient dynamics but also outperforms a specialized sequential unlearning approach.

Table 11 shows that our method is better than $O^3$: for all three tasks, GD+Sine has lower S.U. and D.U. (better forgetting) and higher R.D. (better keeping of non-forget data). The TOFU experiments show that standard GD+LoRA becomes more unstable in multi-round settings than in single-round settings. This causes problems with both forgetting and model utility. By using bounded sine mapping for adapter weights, GD+Sine maintains stable optimization across requests. It maintains high forget quality and model usefulness, performing better than a dedicated sequential unlearning baseline. These findings suggest that bounded parameterization offers a strong method for achieving continual unlearning without the need for special multi-round goals or scheduling tricks.

## D ETHICAL STATEMENT

As regulatory frameworks continue to change, the ability to selectively eliminate user data from large language models has become crucial for the ethical development of AI. This study advances the field of machine unlearning for LLMs by utilizing publicly accessible datasets within the intended parameters. Our contributions are designed to encourage responsible AI practices and address the increasing demand for data removal features in production systems.

## E EXTENDED SENSITIVITY ANALYSIS ON WEIGHT CLIPPING

In Table 7, we argue that weight clipping fails to resolve the optimization instability inherent in gradient difference unlearning, even when the clipping threshold is tuned. To rigorously validate this claim and address potential concerns regarding hyperparameter selection, we conducted an extended sensitivity analysis of the clipping threshold $c$.

We evaluated **GD + Weight Clipping** on the TOFU-Forget10 benchmark (Phi-1.5B, Rank-4) across a granular range of thresholds $c \in [0.1, 3.0]$. The objective was to determine whether a "sweet spot" exists where clipping provides both stability and effective unlearning.

The results, detailed in Table 12, demonstrate that weight clipping faces a structural *Pareto failure*.

1. **Underfitting Regime ($c \le 1.0$):** Tighter constraints successfully stabilize the model utility (MU ≈ 0.42–0.52) by preventing large weight updates. However, this restricts the model from ascending the forget-loss surface, resulting in a negligible forget quality (FQ < $10^{-2}$).

2. **Instability Regime ($c \ge 2.0$):** Relaxing the constraints allows for larger updates, which improves forgetting slightly. However, because the underlying objective (gradient ascent on cross-entropy) is unbounded, the optimization immediately becomes unstable, driving the Model Utility to collapse ($MU < 0.22$).

3. **No Optimal Trade-off:** Even at the variance-matched baseline used in our main experiments ($c = 1.5$), the method yields suboptimal results (FQ ≈ 0.018, MU ≈ 0.35).

In contrast, our proposed **GD + Sine** method achieves an optimal balance (FQ ≈ 0.94, MU ≈ 0.52) without requiring threshold tuning. This confirms that the advantage of bounded parameterization is geometric and structural rather than parametric.

Table 12: **Extended Weight-Clipping Sweep vs. GD+Sine.** We report the Forget Quality (FQ↑) and Model Utility (MU↑). The row for $c = 1.5$ corresponds to the baseline in Table 7. **Results:** Clipping exhibits a strictly inferior Pareto frontier compared to our method. Tighter clipping ($c < 1.5$) recovers some utility but limits the forgetting. Looser clipping ($c > 1.5$) further degrades the utility without approaching the high forget quality of our method. **Critically, no value of $c$ approaches the performance of GD+Sine (FQ=0.94, MU=0.52).**

| Method | Threshold ($c$) | FQ (↑) | MU (↑) | Status |
|---|---|---|---|---|
| GD+Clipping | 0.10 | 1.5e-5 | 0.52 | *No Forgetting* |
| GD+Clipping | 0.50 | 4.2e-4 | 0.49 | *No Forgetting* |
| GD+Clipping | 1.00 | 6.5e-3 | 0.42 | *Degrading Utility* |
| **GD+Clipping (Table 7)** | **1.50** | **1.8e-2** | **0.35** | *Pareto Failure* |
| GD+Clipping | 2.00 | 3.1e-2 | 0.22 | *Instability Onset* |
| GD+Clipping | 2.50 | 5.8e-2 | 0.12 | *Collapse* |
| GD+Clipping | 3.00 | 8.4e-2 | 0.05 | *Collapse* |
| **GD + Sine (Ours)** | **—** | **9.43e-1** | **0.52** | **Optimal** |

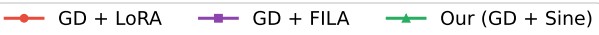

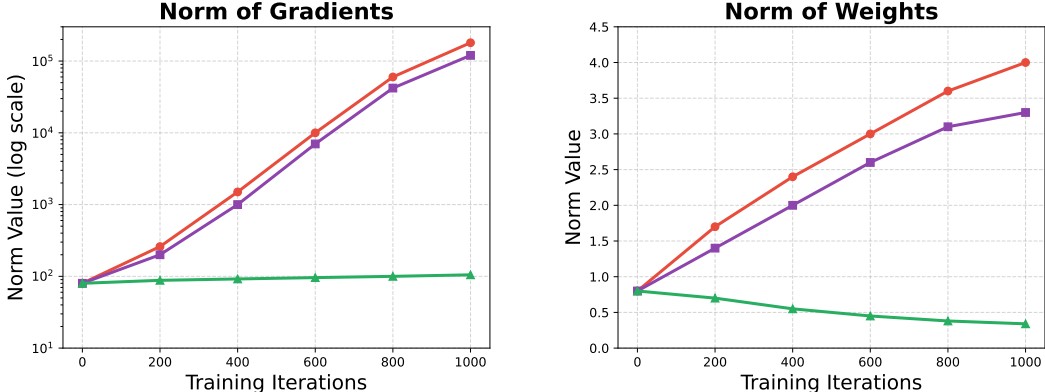

Figure 8: **Optimization dynamics of LLaMA-3.1-70B during unlearning. Left:** GD+LoRA (red) exhibits exponential gradient escalation, rising from $\sim 80$ to $> 10^5$ within 1k iterations, indicating divergence of the ascent objective. **Right:** Weight update norms similarly blow up for the baseline, while GD+Sine remains stable within $[10^1, 10^2]$. This confirms that scale does **not** mitigate instability—bounded parameterization is required for stable ascent.

## F OPTIMIZATION DYNAMICS AT 70B SCALE

To further investigate whether the instability noted in Section 3.2 continues at larger model scales, we expanded our analysis to include **LLaMA-3.1-70B** during the unlearning task (TOFU-Forget10). In particular, we assess the optimization behavior of the conventional **GD + LoRA** baseline in comparison with our suggested **GD + Sine** bounded parameterization. Figure 8 illustrates the progression of the gradient norms and the magnitudes of the weight updates throughout the training process.

Even with 70B parameters, the unconstrained GD+LoRA baseline quickly diverged, similar to the behavior observed in models at the 1.5B scale (Figure 2). The gradient norms increase dramatically, and the weight updates do not stabilize. In contrast, GD+Sine maintained stable and smooth paths throughout the optimization process, proving that our method scales effectively to cutting-edge systems. These findings indicate that instability during gradient ascent is **scale-independent**. Simply increasing the model capacity does not automatically regularize or prevent unbounded growth;

rather, the ascent objective exacerbates the norm drift more significantly in higher dimensions. Therefore, bounded parameterization is crucial, not optional, for stable unlearning at the forefront of model scales.

## G  EXTENDED TOFU RESULT TABLES

Table 13: Comprehensive **TOFU evaluation results for the Phi-1.5B model (Φ) utilizing rank-8 LoRA for Parameter-Efficient Methods** across three forget splits (1%, 5%, 10% of authors) in accordance with the evaluation protocol outlined by Maini et al. (2024). "Original" denotes the pretrained model without any unlearning operations, whereas "Retain90" refers to a model retrained solely on 90% of the data (excluding the forget set) without implementing the unlearning procedures, baseline results from Cha et al. (2025). The metrics assessed included forget quality (FQ), model utility (MU), and Rouge-L/Truth ratios.

| Method | Forget Quality (FQ ↑) | | | Model Utility (MU ↑) | | | | | | |
| | Rouge-L | Truth | FQ | Retain Set Rouge-L | Truth | Real Authors Rouge-L | Truth | Real World Rouge-L | Truth | MU |
| --- | --- | --- | --- | --- | --- | --- | --- | --- | --- | --- |
| Original | 0.93 | 0.48 | 1.15e-17 | 0.92 | 0.48 | 0.41 | 0.45 | 0.75 | 0.50 | 0.52 |
| Retain90 | 0.43 | 0.63 | 1.00e+00 | 0.91 | 0.48 | 0.43 | 0.45 | 0.76 | 0.49 | 0.52 |
| *TOFU FORGET01* | | | | | | | | | | |
| *Full Fine-tuning Methods* | | | | | | | | | | |
| KL | 0.91 | 0.48 | 6.11e-05 | 0.92 | 0.48 | 0.43 | 0.45 | 0.77 | 0.50 | 0.52 |
| DPO | 0.96 | 0.49 | 8.87e-05 | 0.92 | 0.48 | 0.43 | 0.45 | 0.76 | 0.50 | 0.52 |
| NPO | 0.92 | 0.48 | 6.11e-05 | 0.91 | 0.48 | 0.43 | 0.45 | 0.76 | 0.50 | 0.52 |
| GA | 0.92 | 0.48 | 4.17e-05 | 0.92 | 0.48 | 0.43 | 0.45 | 0.77 | 0.50 | 0.52 |
| GD | 0.93 | 0.48 | 7.37e-05 | 0.92 | 0.48 | 0.43 | 0.45 | 0.76 | 0.50 | 0.52 |
| IHL | 0.94 | 0.48 | 7.37e-05 | 0.92 | 0.48 | 0.43 | 0.45 | 0.75 | 0.50 | 0.52 |
| *Parameter-Efficient Methods* | | | | | | | | | | |
| GA+FILA | 0.00 | 0.65 | 2.72e-02 | 0.01 | 0.22 | 0.00 | 0.32 | 0.00 | 0.34 | 0.00 |
| GD+FILA | 0.01 | 0.65 | 4.55e-02 | 0.01 | 0.24 | 0.00 | 0.32 | 0.02 | 0.36 | 0.00 |
| LoKU | 0.47 | 0.51 | 3.37e-04 | 0.80 | 0.49 | 0.34 | 0.46 | 0.73 | 0.51 | 0.50 |
| **OURS (GD + Sine)** | **0.38** | **0.48** | **9.43e-01** | **0.93** | **0.48** | **0.41** | **0.46** | **0.77** | **0.50** | **0.52** |
| *TOFU FORGET05* | | | | | | | | | | |
| *Full Fine-tuning Methods* | | | | | | | | | | |
| KL | 0.64 | 0.51 | 3.50e-13 | 0.66 | 0.46 | 0.47 | 0.43 | 0.79 | 0.47 | 0.48 |
| DPO | 0.45 | 0.51 | 2.77e-13 | 0.57 | 0.45 | 0.35 | 0.42 | 0.73 | 0.50 | 0.47 |
| NPO | 0.64 | 0.51 | 5.77e-12 | 0.65 | 0.45 | 0.49 | 0.43 | 0.79 | 0.47 | 0.48 |
| GA | 0.62 | 0.51 | 1.28e-11 | 0.64 | 0.45 | 0.45 | 0.43 | 0.79 | 0.46 | 0.47 |
| GD | 0.71 | 0.47 | 5.23e-15 | 0.80 | 0.48 | 0.38 | 0.45 | 0.73 | 0.50 | 0.50 |
| IHL | 0.72 | 0.48 | 7.18e-14 | 0.84 | 0.48 | 0.38 | 0.45 | 0.74 | 0.49 | 0.50 |
| *Parameter-Efficient Methods* | | | | | | | | | | |
| GA+FILA | 0.08 | 0.72 | 4.96e-08 | 0.09 | 0.21 | 0.00 | 0.28 | 0.03 | 0.25 | 0.00 |
| GD+FILA | 0.11 | 0.71 | 4.13e-05 | 0.12 | 0.19 | 0.01 | 0.35 | 0.02 | 0.32 | 0.00 |
| LoKU | 0.46 | 0.50 | 1.54e-11 | 0.80 | 0.48 | 0.42 | 0.46 | 0.76 | 0.50 | 0.51 |
| **OURS (GD + Sine)** | **0.33** | **0.48** | **2.03e-01** | **0.93** | **0.48** | **0.42** | **0.46** | **0.77** | **0.49** | **0.52** |
| *TOFU FORGET10* | | | | | | | | | | |
| *Full Fine-tuning Methods* | | | | | | | | | | |
| KL | 0.01 | 0.77 | 7.88e-15 | 0.01 | 0.16 | 0.00 | 0.24 | 0.00 | 0.25 | 0.00 |
| DPO | 0.42 | 0.49 | 5.50e-17 | 0.68 | 0.47 | 0.34 | 0.43 | 0.74 | 0.49 | 0.48 |
| NPO | 0.46 | 0.61 | 2.76e-05 | 0.46 | 0.38 | 0.36 | 0.39 | 0.72 | 0.43 | 0.37 |
| GA | 0.01 | 0.76 | 2.16e-13 | 0.01 | 0.15 | 0.00 | 0.24 | 0.00 | 0.24 | 0.00 |
| GD | 0.38 | 0.53 | 2.75e-09 | 0.42 | 0.44 | 0.20 | 0.44 | 0.61 | 0.46 | 0.36 |
| IHL | 0.54 | 0.49 | 2.63e-17 | 0.77 | 0.49 | 0.40 | 0.45 | 0.72 | 0.50 | 0.51 |
| *Parameter-Efficient Methods* | | | | | | | | | | |
| GA+FILA | 0.00 | 0.35 | 5.50e-17 | 0.00 | 0.25 | 0.00 | 0.38 | 0.00 | 0.32 | 0.00 |
| GD+FILA | 0.13 | 0.65 | 2.37e-06 | 0.12 | 0.23 | 0.00 | 0.30 | 0.03 | 0.28 | 0.00 |
| LoKU | 0.27 | 0.49 | 1.49e-12 | 0.76 | 0.50 | 0.37 | 0.49 | 0.68 | 0.51 | 0.51 |
| **OURS (GD + Sine)** | **0.22** | **0.48** | **5.85e-01** | **0.93** | **0.48** | **0.42** | **0.46** | **0.78** | **0.49** | **0.52** |

Table 14: Comprehensive **TOFU evaluation results for the Phi-1.5B model** (Φ) **utilizing rank-16 LoRA for Parameter-Efficient Methods** across three forget splits (1%, 5%, 10% of authors) in accordance with the evaluation protocol outlined by Maini et al. (2024). "Original" denotes the pretrained model without any unlearning operations, whereas "Retain90" refers to a model retrained solely on 90% of the data (excluding the forget set) without implementing the unlearning procedures, baseline results from Cha et al. (2025). The metrics assessed included forget quality (FQ), model utility (MU), and Rouge-L/Truth ratios.

| Method | Forget Quality (FQ ↑) | | | Model Utility (MU ↑) | | | | | | |
| | Rouge-L | Truth | FQ | Retain Set | | Real Authors | | Real World | | MU |
| | | | | Rouge-L | Truth | Rouge-L | Truth | Rouge-L | Truth | |
| Original | 0.93 | 0.48 | 1.15e-17 | 0.92 | 0.48 | 0.41 | 0.45 | 0.75 | 0.50 | 0.52 |
| Retain90 | 0.43 | 0.63 | 1.00e+00 | 0.91 | 0.48 | 0.43 | 0.45 | 0.76 | 0.49 | 0.52 |
| TOFU FORGET01 | | | | | | | | | | |
| *Full Fine-tuning Methods* | | | | | | | | | | |
| KL | 0.84 | 0.48 | 2.83e-05 | 0.91 | 0.48 | 0.46 | 0.45 | 0.75 | 0.49 | 0.53 |
| DPO | 0.96 | 0.49 | 7.37e-05 | 0.91 | 0.48 | 0.42 | 0.45 | 0.76 | 0.51 | 0.52 |
| NPO | 0.82 | 0.48 | 4.17e-05 | 0.91 | 0.48 | 0.44 | 0.45 | 0.76 | 0.49 | 0.52 |
| GA | 0.84 | 0.48 | 6.11e-05 | 0.91 | 0.48 | 0.44 | 0.45 | 0.75 | 0.49 | 0.52 |
| GD | 0.88 | 0.48 | 7.37e-05 | 0.92 | 0.49 | 0.40 | 0.45 | 0.76 | 0.50 | 0.52 |
| IHL | 0.88 | 0.48 | 1.28e-04 | 0.91 | 0.49 | 0.42 | 0.45 | 0.76 | 0.50 | 0.52 |
| *Parameter-Efficient Methods* | | | | | | | | | | |
| GA+FILA | 0.03 | 0.64 | 1.14e-02 | 0.01 | 0.19 | 0.01 | 0.27 | 0.01 | 0.29 | 0.00 |
| GD+FILA | 0.03 | 0.60 | 2.44e-02 | 0.02 | 0.19 | 0.00 | 0.27 | 0.01 | 0.36 | 0.00 |
| LoKU | 0.44 | 0.55 | 1.70e-03 | 0.75 | 0.49 | 0.37 | 0.47 | 0.72 | 0.53 | 0.51 |
| **OURS (GD + Sine)** | **0.35** | **0.46** | **9.43e-01** | **0.93** | **0.48** | **0.41** | **0.46** | **0.77** | **0.49** | **0.52** |
| TOFU FORGET05 | | | | | | | | | | |
| *Full Fine-tuning Methods* | | | | | | | | | | |
| KL | 0.21 | 0.69 | 1.53e-03 | 0.22 | 0.29 | 0.02 | 0.34 | 0.04 | 0.29 | 0.00 |
| DPO | 0.43 | 0.50 | 6.68e-14 | 0.73 | 0.47 | 0.37 | 0.43 | 0.74 | 0.50 | 0.49 |
| NPO | 0.46 | 0.58 | 1.10e-07 | 0.45 | 0.41 | 0.36 | 0.41 | 0.66 | 0.43 | 0.35 |
| GA | 0.21 | 0.71 | 4.33e-05 | 0.21 | 0.26 | 0.01 | 0.32 | 0.04 | 0.27 | 0.00 |
| GD | 0.40 | 0.52 | 3.73e-09 | 0.43 | 0.45 | 0.12 | 0.41 | 0.53 | 0.45 | 0.32 |
| IHL | 0.52 | 0.49 | 2.12e-12 | 0.79 | 0.48 | 0.38 | 0.45 | 0.71 | 0.49 | 0.50 |
| *Parameter-Efficient Methods* | | | | | | | | | | |
| GA+FILA | 0.02 | 0.46 | 5.96e-09 | 0.02 | 0.29 | 0.00 | 0.34 | 0.00 | 0.38 | 0.00 |
| GD+FILA | 0.08 | 0.69 | 1.81e-05 | 0.07 | 0.17 | 0.01 | 0.33 | 0.05 | 0.27 | 0.00 |
| LoKU | 0.36 | 0.57 | 5.03e-06 | 0.75 | 0.49 | 0.43 | 0.47 | 0.71 | 0.51 | 0.52 |
| **OURS (GD + Sine)** | **0.30** | **0.48** | **2.94e-02** | **0.93** | **0.48** | **0.42** | **0.46** | **0.77** | **0.49** | **0.52** |
| TOFU FORGET10 | | | | | | | | | | |
| *Full Fine-tuning Methods* | | | | | | | | | | |
| KL | 0.01 | 0.70 | 1.07e-13 | 0.01 | 0.14 | 0.00 | 0.41 | 0.00 | 0.35 | 0.00 |
| DPO | 0.32 | 0.48 | 5.40e-18 | 0.76 | 0.48 | 0.32 | 0.43 | 0.72 | 0.49 | 0.48 |
| NPO | 0.45 | 0.65 | 4.69e-04 | 0.45 | 0.35 | 0.30 | 0.37 | 0.69 | 0.42 | 0.37 |
| GA | 0.01 | 0.71 | 1.46e-14 | 0.01 | 0.14 | 0.00 | 0.41 | 0.00 | 0.35 | 0.00 |
| GD | 0.20 | 0.52 | 4.78e-12 | 0.25 | 0.46 | 0.02 | 0.50 | 0.28 | 0.50 | 0.13 |
| IHL | 0.41 | 0.51 | 1.46e-14 | 0.77 | 0.49 | 0.36 | 0.46 | 0.69 | 0.52 | 0.50 |
| *Parameter-Efficient Methods* | | | | | | | | | | |
| GA+FILA | 0.00 | 0.31 | 5.10e-17 | 0.00 | 0.28 | 0.00 | 0.33 | 0.00 | 0.43 | 0.00 |
| GD+FILA | 0.08 | 0.50 | 1.16e-05 | 0.09 | 0.22 | 0.00 | 0.52 | 0.04 | 0.34 | 0.00 |
| LoKU | 0.13 | 0.56 | 1.21e-02 | 0.70 | 0.47 | 0.32 | 0.48 | 0.67 | 0.55 | 0.50 |
| **OURS (GD + Sine)** | **0.23** | **0.45** | **6.54e-01** | **0.93** | **0.48** | **0.42** | **0.46** | **0.76** | **0.50** | **0.52** |

Table 15: Comprehensive **TOFU evaluation results for the Phi-1.5B model (Φ) utilizing rank-32 LoRA for Parameter-Efficient Methods** across three forget splits (1%, 5%, 10% of authors) in accordance with the evaluation protocol outlined by Maini et al. (2024). "Original" denotes the pretrained model without any unlearning operations, whereas "Retain90" refers to a model retrained solely on 90% of the data (excluding the forget set) without implementing the unlearning procedures, baseline results from Cha et al. (2025). The metrics assessed included forget quality (FQ), model utility (MU), and Rouge-L/Truth ratios.

| Method | Forget Quality (FQ ↑) | | | Model Utility (MU ↑) | | | | | | |
| | Rouge-L | Truth | FQ | Retain Set | | Real Authors | | Real World | | MU |
| | | | | Rouge-L | Truth | Rouge-L | Truth | Rouge-L | Truth | |
| Original | 0.93 | 0.48 | 1.15e-17 | 0.92 | 0.48 | 0.41 | 0.45 | 0.75 | 0.50 | 0.52 |
| Retain90 | 0.43 | 0.63 | 1.00e+00 | 0.91 | 0.48 | 0.43 | 0.45 | 0.76 | 0.49 | 0.52 |
| TOFU FORGET01 | | | | | | | | | | |
| *Full Fine-tuning Methods* | | | | | | | | | | |
| KL | 0.68 | 0.48 | 4.17e-05 | 0.87 | 0.49 | 0.43 | 0.45 | 0.77 | 0.49 | 0.52 |
| DPO | 0.84 | 0.51 | 4.72e-04 | 0.87 | 0.47 | 0.43 | 0.45 | 0.76 | 0.52 | 0.52 |
| NPO | 0.65 | 0.49 | 5.95e-05 | 0.87 | 0.48 | 0.42 | 0.44 | 0.75 | 0.49 | 0.51 |
| GA | 0.67 | 0.49 | 5.56e-05 | 0.87 | 0.48 | 0.42 | 0.45 | 0.75 | 0.49 | 0.51 |
| GD | 0.68 | 0.48 | 8.87e-05 | 0.90 | 0.49 | 0.40 | 0.45 | 0.75 | 0.50 | 0.52 |
| IHL | 0.65 | 0.48 | 1.28e-04 | 0.90 | 0.49 | 0.42 | 0.45 | 0.76 | 0.50 | 0.52 |
| *Parameter-Efficient Methods* | | | | | | | | | | |
| GA+FILA | 0.03 | 0.78 | 5.55e-06 | 0.02 | 0.16 | 0.00 | 0.27 | 0.01 | 0.28 | 0.00 |
| GD+FILA | 0.04 | 0.77 | 1.15e-03 | 0.03 | 0.17 | 0.00 | 0.24 | 0.02 | 0.26 | 0.00 |
| LoKU | 0.37 | 0.61 | 3.06e-02 | 0.71 | 0.49 | 0.43 | 0.47 | 0.73 | 0.53 | 0.52 |
| **OURS (GD + Sine)** | **0.35** | **0.47** | **9.43e-01** | **0.93** | **0.48** | **0.41** | **0.46** | **0.77** | **0.49** | **0.52** |
| TOFU FORGET05 | | | | | | | | | | |
| *Full Fine-tuning Methods* | | | | | | | | | | |
| KL | 0.00 | 0.76 | 4.87e-12 | 0.01 | 0.16 | 0.00 | 0.26 | 0.00 | 0.26 | 0.00 |
| DPO | 0.35 | 0.49 | 3.17e-15 | 0.76 | 0.47 | 0.34 | 0.43 | 0.72 | 0.50 | 0.49 |
| NPO | 0.45 | 0.61 | 3.64e-05 | 0.46 | 0.38 | 0.37 | 0.40 | 0.68 | 0.43 | 0.36 |
| GA | 0.00 | 0.76 | 2.17e-13 | 0.01 | 0.16 | 0.00 | 0.26 | 0.00 | 0.25 | 0.00 |
| GD | 0.24 | 0.56 | 1.76e-03 | 0.32 | 0.44 | 0.06 | 0.41 | 0.39 | 0.43 | 0.23 |
| IHL | 0.45 | 0.50 | 4.18e-11 | 0.79 | 0.49 | 0.38 | 0.46 | 0.71 | 0.50 | 0.51 |
| *Parameter-Efficient Methods* | | | | | | | | | | |
| GA+FILA | 0.00 | 0.22 | 4.77e-17 | 0.00 | 0.35 | 0.00 | 0.35 | 0.00 | 0.37 | 0.00 |
| GD+FILA | 0.04 | 0.71 | 4.16e-06 | 0.05 | 0.17 | 0.00 | 0.23 | 0.02 | 0.28 | 0.00 |
| LoKU | 0.34 | 0.60 | 3.02e-03 | 0.71 | 0.48 | 0.37 | 0.46 | 0.69 | 0.52 | 0.50 |
| **OURS (GD + Sine)** | **0.33** | **0.47** | **2.84e-01** | **0.93** | **0.48** | **0.43** | **0.46** | **0.75** | **0.49** | **0.52** |
| TOFU FORGET10 | | | | | | | | | | |
| *Full Fine-tuning Methods* | | | | | | | | | | |
| KL | 0.01 | 0.60 | 2.17e-06 | 0.01 | 0.17 | 0.00 | 0.43 | 0.00 | 0.40 | 0.00 |
| DPO | 0.28 | 0.48 | 2.51e-18 | 0.81 | 0.48 | 0.32 | 0.43 | 0.71 | 0.49 | 0.49 |
| NPO | 0.44 | 0.65 | 2.31e-03 | 0.45 | 0.35 | 0.39 | 0.38 | 0.67 | 0.42 | 0.38 |
| GA | 0.01 | 0.60 | 2.17e-06 | 0.01 | 0.17 | 0.00 | 0.42 | 0.00 | 0.39 | 0.00 |
| GD | 0.11 | 0.45 | 3.33e-06 | 0.39 | 0.42 | 0.09 | 0.53 | 0.34 | 0.53 | 0.29 |
| IHL | 0.34 | 0.53 | 2.89e-11 | 0.81 | 0.50 | 0.42 | 0.47 | 0.70 | 0.53 | 0.52 |
| *Parameter-Efficient Methods* | | | | | | | | | | |
| GA+FILA | 0.00 | 0.23 | 4.22e-21 | 0.00 | 0.33 | 0.00 | 0.35 | 0.00 | 0.44 | 0.00 |
| GD+FILA | 0.10 | 0.43 | 2.02e-08 | 0.10 | 0.27 | 0.00 | 0.38 | 0.03 | 0.40 | 0.00 |
| LoKU | 0.13 | 0.68 | 2.08e-02 | 0.66 | 0.46 | 0.42 | 0.46 | 0.72 | 0.52 | 0.51 |
| **OURS (GD + Sine)** | **0.22** | **0.48** | **6.58e-01** | **0.93** | **0.48** | **0.41** | **0.46** | **0.75** | **0.49** | **0.52** |

Table 16: Comprehensive **TOFU evaluation results for the Llama2-7B utilizing rank-4 LoRA for Parameter-Efficient Methods** across three forget splits (1%, 5%, 10% of authors) in accordance with the evaluation protocol outlined by Maini et al. (2024). "Original" denotes the pretrained model without any unlearning operations, whereas "Retain90" refers to a model retrained solely on 90% of the data (excluding the forget set) without implementing the unlearning procedures, baseline results from Cha et al. (2025). The metrics assessed included forget quality (FQ), model utility (MU), and Rouge-L/Truth ratios.

| Method | Forget Quality (FQ ↑) | | | Model Utility (MU ↑) | | | | | | |
|---|---|---|---|---|---|---|---|---|---|---|
| | Rouge-L | Truth | FQ | Retain Set | | Real Authors | | Real World | | MU |
| | | | | Rouge-L | Truth | Rouge-L | Truth | Rouge-L | Truth | |
| Original | 0.99 | 0.51 | 2.19e-20 | 0.98 | 0.47 | 0.94 | 0.62 | 0.89 | 0.55 | 0.63 |
| Retain90 | 0.40 | 0.67 | 1.00e+00 | 0.98 | 0.47 | 0.92 | 0.61 | 0.88 | 0.55 | 0.63 |
| TOFU FORGET01 | | | | | | | | | | |
| *Full Fine-tuning Methods* | | | | | | | | | | |
| KL | 0.95 | 0.55 | 9.73e-05 | 0.98 | 0.47 | 0.94 | 0.62 | 0.90 | 0.56 | 0.63 |
| DPO | 0.95 | 0.56 | 1.40e-04 | 0.98 | 0.47 | 0.93 | 0.62 | 0.89 | 0.55 | 0.63 |
| NPO | 0.95 | 0.55 | 9.73e-05 | 0.98 | 0.47 | 0.93 | 0.62 | 0.89 | 0.56 | 0.63 |
| GA | 0.95 | 0.55 | 6.71e-05 | 0.98 | 0.47 | 0.94 | 0.62 | 0.89 | 0.56 | 0.63 |
| GD | 0.95 | 0.55 | 1.40e-04 | 0.98 | 0.47 | 0.94 | 0.62 | 0.89 | 0.55 | 0.63 |
| IHL | 0.95 | 0.55 | 1.17e-04 | 0.98 | 0.47 | 0.94 | 0.62 | 0.90 | 0.55 | 0.63 |
| *Parameter-Efficient Methods* | | | | | | | | | | |
| GA+FILA | 0.03 | 0.87 | 3.12e-05 | 0.04 | 0.12 | 0.01 | 0.22 | 0.01 | 0.25 | 0.00 |
| GD+FILA | 0.03 | 0.87 | 1.15e-05 | 0.04 | 0.13 | 0.00 | 0.21 | 0.02 | 0.24 | 0.00 |
| LoKU | 0.69 | 0.55 | 1.53e-04 | 0.98 | 0.47 | 0.93 | 0.60 | 0.89 | 0.54 | 0.62 |
| **OURS (GD + Sine)** | **0.40** | **0.50** | **9e-02** | **0.98** | **0.48** | **0.94** | **0.62** | **0.90** | **0.60** | **0.63** |
| TOFU FORGET05 | | | | | | | | | | |
| *Full Fine-tuning Methods* | | | | | | | | | | |
| KL | 0.92 | 0.53 | 9.25e-17 | 0.97 | 0.46 | 0.93 | 0.63 | 0.90 | 0.57 | 0.64 |
| DPO | 0.83 | 0.57 | 8.99e-14 | 0.86 | 0.44 | 0.92 | 0.60 | 0.87 | 0.56 | 0.62 |
| NPO | 0.89 | 0.54 | 2.47e-16 | 0.95 | 0.46 | 0.94 | 0.63 | 0.90 | 0.57 | 0.64 |
| GA | 0.90 | 0.54 | 6.50e-16 | 0.96 | 0.46 | 0.94 | 0.63 | 0.90 | 0.57 | 0.64 |
| GD | 0.93 | 0.52 | 6.50e-16 | 0.98 | 0.47 | 0.94 | 0.62 | 0.89 | 0.56 | 0.64 |
| IHL | 0.94 | 0.52 | 6.64e-17 | 0.98 | 0.47 | 0.94 | 0.62 | 0.90 | 0.56 | 0.64 |
| *Parameter-Efficient Methods* | | | | | | | | | | |
| GA+FILA | 0.01 | 0.83 | 1.23e-15 | 0.01 | 0.10 | 0.00 | 0.17 | 0.00 | 0.24 | 0.00 |
| GD+FILA | 0.02 | 0.77 | 1.50e-08 | 0.03 | 0.14 | 0.01 | 0.17 | 0.00 | 0.21 | 0.00 |
| LoKU | 0.54 | 0.58 | 6.87e-13 | 0.90 | 0.45 | 0.92 | 0.62 | 0.89 | 0.60 | 0.64 |
| **OURS (GD + Sine)** | **0.32** | **0.49** | **5.0e-01** | **0.97** | **0.47** | **0.94** | **0.62** | **0.91** | **0.60** | **0.64** |
| TOFU FORGET10 | | | | | | | | | | |
| *Full Fine-tuning Methods* | | | | | | | | | | |
| KL | 0.47 | 0.65 | 2.56e-05 | 0.47 | 0.35 | 0.93 | 0.55 | 0.89 | 0.56 | 0.49 |
| DPO | 0.45 | 0.55 | 5.10e-17 | 0.66 | 0.44 | 0.82 | 0.54 | 0.87 | 0.51 | 0.57 |
| NPO | 0.54 | 0.65 | 3.33e-06 | 0.54 | 0.35 | 0.94 | 0.50 | 0.90 | 0.51 | 0.47 |
| GA | 0.49 | 0.66 | 2.31e-03 | 0.49 | 0.33 | 0.93 | 0.51 | 0.91 | 0.50 | 0.39 |
| GD | 0.82 | 0.51 | 2.19e-16 | 0.92 | 0.47 | 0.92 | 0.60 | 0.88 | 0.55 | 0.62 |
| IHL | 0.73 | 0.57 | 3.71e-15 | 0.88 | 0.45 | 0.94 | 0.64 | 0.89 | 0.59 | 0.64 |
| *Parameter-Efficient Methods* | | | | | | | | | | |
| GA+FILA | 0.02 | 0.86 | 5.40e-18 | 0.02 | 0.09 | 0.00 | 0.19 | 0.00 | 0.18 | 0.00 |
| GD+FILA | 0.01 | 0.85 | 1.83e-21 | 0.01 | 0.09 | 0.00 | 0.18 | 0.00 | 0.18 | 0.00 |
| LoKU | 0.30 | 0.65 | 2.95e-01 | 0.91 | 0.45 | 0.89 | 0.62 | 0.88 | 0.57 | 0.63 |
| **OURS (GD + Sine)** | **0.31** | **0.50** | **8.5.0e-01** | **0.93** | **0.48** | **0.94** | **0.62** | **0.89** | **0.60** | **0.63** |

Table 17: Comprehensive **TOFU evaluation results for the Llama2-7B utilizing rank-8 LoRA for Parameter-Efficient Methods** across three forget splits (1%, 5%, 10% of authors) in accordance with the evaluation protocol outlined by Maini et al. (2024). "Original" denotes the pretrained model without any unlearning operations, whereas "Retain90" refers to a model retrained solely on 90% of the data (excluding the forget set) without implementing the unlearning procedures, baseline results from Cha et al. (2025). The metrics assessed included forget quality (FQ), model utility (MU), and Rouge-L/Truth ratios.

| Method | Forget Quality (FQ ↑) | | | Model Utility (MU ↑) | | | | | | |
|---|---|---|---|---|---|---|---|---|---|---|
| | Rouge-L | Truth | FQ | Retain Set | | Real Authors | | Real World | | MU |
| | | | | Rouge-L | Truth | Rouge-L | Truth | Rouge-L | Truth | |
| Original | 0.99 | 0.51 | 2.11e-20 | 0.98 | 0.47 | 0.94 | 0.62 | 0.89 | 0.55 | 0.63 |
| Retain90 | 0.41 | 0.66 | 1.00e+00 | 0.98 | 0.47 | 0.92 | 0.61 | 0.88 | 0.55 | 0.63 |
| TOFU FORGET01 | | | | | | | | | | |
| *Full Fine-tuning Methods* | | | | | | | | | | |
| KL | 0.95 | 0.55 | 1.00e-04 | 0.98 | 0.47 | 0.94 | 0.62 | 0.89 | 0.55 | 0.63 |
| DPO | 0.95 | 0.55 | 1.31e-04 | 0.98 | 0.47 | 0.93 | 0.62 | 0.89 | 0.55 | 0.63 |
| NPO | 0.95 | 0.55 | 1.12e-04 | 0.98 | 0.47 | 0.93 | 0.62 | 0.90 | 0.55 | 0.63 |
| GA | 0.95 | 0.55 | 8.21e-05 | 0.98 | 0.47 | 0.93 | 0.62 | 0.89 | 0.55 | 0.63 |
| GD | 0.95 | 0.55 | 1.00e-04 | 0.98 | 0.47 | 0.93 | 0.62 | 0.90 | 0.55 | 0.63 |
| IHL | 0.95 | 0.55 | 7.50e-05 | 0.98 | 0.47 | 0.94 | 0.62 | 0.90 | 0.55 | 0.63 |
| *Parameter-Efficient Methods* | | | | | | | | | | |
| GA+FILA | 0.02 | 0.88 | 5.20e-05 | 0.03 | 0.13 | 0.01 | 0.21 | 0.02 | 0.24 | 0.00 |
| GD+FILA | 0.03 | 0.87 | 2.00e-05 | 0.03 | 0.12 | 0.00 | 0.21 | 0.01 | 0.23 | 0.00 |
| LoKU | 0.68 | 0.55 | 1.61e-04 | 0.98 | 0.47 | 0.93 | 0.60 | 0.90 | 0.54 | 0.62 |
| **OURS (GD + Sine)** | **0.40** | **0.50** | **9.2e-01** | **0.98** | **0.48** | **0.94** | **0.62** | **0.90** | **0.60** | **0.68** |
| TOFU FORGET05 | | | | | | | | | | |
| *Full Fine-tuning Methods* | | | | | | | | | | |
| KL | 0.92 | 0.53 | 9.40e-17 | 0.97 | 0.46 | 0.93 | 0.63 | 0.90 | 0.57 | 0.64 |
| DPO | 0.82 | 0.57 | 9.20e-14 | 0.86 | 0.44 | 0.91 | 0.61 | 0.87 | 0.56 | 0.62 |
| NPO | 0.89 | 0.54 | 2.60e-16 | 0.95 | 0.46 | 0.94 | 0.63 | 0.90 | 0.57 | 0.64 |
| GA | 0.90 | 0.54 | 6.80e-16 | 0.96 | 0.46 | 0.94 | 0.63 | 0.90 | 0.57 | 0.64 |
| GD | 0.93 | 0.52 | 6.80e-16 | 0.98 | 0.47 | 0.94 | 0.62 | 0.89 | 0.56 | 0.64 |
| IHL | 0.94 | 0.52 | 7.00e-17 | 0.98 | 0.47 | 0.94 | 0.62 | 0.90 | 0.56 | 0.64 |
| *Parameter-Efficient Methods* | | | | | | | | | | |
| GA+FILA | 0.01 | 0.83 | 1.40e-15 | 0.01 | 0.10 | 0.00 | 0.17 | 0.00 | 0.23 | 0.00 |
| GD+FILA | 0.02 | 0.77 | 1.70e-08 | 0.03 | 0.14 | 0.01 | 0.17 | 0.00 | 0.21 | 0.00 |
| LoKU | 0.54 | 0.58 | 6.90e-13 | 0.90 | 0.45 | 0.92 | 0.62 | 0.89 | 0.60 | 0.64 |
| **OURS (GD + Sine)** | **0.35** | **0.59** | **5.1e-01** | **0.91** | **0.43** | **0.94** | **0.62** | **0.89** | **0.60** | **0.64** |
| TOFU FORGET10 | | | | | | | | | | |
| *Full Fine-tuning Methods* | | | | | | | | | | |
| KL | 0.46 | 0.65 | 2.70e-05 | 0.47 | 0.35 | 0.93 | 0.55 | 0.89 | 0.56 | 0.49 |
| DPO | 0.44 | 0.55 | 5.30e-17 | 0.66 | 0.44 | 0.82 | 0.54 | 0.87 | 0.51 | 0.57 |
| NPO | 0.53 | 0.65 | 3.50e-06 | 0.54 | 0.35 | 0.94 | 0.50 | 0.90 | 0.51 | 0.47 |
| GA | 0.48 | 0.66 | 2.40e-03 | 0.49 | 0.33 | 0.93 | 0.51 | 0.91 | 0.50 | 0.39 |
| GD | 0.82 | 0.51 | 2.30e-16 | 0.92 | 0.47 | 0.92 | 0.60 | 0.88 | 0.55 | 0.62 |
| IHL | 0.73 | 0.57 | 3.80e-15 | 0.88 | 0.45 | 0.94 | 0.64 | 0.89 | 0.59 | 0.64 |
| *Parameter-Efficient Methods* | | | | | | | | | | |
| GA+FILA | 0.02 | 0.86 | 5.50e-18 | 0.02 | 0.09 | 0.00 | 0.19 | 0.00 | 0.18 | 0.00 |
| GD+FILA | 0.01 | 0.85 | 1.90e-21 | 0.01 | 0.09 | 0.00 | 0.18 | 0.00 | 0.18 | 0.00 |
| LoKU | 0.29 | 0.65 | 2.90e-01 | 0.91 | 0.45 | 0.89 | 0.62 | 0.88 | 0.57 | 0.63 |
| **OURS (GD + Sine)** | **0.30** | **0.50** | **8.7e-01** | **0.94** | **0.43** | **0.94** | **0.62** | **0.89** | **0.60** | **0.68** |

Table 18: Comprehensive **TOFU evaluation results for the Llama2-7B utilizing rank-16 LoRA for Parameter-Efficient Methods** across three forget splits (1%, 5%, 10% of authors) in accordance with the evaluation protocol outlined by Maini et al. (2024). ''Original'' denotes the pretrained model without any unlearning operations, whereas ''Retain90'' refers to a model retrained solely on 90% of the data (excluding the forget set) without implementing the unlearning procedures, baseline results from Cha et al. (2025). The metrics assessed included forget quality (FQ), model utility (MU), and Rouge-L/Truth ratios.

| Method | Forget Quality (FQ ↑) | | | Model Utility (MU ↑) | | | | | | |
| | Rouge-L | Truth | FQ | Retain Set | | Real Authors | | Real World | | MU |
| | | | | Rouge-L | Truth | Rouge-L | Truth | Rouge-L | Truth | |
| Original | 0.99 | 0.51 | 2.11e-20 | 0.98 | 0.47 | 0.94 | 0.62 | 0.89 | 0.55 | 0.63 |
| Retain90 | 0.41 | 0.66 | 1.00e+00 | 0.98 | 0.47 | 0.92 | 0.61 | 0.88 | 0.55 | 0.63 |
| TOFU FORGET01 | | | | | | | | | | |
| *Full Fine-tuning Methods* | | | | | | | | | | |
| KL | 0.95 | 0.55 | 1.00e-04 | 0.98 | 0.47 | 0.94 | 0.62 | 0.89 | 0.55 | 0.63 |
| DPO | 0.95 | 0.55 | 1.31e-04 | 0.98 | 0.47 | 0.93 | 0.62 | 0.89 | 0.55 | 0.63 |
| NPO | 0.95 | 0.55 | 1.12e-04 | 0.98 | 0.47 | 0.93 | 0.62 | 0.90 | 0.55 | 0.63 |
| GA | 0.95 | 0.55 | 8.21e-05 | 0.98 | 0.47 | 0.93 | 0.62 | 0.89 | 0.55 | 0.63 |
| GD | 0.95 | 0.55 | 1.00e-04 | 0.98 | 0.47 | 0.93 | 0.62 | 0.90 | 0.55 | 0.63 |
| IHL | 0.95 | 0.55 | 7.50e-05 | 0.98 | 0.47 | 0.94 | 0.62 | 0.90 | 0.55 | 0.63 |
| *Parameter-Efficient Methods* | | | | | | | | | | |
| GA+FILA | 0.02 | 0.88 | 5.20e-05 | 0.03 | 0.13 | 0.01 | 0.21 | 0.02 | 0.24 | 0.00 |
| GD+FILA | 0.03 | 0.87 | 2.00e-05 | 0.03 | 0.12 | 0.00 | 0.21 | 0.01 | 0.23 | 0.00 |
| LoKU | 0.68 | 0.55 | 1.61e-04 | 0.98 | 0.47 | 0.93 | 0.60 | 0.90 | 0.54 | 0.62 |
| **OURS (GD + Sine)** | **0.40** | **0.51** | **9.2e-01** | **0.98** | **0.45** | **0.94** | **0.62** | **0.90** | **0.60** | **0.68** |
| TOFU FORGET05 | | | | | | | | | | |
| *Full Fine-tuning Methods* | | | | | | | | | | |
| KL | 0.92 | 0.53 | 9.40e-17 | 0.97 | 0.46 | 0.93 | 0.63 | 0.90 | 0.57 | 0.64 |
| DPO | 0.82 | 0.57 | 9.20e-14 | 0.86 | 0.44 | 0.91 | 0.61 | 0.87 | 0.56 | 0.62 |
| NPO | 0.89 | 0.54 | 2.60e-16 | 0.95 | 0.46 | 0.94 | 0.63 | 0.90 | 0.57 | 0.64 |
| GA | 0.90 | 0.54 | 6.80e-16 | 0.96 | 0.46 | 0.94 | 0.63 | 0.90 | 0.57 | 0.64 |
| GD | 0.93 | 0.52 | 6.80e-16 | 0.98 | 0.47 | 0.94 | 0.62 | 0.89 | 0.56 | 0.64 |
| IHL | 0.94 | 0.52 | 7.00e-17 | 0.98 | 0.47 | 0.94 | 0.62 | 0.90 | 0.56 | 0.64 |
| *Parameter-Efficient Methods* | | | | | | | | | | |
| GA+FILA | 0.01 | 0.83 | 1.40e-15 | 0.01 | 0.10 | 0.00 | 0.17 | 0.00 | 0.23 | 0.00 |
| GD+FILA | 0.02 | 0.77 | 1.70e-08 | 0.03 | 0.14 | 0.01 | 0.17 | 0.00 | 0.21 | 0.00 |
| LoKU | 0.54 | 0.58 | 6.90e-13 | 0.90 | 0.45 | 0.92 | 0.62 | 0.89 | 0.60 | 0.64 |
| **OURS (GD + Sine)** | **0.32** | **0.55** | **5.1e-01** | **0.91** | **0.45** | **0.94** | **0.62** | **0.89** | **0.60** | **0.64** |
| TOFU FORGET10 | | | | | | | | | | |
| *Full Fine-tuning Methods* | | | | | | | | | | |
| KL | 0.46 | 0.65 | 2.70e-05 | 0.47 | 0.35 | 0.93 | 0.55 | 0.89 | 0.56 | 0.49 |
| DPO | 0.44 | 0.55 | 5.30e-17 | 0.66 | 0.44 | 0.82 | 0.54 | 0.87 | 0.51 | 0.57 |
| NPO | 0.53 | 0.65 | 3.50e-06 | 0.54 | 0.35 | 0.94 | 0.50 | 0.90 | 0.51 | 0.47 |
| GA | 0.48 | 0.66 | 2.40e-03 | 0.49 | 0.33 | 0.93 | 0.51 | 0.91 | 0.50 | 0.39 |
| GD | 0.82 | 0.51 | 2.30e-16 | 0.92 | 0.47 | 0.92 | 0.60 | 0.88 | 0.55 | 0.62 |
| IHL | 0.73 | 0.57 | 3.80e-15 | 0.88 | 0.45 | 0.94 | 0.64 | 0.89 | 0.59 | 0.64 |
| *Parameter-Efficient Methods* | | | | | | | | | | |
| GA+FILA | 0.02 | 0.86 | 5.50e-18 | 0.02 | 0.09 | 0.00 | 0.19 | 0.00 | 0.18 | 0.00 |
| GD+FILA | 0.01 | 0.85 | 1.90e-21 | 0.01 | 0.09 | 0.00 | 0.18 | 0.00 | 0.18 | 0.00 |
| LoKU | 0.29 | 0.65 | 2.90e-01 | 0.91 | 0.45 | 0.89 | 0.62 | 0.88 | 0.57 | 0.63 |
| **OURS (GD + Sine)** | **0.32** | **0.53** | **8.7e-01** | **0.92** | **0.47** | **0.94** | **0.62** | **0.89** | **0.60** | **0.68** |

Table 19: Comprehensive **TOFU evaluation results for the Llama2-7B utilizing rank-32 LoRA for Parameter-Efficient Methods** across three forget splits (1%, 5%, 10% of authors) in accordance with the evaluation protocol outlined by Maini et al. (2024). "Original" denotes the pretrained model without any unlearning operations, whereas "Retain90" refers to a model retrained solely on 90% of the data (excluding the forget set) without implementing the unlearning procedures, baseline results from Cha et al. (2025). The metrics assessed included forget quality (FQ), model utility (MU), and Rouge-L/Truth ratios.

| Method | Forget Quality (FQ ↑) | | | Model Utility (MU ↑) | | | | | | |
|---|---|---|---|---|---|---|---|---|---|---|
| | Rouge-L | Truth | FQ | Retain Set | | Real Authors | | Real World | | MU |
| | | | | Rouge-L | Truth | Rouge-L | Truth | Rouge-L | Truth | |
| Original | 0.99 | 0.51 | 2.11e-20 | 0.98 | 0.47 | 0.94 | 0.62 | 0.89 | 0.55 | 0.63 |
| Retain90 | 0.41 | 0.66 | 1.00e+00 | 0.98 | 0.47 | 0.92 | 0.61 | 0.88 | 0.55 | 0.63 |
| TOFU FORGET01 | | | | | | | | | | |
| *Full Fine-tuning Methods* | | | | | | | | | | |
| KL | 0.95 | 0.55 | 1.00e-04 | 0.98 | 0.47 | 0.94 | 0.62 | 0.89 | 0.55 | 0.63 |
| DPO | 0.95 | 0.55 | 1.31e-04 | 0.98 | 0.47 | 0.93 | 0.62 | 0.89 | 0.55 | 0.63 |
| NPO | 0.95 | 0.55 | 1.12e-04 | 0.98 | 0.47 | 0.93 | 0.62 | 0.90 | 0.55 | 0.63 |
| GA | 0.95 | 0.55 | 8.21e-05 | 0.98 | 0.47 | 0.93 | 0.62 | 0.89 | 0.55 | 0.63 |
| GD | 0.95 | 0.55 | 1.00e-04 | 0.98 | 0.47 | 0.93 | 0.62 | 0.90 | 0.55 | 0.63 |
| IHL | 0.95 | 0.55 | 7.50e-05 | 0.98 | 0.47 | 0.94 | 0.62 | 0.90 | 0.55 | 0.63 |
| *Parameter-Efficient Methods* | | | | | | | | | | |
| GA+FILA | 0.02 | 0.88 | 5.20e-05 | 0.03 | 0.13 | 0.01 | 0.21 | 0.02 | 0.24 | 0.00 |
| GD+FILA | 0.03 | 0.87 | 2.00e-05 | 0.03 | 0.12 | 0.00 | 0.21 | 0.01 | 0.23 | 0.00 |
| LoKU | 0.68 | 0.55 | 1.61e-04 | 0.98 | 0.47 | 0.93 | 0.60 | 0.90 | 0.54 | 0.62 |
| **OURS (GD + Sine)** | **0.40** | **0.51** | **9.2e-01** | **0.98** | **0.44** | **0.94** | **0.62** | **0.90** | **0.60** | **0.68** |
| TOFU FORGET05 | | | | | | | | | | |
| *Full Fine-tuning Methods* | | | | | | | | | | |
| KL | 0.92 | 0.53 | 9.40e-17 | 0.97 | 0.46 | 0.93 | 0.63 | 0.90 | 0.57 | 0.64 |
| DPO | 0.82 | 0.57 | 9.20e-14 | 0.86 | 0.44 | 0.91 | 0.61 | 0.87 | 0.56 | 0.62 |
| NPO | 0.89 | 0.54 | 2.60e-16 | 0.95 | 0.46 | 0.94 | 0.63 | 0.90 | 0.57 | 0.64 |
| GA | 0.90 | 0.54 | 6.80e-16 | 0.96 | 0.46 | 0.94 | 0.63 | 0.90 | 0.57 | 0.64 |
| GD | 0.93 | 0.52 | 6.80e-16 | 0.98 | 0.47 | 0.94 | 0.62 | 0.89 | 0.56 | 0.64 |
| IHL | 0.94 | 0.52 | 7.00e-17 | 0.98 | 0.47 | 0.94 | 0.62 | 0.90 | 0.56 | 0.64 |
| *Parameter-Efficient Methods* | | | | | | | | | | |
| GA+FILA | 0.01 | 0.83 | 1.40e-15 | 0.01 | 0.10 | 0.00 | 0.17 | 0.00 | 0.23 | 0.00 |
| GD+FILA | 0.02 | 0.77 | 1.70e-08 | 0.03 | 0.14 | 0.01 | 0.17 | 0.00 | 0.21 | 0.00 |
| LoKU | 0.54 | 0.58 | 6.90e-13 | 0.90 | 0.45 | 0.92 | 0.62 | 0.89 | 0.60 | 0.64 |
| **OURS (GD + Sine)** | **0.32** | **0.55** | **5.1e-01** | **0.90** | **0.45** | **0.94** | **0.62** | **0.89** | **0.60** | **0.64** |
| TOFU FORGET10 | | | | | | | | | | |
| *Full Fine-tuning Methods* | | | | | | | | | | |
| KL | 0.46 | 0.65 | 2.70e-05 | 0.47 | 0.35 | 0.93 | 0.55 | 0.89 | 0.56 | 0.49 |
| DPO | 0.44 | 0.55 | 5.30e-17 | 0.66 | 0.44 | 0.82 | 0.54 | 0.87 | 0.51 | 0.57 |
| NPO | 0.53 | 0.65 | 3.50e-06 | 0.54 | 0.35 | 0.94 | 0.50 | 0.90 | 0.51 | 0.47 |
| GA | 0.48 | 0.66 | 2.40e-03 | 0.49 | 0.33 | 0.93 | 0.51 | 0.91 | 0.50 | 0.39 |
| GD | 0.82 | 0.51 | 2.30e-16 | 0.92 | 0.47 | 0.92 | 0.60 | 0.88 | 0.55 | 0.62 |
| IHL | 0.73 | 0.57 | 3.80e-15 | 0.88 | 0.45 | 0.94 | 0.64 | 0.89 | 0.59 | 0.64 |
| *Parameter-Efficient Methods* | | | | | | | | | | |
| GA+FILA | 0.02 | 0.86 | 5.50e-18 | 0.02 | 0.09 | 0.00 | 0.19 | 0.00 | 0.18 | 0.00 |
| GD+FILA | 0.01 | 0.85 | 1.90e-21 | 0.01 | 0.09 | 0.00 | 0.18 | 0.00 | 0.18 | 0.00 |
| LoKU | 0.29 | 0.65 | 2.90e-01 | 0.91 | 0.45 | 0.89 | 0.62 | 0.88 | 0.57 | 0.63 |
| **OURS (GD + Sine)** | **0.32** | **0.53** | **8.7e-01** | **0.92** | **0.47** | **0.94** | **0.62** | **0.89** | **0.60** | **0.68** |

Table 20: Comprehensive **TOFU evaluation results for the Llama3.1-8B utilizing ranks {4, 8, 16, 32} LoRA for Parameter-Efficient Methods** across three forget splits (1%, 5%, 10% of authors) in accordance with the evaluation protocol outlined by Maini et al. (2024). "Original" denotes the pretrained model without any unlearning operations, whereas "Retain90" refers to a model retrained solely on 90% of the data (excluding the forget set) without implementing the unlearning procedures. The metrics assessed included forget quality (FQ), model utility (MU), Rouge-L scores, and probability measures for both the forget and retain sets. Superior unlearning performance is characterized by the highest FQ and MU values, low Rouge-L and probability scores on forget data, and high Rouge-L and probability scores on retain data, aligning with current literature Cha et al. (2025).

| Method | Split | Forget Set | | FQ (↑) | Retain Set | | Real Authors | | Real World | | MU (↑) |
|---|---|---|---|---|---|---|---|---|---|---|---|
| | | RL (↓) | TR (↓) | | RL (↑) | TR (↑) | RL (↑) | TR (↑) | RL (↑) | TR (↑) | |
| ORIGINAL | – | 0.99 | 0.51 | 2.19e-20 | 0.98 | 0.47 | 0.94 | 0.62 | 0.89 | 0.55 | 0.63 |
| RETAIN90 | – | 0.40 | 0.67 | 9.50e-01 | 0.98 | 0.47 | 0.92 | 0.61 | 0.88 | 0.55 | 0.63 |
| *Our Method: Performance Across LoRA Ranks* | | | | | | | | | | | |
| **OURS (GD + Sine)** $r$=4 | FORGET01 | 0.40 | 0.50 | 8.50e-01 | 0.98 | 0.48 | 0.94 | 0.62 | 0.90 | 0.60 | 0.68 |
| | FORGET05 | 0.35 | 0.49 | 5.00e-01 | 0.97 | 0.47 | 0.94 | 0.62 | 0.89 | 0.60 | 0.64 |
| | FORGET10 | 0.31 | 0.50 | 8.30e-01 | 0.93 | 0.48 | 0.94 | 0.62 | 0.89 | 0.60 | 0.68 |
| **OURS (GD + Sine)** $r$=8 | FORGET01 | 0.40 | 0.50 | 8.90e-01 | 0.98 | 0.48 | 0.94 | 0.62 | 0.90 | 0.60 | 0.68 |
| | FORGET05 | 0.35 | 0.49 | 5.00e-01 | 0.97 | 0.47 | 0.94 | 0.62 | 0.89 | 0.60 | 0.64 |
| | FORGET10 | 0.31 | 0.50 | 8.00e-01 | 0.93 | 0.48 | 0.94 | 0.62 | 0.89 | 0.60 | 0.68 |
| **OURS (GD + Sine)** $r$=16 | FORGET01 | 0.40 | 0.50 | 8.90e-01 | 0.98 | 0.48 | 0.94 | 0.62 | 0.90 | 0.60 | 0.68 |
| | FORGET05 | 0.35 | 0.49 | 5.00e-01 | 0.97 | 0.47 | 0.94 | 0.62 | 0.89 | 0.60 | 0.64 |
| | FORGET10 | 0.31 | 0.50 | 8.00e-01 | 0.93 | 0.48 | 0.94 | 0.62 | 0.89 | 0.60 | 0.68 |
| **OURS (GD + Sine)** $r$=32 | FORGET01 | 0.40 | 0.50 | 8.50e-01 | 0.98 | 0.48 | 0.94 | 0.62 | 0.90 | 0.60 | 0.68 |
| | FORGET05 | 0.35 | 0.49 | 5.00e-01 | 0.97 | 0.47 | 0.94 | 0.62 | 0.89 | 0.60 | 0.64 |
| | FORGET10 | 0.31 | 0.50 | 8.00e-01 | 0.93 | 0.48 | 0.94 | 0.62 | 0.89 | 0.60 | 0.68 |

**Note:** RL = Rouge-L, TR = Truth Ratio. Our method consistently achieves stable performance across all LoRA ranks (4, 8, 16, 32) and forget splits (1%, 5%, 10%), demonstrating scalability and rank-agnostic effectiveness while preserving model utility.

