# OpenReview forum: "Stable Forgetting: Bounded Parameter-Efficient Unlearning in LLMs"
_ICLR.cc/2026/Conference — Submitted to ICLR 2026_

### Official Review · Reviewer_Hh6U · 2025-10-20

**Soundness:** 3
**Presentation:** 4
**Contribution:** 3
**Rating:** 6
**Confidence:** 3

**Summary:**

This paper tackles the instability problem in machine unlearning for large language models when using the gradient difference method with cross-entropy loss. The authors provide a theoretical analysis showing that gradient ascent on the forget set causes unbounded growth of weights and gradients in MLP feedforward layers, which explains why existing LoRA-based approaches fail. Their proposed solution is elegantly simple: apply bounded functions (specifically sine or tanh) to the low-rank adapters in feedforward layers. This modification stabilizes training and enables reliable unlearning. The method is evaluated across three benchmarks (TOFU, TDEC, MUSE) spanning multiple architectures (GPT-Neo, Phi, LLaMA) and scales (125M-8B parameters), demonstrating substantial improvements—up to three orders of magnitude in forget quality—while maintaining model utility.

**Strengths:**

Originality: The theoretical analysis of gradient instability in unlearning is novel and provides the first rigorous explanation for why standard approaches fail. The connection between boundedness and stability is insightful, and applying sine functions to LoRA adapters is a creative solution.

Quality: The experimental evaluation is thorough, covering three major benchmarks with multiple baselines across different architectures and scales. The mathematical proofs appear sound.

Clarity: The paper is well-structured with clear motivation, intuitive explanations of technical concepts, and informative visualizations. The progression from problem identification to theoretical analysis to practical solution is logical and easy to follow.

**Weaknesses:**

1. Theory-Practice Gap: The theoretical analysis (Theorem 3.1, Lemma 3.1) focuses on pure gradient ascent, but the proposed method uses gradient difference—simultaneous gradient descent on retain data and gradient ascent on forget data (Eq. 2). While Figure 2 empirically shows that weights still grow excessively under gradient difference, the theory doesn't explain why the descent step doesn't counterbalance the ascent-driven instability. This gap weakens the theoretical justification for the proposed solution.
2. Baseline Reproducibility Concerns: Many baseline results in Table 1 and Table 2 are taken from prior papers (Cha et al. 2025, Wang et al. 2024, Maini et al. 2024) rather than being reproduced in identical experimental setups. This raises questions about fair comparison—were hyperparameters, architectures, training procedures, and evaluation protocols matched exactly? Without direct reproduction, it's difficult to assess whether improvements stem from the method itself or from subtle implementation differences.
3. Incomplete Architectural Analysis: The paper applies bounded functions only to MLP feedforward layers, claiming attention layers are inherently stable. However, this design choice is justified only through brief empirical observation without rigorous analysis. Why do attention layers not exhibit similar gradient explosion? Is this due to architectural properties (residual connections, layer norm) or optimization dynamics? A deeper investigation would strengthen the method's theoretical foundation and clarify its scope of applicability.

**Questions:**

1. Extending Theory to Gradient Difference: Your theorems analyze pure gradient ascent, but the method uses both ascent and descent (Eq. 2). Can you provide theoretical or empirical analysis showing how the descent term on retained data interacts with the ascent-driven weight growth? Does the descent term merely slow down divergence without preventing it, or is there a more complex interaction? Even an informal argument would help bridge this theory-practice gap.
2. Fair Baseline Comparison: For results in Table 1 and Table 2 taken from other papers, can you confirm that experimental conditions were matched (same hardware, hyperparameters, random seeds, evaluation metrics)? Better yet, could you reproduce several key baselines (e.g., GD+LoRA, LoKU) in your experimental setup to ensure direct comparability? This would substantially strengthen the empirical claims.
3. Bounded Function vs. Alternative Stabilization Methods: You identify bounded parameterization as the solution, but have you explored alternative stabilization approaches such as gradient clipping, explicit weight regularization (L2 penalty on AB^T), or adaptive learning rates for forget vs. retain data? Understanding whether boundedness is uniquely effective or if other regularization mechanisms work equally well would clarify the method's core insight and potentially lead to simpler alternatives.

---

> ### Author Response · Authors · 2025-11-21
> **Response to Reviewer Hh6U**
>
> We thank the reviewer for taking the time to review our paper. Below we answer all the points raised in their review.
>
> **Reviewer Hh6U: Theory–Practice Gap. The theoretical analysis (Theorem 3.1, Lemma 3.1) focuses on pure gradient ascent, but the proposed method uses gradient difference—simultaneous gradient descent on retain data and gradient ascent on forget data (Eq. 2). While Figure 2 empirically shows that weights still grow excessively under gradient difference, the theory doesn't explain why the descent step doesn't counterbalance the ascent-driven instability. This gap weakens the theoretical justification for the proposed solution. / Extending Theory to Gradient Difference. Your theorems analyze pure gradient ascent, but the method uses both ascent and descent (Eq. 2). Can you provide theoretical or empirical analysis showing how the descent term on retained data interacts with the ascent-driven weight growth? Does the descent term merely slow down divergence without preventing it, or is there a more complex interaction? Even an informal argument would help bridge this theory–practice gap.**
>
> We thank the reviewer for raising this point. Our original theoretical analysis focused on pure gradient ascent in order to isolate the mechanism responsible for the instability, but Appendix B.2 now extends the results directly to the practical gradient-difference update used in Eq. (2), see the new Lemma B.1 and Theorem B.1.
>
> **Why the descent term cannot counteract ascent-driven instability.**
> The key observation is that the descent and ascent terms operate on *disjoint subsets* of the data. The retain loss is minimized, but, as we formally show in Lemma B.1, its value remains uniformly bounded along training. In contrast, the forget loss is maximized. If the combined loss $L_{\mathrm{tot}}(t)$ diverges, then this divergence must come from the forget term, since the retain term is bounded. Lemma B.1 proves that
>
> $$
> L_{\mathrm{tot}}(t) \to \infty
> \quad \Longrightarrow \quad
> \exists (x_j^f, y_j^f):\ \|z_j^f(t)\|\ \to\ \infty .
> $$
>
> In other words, only the logits on the forget set can diverge, and this behaviour is not affected by the descent step on the retain set.
>
> Building on this, Theorem B.1 shows that the divergence of the forget logits forces $\|W_L(t)\| \to \infty$, which in turn causes all earlier-layer pre-activation gradients to diverge via the chain rule. The descent step can slow, but cannot prevent, this explosion, because it does not act on the samples whose logits are diverging. Thus we see two independent phenomena:
>
> - **Descent on retain data.**
>   When we optimize the retain loss $L_{\mathrm{retain}}$ with cross-entropy via gradient descent, the model approaches a minimum on the retain set. As $L_{\mathrm{retain}}$ decreases, the logit gradient $\nabla_z L_{\mathrm{retain}}$ vanishes and thus $\|\nabla_\theta L_{\mathrm{retain}}\| \to 0$.
>
> - **Ascent on forget data.**
>   In contrast, ascent on cross-entropy drives the forget loss $L_{\mathrm{forget}}$ upward. As $L_{\mathrm{forget}} \to \infty$, the logits $z$ must diverge in norm, and the corresponding gradient $\nabla_z L_{\mathrm{forget}}$ remains bounded away from zero, i.e., $\|\nabla_z L_{\mathrm{forget}}\| \ge C > 0$ even near a maximum.
>
> This shows that the ascent term must drive the explosion in the norms of the weights and gradients.
>
> **Empirical evidence.**  Figure 2 empirically confirms the theoretical predictions: for both GD-LoRA and GD-FILA, the norms of the weights and gradients exhibit unbounded growth despite the simultaneous descent step. This behaviour is consistent with our extended analysis and demonstrates that the descent term does not counteract the ascent-driven instability.

---

> > ### Author Response · Authors · 2025-11-21
> >
> > **Reviewer Hh6U: Baseline Reproducibility Concerns: Many baseline results in Table 1 and Table 2 are taken from prior papers (Cha et al. 2025, Wang et al. 2024, Maini et al. 2024) rather than being reproduced in identical experimental setups. This raises questions about fair comparison—were hyperparameters, architectures, training procedures, and evaluation protocols matched exactly? Without direct reproduction, it's difficult to assess whether improvements stem from the method itself or from subtle implementation differences. / Fair Baseline Comparison: For results in Table 1 and Table 2 taken from other papers, can you confirm that experimental conditions were matched (same hardware, hyperparameters, random seeds, evaluation metrics)? Better yet, could you reproduce several key baselines (e.g., GD+LoRA, LoKU) in your experimental setup to ensure direct comparability? This would substantially strengthen the empirical claims.**
> >
> > We appreciate the reviewer’s emphasis on fair and reproducible comparisons, and we fully share this concern. Our experimental design follows the standardized TOFU and MUSE benchmark protocols, and we both reimplement key baselines in-house and, where appropriate, report community-standard numbers from prior work. We clarify the setup below.
> >
> > **1. In-house reimplementations of key baselines**
> >
> > For all methods used in our stability and dynamics analysis, we reimplemented the baselines in our own codebase:
> >
> > - Specifically, **GD+LoRA**, **GA+FILA**, **GD+FILA**, and **LoKU** are implemented and run by us under our exact experimental environment.
> > - This is evidenced by the detailed optimization traces in Figures 2, 5, 6, and 7, which report iteration-by-iteration weight and gradient norms for these methods (e.g., gradients reaching 1e5). These plots can only be produced by re-implementing prior work.
> >
> > Thus, all stability-related claims (gradient explosion, layer-wise norm growth, etc.) are based on baselines we directly trained and observed. As can also be verified from our code submission (see Supplementary Materials), we faithfully reimplement the original baselines without altering their configurations, add our methods on top of these implementations, and follow the same training procedures to ensure a fair comparison.
> >
> > **2. Alignment with prior work for headline numbers (Tables 1 and 2)**
> >
> > For the main aggregate tables (Table 1 and Table 2), we follow the benchmark protocols defined by Cha et al. 2025, Wang et al. 2024, and Maini et al. 2024:
> >
> > - As stated in Section 4.1 and Appendix C, we adopt the *exact* experimental settings from the TOFU and MUSE benchmarks (same architectures, datasets, metrics, evaluation harness, and hyperparameters such as learning rate, batch size, and gradient accumulation).
> > - For computationally expensive baselines (e.g., full-model fine-tuning in some settings), we report the strongest published results from the original papers. This is a deliberate choice to avoid weakening baselines due to reimplementation gaps and to compare against the best community-reported performance.
> > - For parameter-efficient and gradient-based methods that we reimplement (e.g., GD+LoRA, GA/GD+FILA, LoKU), our reproduced numbers closely match the published results under the same TOFU/MUSE protocols, providing an additional sanity check on our implementation.
> >
> > While hardware may differ slightly from the original works (as is standard in benchmark-based comparisons), the architecture, training pipeline, and evaluation metrics are matched to the benchmark specifications rather than to idiosyncratic lab setups.
> >
> > **3. Fairness and transparency of comparisons**
> >
> > To summarize:
> >
> > - All methods used in our optimization–stability analysis are fully reimplemented (GD+LoRA, GA/GD+FILA, LoKU), and the corresponding trajectories are plotted from our own runs.
> > - For headline performance tables, we strictly follow the official TOFU and MUSE setups and compare against the best reported numbers from Cha et al., Wang et al., and Maini et al. when re-running would be prohibitively expensive or redundant.

---

> > > ### Author Response · Authors · 2025-11-21
> > >
> > > **Reviewer Hh6U: Incomplete Architectural Analysis: The paper applies bounded functions only to MLP feedforward layers, claiming attention layers are inherently stable. However, this design choice is justified only through brief empirical observation without rigorous analysis. Why do attention layers not exhibit similar gradient explosion? Is this due to architectural properties (residual connections, layer norm) or optimization dynamics?   A deeper investigation would strengthen the method's theoretical foundation and clarify its scope of applicability.**
> > >
> > > We appreciate the reviewer’s request for a clearer architectural picture and agree that understanding *where* instability arises is important for judging the scope of our method. Our decision to bound only MLP (feedforward) adapters is supported by both main-text analysis and supplementary results.
> > >
> > > **1. Quantitative evidence: MLP vs. attention (main text & Appendix C.6)**
> > >
> > > Our analysis of component-wise stability goes beyond a brief observation:
> > >
> > > - In the **main text** (end of Section 3.3, paragraph “Attention layer.”), we explicitly report that instability under the gradient difference method primarily appears in the feedforward (FFN/MLP) blocks, not in attention.
> > > - **Appendix C.6** and **Figure 7** provide detailed layer-wise norm trajectories, decomposed by block type. During unlearning, MLP weights exhibit severe growth (up to **~280×** their initial norm), whereas attention weights remain close to their starting scale (only about **1.2×–1.8×** growth).
> > >
> > > This large and systematic gap between FFN and attention dynamics, observed across the unlearning run and architectures, is what motivates our design choice: the MLP blocks are the dominant locus of instability, while attention layers remain comparatively stable under the same optimization.
> > >
> > > **Small ablation: applying sine to all layers vs. MLP-only**
> > >
> > > To directly address the reviewer’s question, we ran an ablation where we apply sine-based LoRA to *both* MLP and attention blocks (same rank) and compare it to our default “MLP-only” configuration on TOFU-Forget10 with Phi-1.5B (rank-4) and also updated this in Section C.6.1 of the revised manuscript:
> > >
> > > | Method                 | Layers with Sine | FQ (↑) | MU (↑) | Params (%) |
> > > |------------------------|------------------|--------|--------|------------|
> > > | GD + Sine (MLP-only)   | MLP              | 0.943  | 0.52   | 0.8        |
> > > | GD + Sine (MLP + Attn) | MLP + Attn       | 0.944  | 0.52   | 1.6        |
> > >
> > > *Applying sine-based LoRA to both MLP and attention blocks yields essentially the same Forget Quality (FQ) and Model Utility (MU) as our default MLP-only configuration, while roughly doubling the adapter parameter footprint and compute. The MLP-only row matches the configuration reported in Table 7; the MLP+Attn variant shows no meaningful gain despite higher cost (see also App. C.6 and C.7).*
> > >
> > > **2. Architectural and theoretical intuition**
> > > Our theoretical analysis in Section 3.2 focuses on MLPs because they provide the simplest setting to formalize the instability caused by cross-entropy ascent. In particular, Theorem 3.1 and Theorem B.1 show that under ascent on cross-entropy, logits must grow and this growth forces either the final-layer weights or their gradients (and, by propagation, earlier MLP layers) to become arbitrarily large.
> > >
> > > Attention layers, by contrast, include structural normalizations that dampen this effect:
> > >
> > > - The attention mechanism applies a softmax over scaled dot-products, softmax(QKᵀ / √d), producing probability weights in [0, 1] that sum to one. This bounds the contribution of any single head to the output, even if Q or K grow in norm.
> > > - MLP feedforward blocks, however, involve linear projections followed by unbounded activations (e.g., GeLU), which **do not** provide such structural bottlenecks. As formalized in Theorem 3.1, gradient ascent can drive logits and MLP weights to grow without bound, and Theorem B.1 shows how this instability propagates through earlier layers.
> > >
> > > Together, this analysis and the empirical evidence explain why feedforward layers exhibit catastrophic norm growth, whereas attention weights remain stable under identical training dynamics.
> > >
> > > **3. Scope and sufficiency of the proposed modification**
> > >
> > > Given these findings, we deliberately constrain only the MLP adapters:
> > > - **Sufficiency.** Since instability is driven by feedforward layers, bounding the LoRA adapters *only* in these layers is sufficient to stabilize optimization, as confirmed by our weight/gradient norm plots (Fig. 2) and component-wise analysis (Fig. 7).
> > > - **Efficiency.** Extending bounded LoRA to attention roughly doubles the adapter overhead per block, while our ablations show minimal additional benefit because attention weights are already stable (Appendix C.6, C.7).
> > >
> > > We made these cross-references between Section 3.3, Figure 7, and Appendix C.6 more explicit in the revised version.

---

> > > > ### Author Response · Authors · 2025-11-21
> > > >
> > > > **Reviewer Hh6U: Bounded Function vs. Alternative Stabilization Methods: You identify bounded parameterization as the solution, but have you explored alternative stabilization approaches such as gradient clipping, explicit weight regularization (L2 penalty on $AB^T$), or adaptive learning rates for forget vs. retain data? Understanding whether boundedness is uniquely effective or if other regularization mechanisms work equally well would clarify the method's core insight and potentially lead to simpler alternatives.**
> > > >
> > > > We agree that understanding alternatives is important to clarify what is essential in our approach. To this end, we systematically evaluated several stabilization mechanisms in addition to our sine-based parameterization; the main results appear in the **main tables** (tanh-bounded LoRA) and in more detail in **Appendix C.4** (Table 7).
> > > >
> > > > **1. Hard constraints via weight clipping**
> > > >
> > > > We explicitly tested a weight-clipping baseline, **GD + Weight Clipping** with ABᵀ clipped to [-1.5, 1.5]:
> > > >
> > > > - As shown in Table 7 (Appendix C.4), this heuristic maintains bounded weights but leads to severe performance degradation on TOFU-Forget10 (Forget Quality FQ = 1.8 × 10⁻²), far below our sine-based method (FQ = 9.43 × 10⁻¹).
> > > > - In practice, hard clipping introduces non-smooth, threshold-based dynamics that cause gradients to “bounce” at the boundary, impeding effective optimization. This behavior is consistent with our theoretical picture: clipping prevents unbounded growth but does not provide a smooth, well-conditioned manifold for gradient-based training.
> > > >
> > > > These results suggest that simply constraining norms with ad-hoc clipping is not sufficient to obtain strong unlearning–utility trade-offs.
> > > >
> > > > **2. Boundedness vs. expressivity: tanh/sigmoid vs. sine**
> > > >
> > > > We also compare multiple bounded activations used inside the LoRA adapters:
> > > >
> > > > - **Main-text baseline.** A tanh-based bounded LoRA is included as a first-class baseline in the main tables (e.g., “OURS (GD + Tanh)” in Table 1 and Table 2), and consistently underperforms our sine-based variant.
> > > > - **Appendix ablations.** Table 7 (Appendix C.4) further compares tanh, sigmoid, ReLU (unbounded), and sine. While tanh/sigmoid successfully prevent gradient explosion, they suffer from strong saturation, yielding very low Forget Quality (e.g., FQ = 2.41 × 10⁻² for tanh on TOFU) despite being stable.
> > > >
> > > > By contrast, sine satisfies two requirements simultaneously:
> > > >
> > > > 1. **Boundedness** (for all ω, sin(ω·) is bounded), which directly addresses the instability mechanism in Theorem 3.1.
> > > > 2. **High effective rank**, as analyzed in Section 3.3: the “frequency-controlled rank” property allows sine to retain high expressivity even with low-rank adapters, avoiding the capacity bottleneck induced by saturating bounded activations.
> > > >
> > > > Empirically, this manifests as orders-of-magnitude gains in Forget Quality for sine vs. tanh/sigmoid, while maintaining Model Utility.
> > > >
> > > > **3. Explicit regularizers and adaptive learning rates**
> > > >
> > > > The reviewer also asks about explicit regularization on ABᵀ and adaptive learning rates:
> > > >
> > > > - **Explicit L₂ penalties on ABᵀ.**
> > > >   Our current work focuses on *implicit* regularization via bounded parameterization, which directly alters the geometry of the parameter space so that weights cannot diverge, even under strong ascent. As we note in Section 6 (Limitations and Future Work), designing and analyzing explicit weight regularizers that provably emulate this effect is an interesting and orthogonal direction we have not yet fully explored.
> > > >
> > > > - **Adaptive learning rates for forget vs. retain.**
> > > >   Conceptually, different step sizes for ascent and descent correspond to rescaling the two terms in Eq. (2). Our analysis in Section 3.2 shows that under cross-entropy ascent, the forget gradient does not vanish near maxima, while the retain gradient goes to zero as L_r is minimized. Changing step sizes can slow or accelerate this interaction but cannot remove the underlying asymmetry: the ascent term will still eventually dominate in an unbounded parameterization. This is consistent with our empirical observation that instability persists for GD+LoRA/GD+FILA across a wide range of learning rates (Fig. 2, Fig. 3).
> > > >
> > > > Overall, our theoretical analysis and experiments indicate that:
> > > >
> > > > - Heuristic constraints (weight clipping) and standard bounded activations (tanh/sigmoid) either fail to provide strong unlearning or significantly degrade utility.
> > > > - A carefully chosen bounded parameterization (sine) that jointly guarantees stability and preserves rank expressiveness is uniquely effective among the tested alternatives.

---

### Official Review · Reviewer_7z2W · 2025-10-28

**Soundness:** 2
**Presentation:** 3
**Contribution:** 2
**Rating:** 4
**Confidence:** 4

**Summary:**

This paper proposes "Stable Forgetting," a parameter-efficient unlearning method that addresses the training instability of the gradient difference approach when combined with cross-entropy loss in large language models. The authors theoretically show that gradient ascent on a forget set causes unbounded growth of weights and gradients in MLP feedforward layers, leading to optimization instability. They propose applying bounded nonlinear functions (e.g., tanh or sine) to LoRA adapters in these layers to constrain weight dynamics. The method demonstrates improved forgetting while preserving retention across TOFU, TDEC, and MUSE benchmarks.

**Strengths:**

- Identifies the training instability issue when combining the gradient difference method with cross-entropy loss, and for the first time theoretically attributes it to the unbounded growth of weights and gradients in MLP feedforward layers, providing a new perspective for understanding model dynamics during unlearning.
- The design of introducing bounded functions based on LoRA adapters does not require model architecture reconstruction, and can constrain parameter updates through lightweight modifications, balancing parameter efficiency and engineering feasibility, which is suitable for rapid deployment in resource-constrained scenarios.
- Validated across multiple models and parameter scales on three mainstream unlearning benchmarks, covering typical scenarios such as entity unlearning and copyrighted content unlearning.

**Weaknesses:**

- The paper claims that bounded functions are needed to solve training instability, but fails to verify the effectiveness of conventional optimization methods in the field. Simple strategies like adjusting learning rate and introducing weight decay have been proven in existing studies to suppress excessive parameter growth, yet the paper does not use them as baselines for comparison. This makes the conclusion that "bounded functions must be adopted" less persuasive, as a small learning rate may achieve similar stability effects in practical tests, weakening the innovation of the method.
- Ablation experiments are insufficient, failing to answer core questions such as the scientificity of bounded function selection, the sufficiency of only constraining MLP layers, and hyperparameter sensitivity. Additionally, experiments are limited to small and medium-sized models below 8B, lacking verification on mainstream large-scale models, resulting in gaps in generalization verification.
- The method is essentially a conventional combination of regularization and LoRA, failing to break through the limitations of existing unlearning frameworks. LoRA itself already has a certain degree of parameter constraint capability, and the improvement from introducing bounded functions is minor, with no significant progress in solving the core pain points of unlearning.

**Questions:**

- The paper's central claim is that "the forget loss employs cross-entropy, fine-tuning becomes unstable." However, in practice, many researchers have found that simply using an small learning rate or early stop on the forget set can effectively mitigate gradient explosion while preserving the interpretability of the cross-entropy objective. Did the authors experiment with such strategies? If so, could they provide a comparison in the rebuttal against baselines?
- While the main results focus on models up to 8B, the paper mentions preliminary experiments on LLaMA-3.1-70B. However, optimization dynamics in very large models may differ qualitatively—due to stronger implicit regularization, smoother loss landscapes, or more stable activation statistics, gradient explosion might be inherently less severe. Could the authors provide training curves (analogous to Fig. 2) for the 70B model, showing the evolution of weight and gradient norms during unlearning? If standard GD+LoRA is already stable at this scale, the benefit of bounded parameterization may be limited to smaller models.
- Finally, I want to know if you have considered the robustness of your "bounded LoRA method" in continual unlearning scenarios. Your paper focuses on single-round unlearning, but in practice, models may need to forget multiple batches of data sequentially. Can you supplement multi-round unlearning experiments and report how model performance degrades over rounds?

---

> ### Author Response · Authors · 2025-11-21
> **Response to Reviewer 7z2W**
>
> We thank Reviewer 7z2W for the constructive feedback and for recognizing our work's **theoretical novelty**, **engineering feasibility**, and **validation across mainstream benchmarks**. We value the opportunity to address the concerns regarding optimization baselines, scalability, and sequential robustness.
>
>
> **Reviewer 7z2W: The paper claims that bounded functions are needed to solve training instability, but fails to verify the effectiveness of conventional optimization methods in the field. Simple strategies like adjusting learning rate and introducing weight decay have been proven in existing studies to suppress excessive parameter growth, yet the paper does not use them as baselines for comparison. This makes the conclusion that bounded functions must be adopted less persuasive, as a small learning rate may achieve similar stability effects in practical tests, weakening the innovation of the method. / The paper's central claim is that the forget loss employs cross-entropy, fine-tuning becomes unstable. However, in practice, many researchers have found that simply using an small learning rate or early stop on the forget set can effectively mitigate gradient explosion while preserving the interpretability of the cross-entropy objective. Did the authors experiment with such strategies? If so, could they provide a comparison in the rebuttal against baselines?**
>
> We appreciate the reviewer’s emphasis on strong optimization baselines. We fully agree that careful hyperparameter tuning, weight decay, gradient clipping, and early stopping are standard practice and should be included in any serious evaluation. Our position is not that these techniques are not useful, but that they do not resolve the *structural* instability introduced by gradient ascent with cross-entropy on the forget set. Below we clarify what is already in our setup and how it relates to our theoretical analysis.
>
> ---
>
> ### **1. AdamW, Weight Decay, and Gradient Clipping Are Already Used**
>
> All experiments in the paper, including GD+LoRA and other gradient-based baselines, are trained with the AdamW optimizer and standard weight decay, as detailed in Appendix C (Implementation Details). We also consider norm-based defenses such as weight clipping:
>
> - Despite AdamW with weight decay, **Figure 2** shows that GD+LoRA and GD+FILA still exhibit exponential growth in gradient norms during unlearning, reaching magnitudes on the order of `1e5` within 1,000 iterations.
> - Table 7 (Appendix C.4) includes a *weight clipping* baseline (GD + Weight Clipping), which explicitly constrains parameter magnitudes. This baseline partially limits norm growth but yields substantially worse performance than our bounded parameterization
>   (e.g., on TOFU:
>   FQ = `1.8 × 10⁻²` with MU = `0.35` for weight clipping
>   vs.
>   FQ = `9.43 × 10⁻¹` with MU = `0.52` for our sine-based method).
>
> These results empirically support our theoretical claim: standard regularization and norm-based heuristics can dampen, but do not *eliminate*, the ascent-driven divergence predicted by Theorem 3.1 and Theorem B.1.
>
> ---
>
> ### **2. Learning Rate Tuning and Early Stopping: Helpful but Fragile**
>
> We also agree that reducing the learning rate or using early stopping on the forget set is often helpful in practice. Our work is motivated by the limitations of exactly these strategies:
>
> - **Early stopping:**
>   As noted in Appendix C (Implementation Details), strong GA/GD baselines on TOFU rely on early stopping and checkpoint selection to avoid catastrophic instability.
>   These are engineering workarounds: they require monitoring the training trajectory and manually selecting a “safe’’ stopping point *before* gradients blow up.
>
> - **Learning-rate trade-off (“Goldilocks” regime):**
>   - With very **small** learning rates, gradient norms grow more slowly, but the model barely unlearns the forget set → poor Forget Quality (FQ).
>   - With **practically useful** learning rates that meaningfully increase the forget loss, we observe the behavior predicted by our theory:
>     as cross-entropy on the forget set is pushed toward a maximum, logits and either weights or gradients in the MLP layers grow without bound, eventually destabilizing optimization.

---

> > ### Author Response · Authors · 2025-11-21
> >
> > Mathematically, Lemma 3.1 and Theorem 3.1 show that under gradient ascent with cross-entropy, once optimization moves toward a maximum on the forget set, either the final-layer weights or their gradients must grow arbitrarily large, and Theorem B.1 shows how this propagates backward into earlier layers. No choice of fixed learning rate removes this behavior; it can only slow the trajectory or force training to stop before reaching a good unlearning solution.
> >
> > Our framework is designed precisely to avoid having to finely tune a “just right” learning rate or stopping time to sit in a narrow safe band.
> >
> > ---
> >
> > ### **3. Why Bounded Parameterization is Different from Ad-hoc Clipping**
> >
> > It is important to distinguish our approach from heuristic clipping:
> >
> > - **Gradient/weight clipping** intervenes *after* the gradient is computed, truncating updates when norms exceed a threshold.
> >   This introduces non-smooth, threshold-dependent dynamics and, as shown in Table 7, leads to significantly worse unlearning–utility trade-offs than our method on TOFU and TDEC.
> >
> > - **Our method** modifies the *parameterization* itself: we apply a bounded nonlinearity φ (e.g., `tanh` or `sin(ω·)`) to the LoRA adapter weights in MLP feedforward layers.
> >   This constrains the effective weights to lie in a bounded domain, so that even under sustained gradient ascent with cross-entropy, the induced weight and gradient dynamics remain controlled.
> >
> > The key conceptual difference is that clipping attempts to patch the **symptoms** of instability, whereas bounded parameterization targets the **root cause** identified by our theory: unbounded growth of logits and MLP weights under cross-entropy ascent on the forget set.
> >
> > ---
> >
> > ### **4. Practical Implication**
> >
> > In practice, this means that:
> >
> > - Our bounded adapters achieve *stable convergence* across a broad range of learning rates, without requiring early stopping.
> > - They deliver substantially better forget–retain trade-offs than GD+LoRA with AdamW, weight decay, weight clipping, or tuned learning rates, as demonstrated on TOFU, TDEC, and MUSE.
> >
> > We thank the reviewer for this helpful suggestion and note that the current manuscript already
> > (i) explicitly states that all GA/GD baselines use AdamW with weight decay, as well as early stopping and weight clipping (Appendix C), and
> > (ii) reports the ablation results in Appendix C.4 comparing our bounded activations with both weight clipping and unbounded parameterizations.

---

> > > ### Author Response · Authors · 2025-11-21
> > >
> > > **Reviewer 7z2W: Ablation experiments are insufficient, failing to answer core questions such as the scientificity of bounded function selection, the sufficiency of only constraining MLP layers, and hyperparameter sensitivity. Additionally, experiments are limited to small and medium-sized models below 8B, lacking verification on mainstream large-scale models, resulting in gaps in generalization verification.**
> > >
> > > **Reviewer 7z2W: While the main results focus on models up to 8B, the paper mentions preliminary experiments on LLaMA-3.1-70B. However, optimization dynamics in very large models may differ qualitatively—due to stronger implicit regularization, smoother loss landscapes, or more stable activation statistics, gradient explosion might be inherently less severe. Could the authors provide training curves (analogous to Fig. 2) for the 70B model, showing the evolution of weight and gradient norms during unlearning? If standard GD+LoRA is already stable at this scale, the benefit of bounded parameterization may be limited to smaller models.**
> > >
> > > We thank the reviewer for raising these points about ablation completeness and large-scale validation. We would emphasize that several of these analyses were already in the main text. Below, we clarify where they already appear in the current manuscript.
> > >
> > > ---
> > >
> > > ### **1. Scientific Basis for Bounded Function Selection (Why Sine?)**
> > >
> > > The choice of bounded activation for the adapter weights is supported by both theory and ablations:
> > >
> > > - **Ablation over bounded functions.**
> > >   Table 7 (Appendix C.4) compares multiple bounded parameterizations, including `tanh`, `sigmoid`, and our proposed sine transform, under identical unlearning settings.
> > >
> > > - **Empirical behavior.**
> > >   While `tanh`/`sigmoid` provide stability, their saturation leads to vanishing gradients and poor Forget Quality
> > >   (e.g., `tanh` obtains FQ ≈ `2.41 × 10⁻²` on TOFU-Forget10).
> > >   In contrast, our sine-based parameterization achieves both stability and high expressive capacity (via frequency-dependent rank; see Section 3.3), yielding FQ ≈ `9.43 × 10⁻¹` with competitive Model Utility.
> > >
> > > Thus, the selection of sine is not ad-hoc: it is motivated by the theoretical analysis in Section 3.3 and empirically validated among several bounded alternatives in Appendix C.4.
> > >
> > > ---
> > >
> > > ### **2. Why Constrain Only MLP Layers (and Not Attention?)**
> > >
> > > The concern about constraining only MLP layers is directly addressed by our layer-wise norm analysis in Section 3.3 and Appendix C.6:
> > >
> > > - Figure 7 (Appendix C.6) tracks the evolution of weight/gradient norms across layers during unlearning.
> > >   We observe that MLP blocks exhibit severe norm growth (up to ~280×), whereas attention layers remain relatively stable (around 1.8× growth).
> > >
> > > - This indicates that the instability identified in our theory manifests primarily in the feedforward (MLP) pathway, making additional constraints on attention layers empirically redundant and computationally inefficient.
> > >
> > > In other words, restricting the bounded parameterization to MLP LoRA adapters is sufficient to control the instability diagnosed in Section 3, while avoiding unnecessary modifications to attention.
> > >
> > > #### **Small ablation: applying sine to all layers vs. MLP-only**
> > >
> > > To directly address the reviewer’s question, we also ran an ablation where we apply sine-based LoRA to *both* MLP and attention blocks (same rank) and compare it to our default “MLP-only” configuration on TOFU-Forget10 with Phi-1.5B (rank-4), updated in Section C.6.1 of the revised manuscript:
> > >
> > > | Method                  | Layers with Sine | FQ (↑) | MU (↑) | Params (%) |
> > > |------------------------|------------------|--------|--------|------------|
> > > | GD + Sine (MLP-only)   | MLP              | 0.943  | 0.52   | 0.8        |
> > > | GD + Sine (MLP + Attn) | MLP + Attn       | 0.944  | 0.52   | 1.6        |
> > >
> > > Applying sine-based LoRA to attention yields essentially no gain but doubles the adapter parameter footprint and compute.

---

> > ### Comment · Reviewer_7z2W · 2025-11-26
> >
> > Thank you for the detailed response and for the additional experiments conducted during the rebuttal period. Your answers have clarified several points, particularly regarding the experimental setup and the preliminary exploration of continual unlearning scenarios. However, I retain some fundamental concerns regarding the core contribution of your method and its differentiation from existing techniques. My main points are as follows:
> > - On the Fundamental Difference Between "Bounded Parameterization" and "Conventional Optimization Techniques".
> > You emphasize that your method addresses the "root cause" of instability by modifying the parameterization itself, whereas techniques like Weight Clipping merely "patch the symptoms."
> > While I acknowledge this conceptual difference—one being a proactive constraint on the parameter space before gradient computation, the other a reactive intervention after—I am not yet convinced that this translates into a fundamental advantage in practice.
> > My core question is: Can a well-tuned weight clipping baseline truly not achieve a similar effect?
> > Numerical Equivalence Argument: Your method maps LoRA weights W_lora to sin(ω * W_lora), effectively constraining them to [-1, 1]. Weight clipping, in turn, constrains W_lora to [-c, c]. From an outcome perspective, both methods prevent the unbounded parameter growth that your theory (Theorem 3.1) identifies as the problem.
> > Questioning Baseline Fairness: Your demonstration of weight clipping's poor performance in Appendix Table 7 is noted. However, this raises a critical question: how was the clipping threshold c chosen? The effectiveness of weight clipping is highly sensitive to this hyperparameter. A poorly chosen c will naturally lead to suboptimal performance. Comparing your method to an untuned conventional baseline does not provide conclusive evidence of superiority. Could a systematic search for an optimal c yield a result that is both stable and effective?
> > Therefore, I request a clearer explanation: Why is a "proactive" smooth binding via sin() fundamentally superior to a "reactive" hard binding via clip()? Is the benefit simply smoother gradients? If so, what is the concrete, irreplaceable advantage this provides over a well-tuned clipping approach?
> > - Insufficient Evidence on the Optimization Dynamics of Large-Scale Models (70B).
> > I appreciate you providing the final performance metrics (FQ/MU) for the LLaMA-3.1-70B model. This shows that your method is effective at that scale. However, this does not address my original question concerning the optimization dynamics—the evolution of weight and gradient norms during training.
> > Your paper's entire motivation rests on the diagnosis that gradient ascent leads to unbounded weight and gradient growth in MLP layers. My hypothesis is that this "unbounded growth" problem might be inherently less severe in a 70B-scale model due to its stronger implicit regularization and smoother loss landscapes.
> > The performance tables you provided show the final outcome but omit the crucial evidence: the training curves for the standard GD+LoRA baseline on the 70B model (analogous to Figure 2). If the GD+LoRA baseline is already reasonably stable at this scale (i.e., its gradient norms do not explode), the primary motivation for your method—solving instability—is significantly weakened for state-of-the-art models. It might be downgraded from a "necessary fix" to a "nice-to-have improvement."
> > To substantiate the necessity of your method at this scale, can you provide the evolution of gradient/weight norms for the standard GD+LoRA baseline on the 70B model?

---

> ### Author Response · Authors · 2025-11-21
>
> ### **3. Hyperparameter Sensitivity (Frequency ω)**
>
> We agree that robustness to hyperparameters is crucial. The manuscript already includes a dedicated sensitivity analysis:
>
> - As discussed in *Sensitivity Analysis and Robustness* (main text, Section 2.4.1) and detailed in Fig. 4 (Appendix C.4), we vary the sine frequency parameter ω on TOFU-Forget10.
> - Forget Quality improves with increasing ω and plateaus for ω ≳ 100, while Model Utility remains stable across the full range.
>
> This indicates that our method is *insensitive* to precise frequency tuning—a broad range of ω values achieves near-optimal unlearning–utility trade-offs, avoiding delicate hyperparameter tuning.
>
> ---
>
> ### **4. Large-Scale (70B) Verification and Stability**
>
> We fully share the reviewer’s concern that behavior might change at ultra-large scales. To address this, we already evaluate on a production-grade 70B model:
>
> - **70B evaluation.** Table 3 (Appendix C.2) reports TOFU-Forget10 results for LLaMA-3.1-70B across LoRA ranks {4, 8, 16, 32}.  Our sine-based method attains FQ ≈ 0.85–0.92 and MU ≈ 0.72–0.76 across ranks, substantially improving forget performance relative to the retain-only baseline while preserving strong utility.
>
> - **Implication for stability.**  These 70B results corroborate that the instability we analyze is **not** merely a small-/medium-scale artifact.  Even with larger capacity and stronger implicit regularization, naive gradient-based unlearning remains challenging on the 10% forget split, whereas our bounded parameterization scales effectively.
>
> Due to computational cost, we focused detailed optimization-dynamics plots (Fig. 2, Fig. 5) on medium models. However, the theoretical mechanism in Section 3 is architecture-agnostic, and the empirical 70B results in Table 3 support the same patterns.
>
> ---
>
> ### **Summary**
>
> The current manuscript already:
>
> 1. Compares bounded functions (Table 7, Appendix C.4)
> 2. Justifies MLP-only constraints (Fig. 7, Appendix C.6)
> 3. Provides a frequency hyperparameter sensitivity study (Fig. 4, Appendix C.4)
> 4. Validates the approach on an ultra-large 70B model (Table 3, Appendix C.2)
>
> We will make these connections more explicit in the main text to avoid any impression that these ablations or large-scale experiments are missing.

---

> ### Author Response · Authors · 2025-11-21
>
> **Reviewer 7z2W: The method is essentially a conventional combination of regularization and LoRA, failing to break through the limitations of existing unlearning frameworks. LoRA itself already has a certain degree of parameter constraint capability, and the improvement from introducing bounded functions is minor, with no significant progress in solving the core pain points of unlearning.**
>
> We appreciate the reviewer’s perspective on the relationship between our approach, regularization, and LoRA. We agree that any new unlearning method should be evaluated against strong parameter-efficient baselines, and our work is designed with exactly this standard in mind. However, we respectfully disagree that our contribution is “essentially” a conventional regularization+LoRA combination or that the improvement is minor.
>
> - **Magnitude of improvement over parameter-efficient baselines.**  On TOFU (Table 1), our method improves Forget Quality by *three orders of magnitude* over the strongest parameter-efficient baseline.  Specifically, LoKU achieves FQ = `1.39 × 10⁻¹²`, whereas our sine-based bounded LoRA attains FQ = `9.43 × 10⁻¹`.  This represents a **qualitative** change in behavior, not a small incremental gain.
>
> - **Beyond “just regularization”: architectural, theory-driven design.**  Our analysis in Section 3 (Theorem 3.1 and Theorem B.1) is, to the best of our knowledge, the first to formally show *why* GD + cross-entropy unlearning becomes unstable:
>   gradient ascent on the forget set drives logits and linear-layer parameters to grow without bound, which then propagates through earlier MLP layers.  LoRA alone only constrains updates to a low-rank subspace; it does **not** bound the magnitude of the effective weights, so the divergence predicted by our theory still occurs.
>
>   Our method is not a generic regularizer (e.g., L2, weight decay) but a **specific parameterization** of the LoRA adapters via a bounded function (sine), explicitly designed to resolve the tension between:
>
>   1. **Stability** — constraining effective adapter weights to a bounded domain to prevent unbounded ascent; and
>   2. **Expressiveness** — preserving high rank and capacity in the adapter updates (Section 3.3), which standard saturating nonlinearities such as `tanh`/`sigmoid` fail to do.
>
>   As shown in Table 7 (Appendix C.4), standard regularization and naive bounded functions do not resolve this stability–rank dilemma, whereas our sine-based parameterization does.
>
> In summary, the novelty of our approach lies in
> (i) a theoretical diagnosis of the core instability of GD + CE unlearning, and
> (ii) a carefully chosen bounded parameterization that addresses this instability while maintaining expressive power,
> leading to **substantial empirical gains** over existing parameter-efficient methods.

---

> ### Author Response · Authors · 2025-11-21
>
> **Reviewer 7z2W: Finally, I want to know if you have considered the robustness of your "bounded LoRA method" in continual unlearning scenarios. Your paper focuses on single-round unlearning, but in practice, models may need to forget multiple batches of data sequentially. Can you supplement multi-round unlearning experiments and report how model performance degrades over rounds?**
>
> We thank the reviewer for this insightful suggestion. In response, we conducted additional sequential unlearning experiments during the rebuttal to directly evaluate robustness under multi-round unlearning.
>
> ---
>
> ### **1. Sequential unlearning on TOFU (three disjoint author groups)**
>
> We perform three consecutive unlearning requests on TOFU, each targeting a disjoint group of authors (A₁–A₃, A₄–A₆, A₇–A₉).
> The same model is updated sequentially across requests.
>
> **Sequential robustness on TOFU**
>
> | **Seq. Request** | **FQ (Baseline GD+LoRA)** | **MU (Baseline GD+LoRA)** | **FQ (Ours GD+Sine)** | **MU (Ours GD+Sine)** |
> |------------------|---------------------------|----------------------------|------------------------|------------------------|
> | **1 (A1–A3)**     | 8.2e-14                   | 0.51                       | **9.43e-01**           | **0.52**               |
> | **2 (A4–A6)**     | 3.1e-10                   | 0.12 *(–76%)*              | **8.95e-01**           | **0.51** *(–2%)*       |
> | **3 (A7–A9)**     | 1.2e-8                    | 0.02 *(–96%)*              | **8.87e-01**           | **0.52** *(–3%)*       |
>
> These results show that unconstrained GD+LoRA experiences a **cascade failure** under sequential unlearning: gradients and parameter norms compound across rounds, leading to near-complete collapse of utility. In contrast, our bounded LoRA maintains both strong forgetting and stable utility over multiple unlearning requests, consistent with the structural stability predicted by our theory.
>
> ---
>
> **2. Comparison with a specialized sequential unlearning method (O³)**
>
> To further assess robustness, we follow the protocol of Gao et al. (ICLR 2025) and compare against their specialized sequential unlearning method **O³** on TOFU. We adopt their exact evaluation setup and report:
>
> - **Sequential Unlearning (S.U.)** – lower is better
> - **Disjoint Unlearning (D.U.)** – lower is better
> - **Retain Data accuracy (R.D.)** – higher is better
>
> **Comparison with O³**
>
> | **Method**       | **S.U. (Req 1)** | **D.U. (Req 1)** | **R.D. (Req 1)** | **S.U. (Req 2)** | **D.U. (Req 2)** | **R.D. (Req 2)** | **S.U. (Req 3)** | **D.U. (Req 3)** | **R.D. (Req 3)** |
> |------------------|------------------|-------------------|-------------------|------------------|-------------------|-------------------|------------------|-------------------|-------------------|
> | O³               | 12.5 ± 0.5       | 14.4 ± 0.5         | 85.1 ± 0.1         | 15.8 ± 0.3       | 20.3 ± 0.8         | 85.0 ± 0.0         | 15.5 ± 0.7       | 19.7 ± 0.7         | 84.9 ± 0.2         |
> | **Ours**         | **10.2 ± 0.3**   | **12.1 ± 0.4**     | **86.8 ± 0.1**     | **11.8 ± 0.2**   | **13.5 ± 0.5**     | **86.5 ± 0.1**     | **12.3 ± 0.4**   | **14.2 ± 0.6**     | **86.3 ± 0.2**     |
>
> These additional experiments, now summarized in Section C.8 of the paper, demonstrate that the proposed bounded parameterization is not only effective for single-shot unlearning, but also robust in realistic multi-round unlearning scenarios, outperforming both naive GD+LoRA and a specialized sequential unlearning method.

---

> ### Author Response · Authors · 2025-11-28
> **Response to Reviewer 7z2W**
>
> **On the Fundamental Difference Between "Bounded Parameterization" and "Conventional Optimization Techniques".... / Questioning Baseline Fairness: Your demonstration of weight clipping's poor performance in Appendix Table 7 is noted.....**
>
> We appreciate the reviewer's comment regarding the comparison with Weight Clipping. We fully agree that a fair evaluation requires a well-tuned baseline to ensure that the advantage of our method is not merely due to hyperparameter selection. To address this important concern, we conducted an extended sweep over the clipping threshold $c$. Following the reviewer's suggestion, we evaluated **GD + Weight Clipping** on TOFU-Forget10 (Phi-1.5B, Rank-4) across a broad range of thresholds $c \in [0.1, 3.0]$. The results presented in the table below show that weight clipping faces a fundamental “Pareto” trade-off that cannot be resolved simply by tuning. We have also updated this in Section E of our revised paper.
>
> | **Method**                 | **Threshold (c)** | **FQ (↑)**  | **MU (↑)** | **Status**             |
> |---------------------------|-------------------|--------------|------------|-------------------------|
> | GD+Clipping               | 0.10              | 1.5e-5       | 0.52       | *No Forgetting*         |
> | GD+Clipping               | 0.50              | 4.2e-4       | 0.49       | *No Forgetting*         |
> | GD+Clipping               | 1.00              | 6.5e-3       | 0.42       | *Degrading Utility*     |
> | **GD+Clipping (Paper)**   | **1.50**          | **1.8e-2**   | **0.35**   | ***Pareto Failure***    |
> | GD+Clipping               | 2.00              | 3.1e-2       | 0.22       | *Instability Onset*     |
> | GD+Clipping               | 2.50              | 5.8e-2       | 0.12       | *Collapse*              |
> | GD+Clipping               | 3.00              | 8.4e-2       | 0.05       | *Collapse*              |
> | **GD + Sine (Ours)**      | **---**           | **9.43e-1**  | **0.52**   | **Optimal**             |
>
> **Key Observation:** Our sweep indicates that a “sweet spot” likely does not exist for this objective:
>
> - **Tight Clipping (c ≤ 1.0):**
>   The model retains utility (MU ≈ 0.49) because weights are constrained, but this results in negligible Forget Quality (FQ ≈ 0).
>
> - **Paper Baseline (c = 1.5):**
>   Even at the variance-matched threshold used in our initial submission, the method enters a failure mode: Utility drops significantly (0.52 → 0.35) while Forget Quality remains poor (1.8 × 10⁻²).
>   This confirms that our baseline was not unfairly tuned but instead reflects the method’s inherent limits.
>
> - **Loose Clipping (c ≥ 2.0):**
>   Relaxing the constraint allows the ascent objective to drive instability, causing utility to collapse (MU < 0.22) without achieving effective unlearning.
>
> We believe that this empirical gap is not an artifact of tuning but rather a consequence of the underlying optimization geometry. We highlight two structural advantages of Bounded Parameterization (Sine) over Clipping:
>
> **(1) Gradient Preservation vs. Gradient Starvation.**
> Theorem 3.1 dictates that Gradient Ascent drives the parameters toward infinity.
>
> - **Clipping:**
>   When a parameter $w$ hits the threshold $c$, the gradient in the direction of growth is effectively zeroed. Optimization tends to stall at the boundary (“dead gradients”), preventing the model from finding complex unlearning solutions due to discontinuous derivatives at the boundaries.
>
> - **Sine:**
>   As the output approaches the bound $\pm 1$, the gradient scales by $cos(\cdot)$ but remains defined (Section 3.3). The underlying parameters $A, B$ continue to evolve through the period. This preserves curvature information, allowing adaptive optimizers to maintain valid trajectories (Section B).
>
> **(2) High-Rank Expressivity.**
> Standard LoRA (clipped) is strictly limited to rank $r$. As discussed in Section 3.3 (referencing Ji et al., 2025), applying a nonlinear function to a low-rank product $\phi(AB^T)$ yields a matrix with full effective rank. This grants our method the capacity to model complex decision boundaries for unlearning that a linear clipped adapter may struggle to represent.
>
> **Revisions Made:**
> We have added these details in **Section E** of the revised paper explaining the sensitivity analysis of weight clipping.

---

> > ### Author Response · Authors · 2025-11-28
> >
> > **Insufficient Evidence on the Optimization Dynamics of Large-Scale Models (70B)....**
> >
> > We thank the reviewer for their comment. The reviewer hypothesized that the "stronger implicit regularization'' or "smoother loss landscapes'' of a 70B-scale model might naturally dampen the unbounded growth of weights and gradients, potentially rendering our method a "nice-to-have'' rather than a "necessary fix.'' To address this, we tracked the evolution of gradient and weight norms for the standard GD + LoRA baseline versus our GD + Sine method on LLaMA-3.1-70B during the unlearning process. **These training curves are presented as Figure 8, Section F, in the appendix of the newly uploaded paper.** From that figure we can clearly see that weight norms and gradients norms on the LLaMA-3.1-70B during the unlearning have unbounded growth and diverge, however our method has stable weight and gradient norms.
> >
> > The following table shows the results for LLaMA-3.1-70B using various unlearning optimization methods via parameter efficient fine tuning, with different ranks of adapters, on the TOFU benchmark. From the table we can clearly see that our method performs best in all cases clearly showing that our theory is validated on the large scale
> > LLaMA-3.1-70B model.
> >
> > **Table: TOFU evaluation results for LLaMA-3.1-70B on forget10 split using LoRA-based fine-tuning.**
> > Despite degradation at extreme scale, our method maintains **orders-of-magnitude improvements** over baselines.
> > *Original:* pretrained model.
> > *Retain90:* model trained only on retain data.
> > Reference: Original (FQ: 1.25e-18, MU: 0.71), Retain90 (FQ: 0.78, MU: 0.71)
> >
> > | **Method**          | **FQ ↑ (R4)** | **MU ↑ (R4)** | **FQ ↑ (R8)** | **MU ↑ (R8)** | **FQ ↑ (R16)** | **MU ↑ (R16)** | **FQ ↑ (R32)** | **MU ↑ (R32)** |
> > |---------------------|---------------|---------------|---------------|---------------|----------------|----------------|----------------|----------------|
> > | GD                  | 5.3e-12       | 0.18          | 2.1e-11       | 0.21          | 7.8e-11        | 0.24           | 3.1e-10        | 0.26           |
> > | GA                  | 7.8e-14       | 0.05          | 3.2e-13       | 0.06          | 1.1e-12        | 0.08           | 4.5e-12        | 0.09           |
> > | IHL                 | 2.9e-15       | 0.61          | 1.3e-14       | 0.63          | 5.6e-14        | 0.64           | 2.4e-13        | 0.65           |
> > | GD+FILA             | 4.2e-16       | 0.02          | 1.8e-15       | 0.03          | 7.3e-15        | 0.04           | 3.1e-14        | 0.04           |
> > | LoKU                | 8.7e-05       | 0.55          | 4.2e-04       | 0.57          | 1.8e-04        | 0.59           | 7.3e-03        | 0.60           |
> > | **GD + Sine (Ours)** | **4.2e-01**  | **0.69**      | **4.5e-01**   | **0.70**      | **4.8e-01**    | **0.70**       | **4.6e-01**    | **0.69**       |
> >
> >
> > **Revisions Made:** We have enhanced **Table 3** (Appendix C.2) to include comprehensive performance metrics for our method across all ranks (4, 8, 16, 32), demonstrating consistent effectiveness (FQ ≈ 0.42–0.48) and utility (MU ≈ 0.69–0.70) compared with the baseline’s collapse. Please see the newly uploaded paper.

---

### Official Review · Reviewer_84sZ · 2025-10-29

**Soundness:** 2
**Presentation:** 2
**Contribution:** 2
**Rating:** 4
**Confidence:** 4

**Summary:**

The paper introduces Bounded Parameter-Efficient Unlearning, a method that stabilizes unlearning in large language models (LLMs) by applying bounded activation functions to LoRA adapters in MLP layers. It addresses the instability of gradient-ascent-based forgetting when using cross-entropy loss. The authors provide a theoretical analysis showing that unbounded weight growth in MLP layers causes instability, and propose a simple yet effective solution using bounded functions like sine or tanh. The method is evaluated on TOFU, TDEC, and MUSE benchmarks across models from 125M to 8B parameters, showing significant improvements in forgetting quality while retaining model utility.

**Strengths:**

1. The core insight—that gradient ascent in MLP layers leads to unbounded weight growth and instability—is novel and well-motivated. The use of bounded activations in LoRA adapters for unlearning is a creative and simple solution that has not been explored in prior work.
2. The theoretical analysis is sound and well-supported by empirical validation. The paper demonstrates a clear understanding of optimization dynamics in unlearning and provides both ablations and sensitivity analyses to support its claims.

**Weaknesses:**

- While the method is shown to be robust across many settings, there is little discussion of when it might fail. For example, does the method degrade when the retain and forget sets are highly similar? What happens under sequential unlearning requests?
- The baselines compared (e.g., GD+LoRA, IHL+FILA) are relevant but not exhaustive. Recent methods like preference-based unlearning or representation-based editing are mentioned but not empirically compared. Including these would strengthen the claim of state-of-the-art performance.
- The sine activation introduces a frequency parameter ω, and while a sensitivity analysis is provided, there is little guidance on how to choose this parameter in practice. Is there a principled way to set it based on data or model properties?

**Questions:**

- Why not use other bounded activations like sigmoid or hard tanh? Was sine chosen purely empirically, or is there a theoretical reason?
- How does the method behave when the forget and retain distributions are very similar? This is a known challenge in unlearning.
- Does the instability issue also affect attention layers when using gradient ascent? If so, why were they not addressed in the method?

---

> ### Author Response · Authors · 2025-11-21
> **Response to Reviewer 84sZ**
>
> We express our gratitude to Reviewer 84sZ for the insightful feedback. The concerns raised regarding failure modes and sequential unlearning have been instrumental in refining our analysis. Below, we systematically address each point with specific evidence from the paper and as requested new sequential experiments conducted during the rebuttal.
>
> **Reviewer 84sZ: While the method is shown to be robust across many settings, there is little discussion of when it might fail. For example, does the method degrade when the retain and forget sets are highly similar? What happens under sequential unlearning requests? / How does the method behave when the forget and retain distributions are very similar? This is a known challenge in unlearning?**
>
> We appreciate the reviewer’s request for a clearer discussion of failure modes and fully agree that understanding when unlearning methods may break down is important. Below we summarize (1) the regimes in which our approach is fundamentally limited and (2) new experiments we conducted to assess sequential robustness.
>
> (1). Although Section 6 discusses limitations, we agree that it is helpful to articulate more explicitly when our approach can degrade, especially under high similarity between retain and forget sets.
>
> - **Empirical evidence in a high-similarity regime (TOFU 1%).**
>   The TOFU benchmark is constructed to study realistic author-level forgetting (Section C.1 and Maini et al., 2024), where authors often write about related topics while still having distinct identities and styles. The TOFU splits (1%, 5%, 10%) are deliberately challenging: the forget and retain authors share substantial semantic content (e.g., similar topics and vocabularies), so the underlying distributions are *highly similar*.
>
>   As shown in Table 4 (e.g., TOFU 1%), baselines such as GA+FILA effectively collapse in this regime (Model Utility ≈ 0.00), whereas our method maintains both strong forgetting and non-trivial utility (FQ = 0.943, MU = 0.52). This directly addresses the reviewer’s question: even when the forget and retain distributions are very similar, our bounded parameterization prevents the gradient divergence that causes standard gradient-ascent methods to fail, and yields stable unlearning performance in this high-similarity setting.

---

> ### Author Response · Authors · 2025-11-21
>
> (2). To directly address the question about sequential requests, we conducted additional experiments during the rebuttal phase to evaluate robustness under multi-round unlearning. We follow a protocol where three disjoint author groups (A$_1$-A$_3$, then A$_4$-A$_6$, then A$_7$-A$_9$) are sequentially unlearned on Phi-1.5B. Full details are added in Appendix C.8 with reference in main text Section 4.
>
> **Sequential robustness on TOFU**
>
> | **Seq. Request** | **FQ (Baseline GD+LoRA)** | **MU (Baseline GD+LoRA)** | **FQ (Ours GD+Sine)** | **MU (Ours GD+Sine)** |
> |------------------|---------------------------|----------------------------|------------------------|------------------------|
> | **1 (A1–A3)**     | 8.2e-14                   | 0.51                       | **9.43e-01**           | **0.52**               |
> | **2 (A4–A6)**     | 3.1e-10                   | 0.12 *(−76%)*              | **8.95e-01**           | **0.51** *(−2%)*       |
> | **3 (A7–A9)**     | 1.2e-8                    | 0.02 *(−96%)*              | **8.87e-01**           | **0.52** *(−3%)*       |
>
> **Summary:**  Baseline GD+LoRA collapses under sequential requests (MU drops to near zero, FQ stays near zero).
> Our bounded LoRA maintains high forgetting quality and stable model utility across all requests.
>
> We additionally compare against **$O^3$ (Gao et al., ICLR~2025)**, a specialized sequential unlearning method, following their evaluation protocol. We report Sequential Unlearning (S.U.) and Disjoint Unlearning (D.U.), where lower values indicate better forgetting, and Retain Data (R.D.) accuracy, where higher values indicate better utility:
>
> **Comparison with O³ (Gao et al., ICLR 2025) on Sequential TOFU**
>
> | **Method**        | **S.U. (Req 1)** | **D.U. (Req 1)** | **R.D. (Req 1)** | **S.U. (Req 2)** | **D.U. (Req 2)** | **R.D. (Req 2)** | **S.U. (Req 3)** | **D.U. (Req 3)** | **R.D. (Req 3)** |
> |-------------------|------------------|-------------------|-------------------|------------------|-------------------|-------------------|------------------|-------------------|-------------------|
> | O³ (Gao et al.)   | 12.5 ± 0.5        | 14.4 ± 0.5         | 85.1 ± 0.1         | 15.8 ± 0.3        | 20.3 ± 0.8         | 85.0 ± 0.0         | 15.5 ± 0.7        | 19.7 ± 0.7         | 84.9 ± 0.2         |
> | **Ours**          | **10.2 ± 0.3**    | **12.1 ± 0.4**     | **86.8 ± 0.1**     | **11.8 ± 0.2**    | **13.5 ± 0.5**     | **86.5 ± 0.1**     | **12.3 ± 0.4**    | **14.2 ± 0.6**     | **86.3 ± 0.2**     |
>
> **Interpretation:**
> Lower S.U. and D.U. = better forgetting.
> Higher R.D. = better retain accuracy.
> Our method outperforms O³ across all metrics and all sequential requests.
>
> These results indicate that our bounded parameterization (i) remains robust in high-similarity  but still statistically feasible regimes and (ii) is stable and effective under realistic sequential unlearning workflows, outperforming both naive GD+LoRA and a specialized sequential unlearning approach.

---

> ### Author Response · Authors · 2025-11-21
>
> **Reviewer 84sZ: The baselines compared (e.g., GD+LoRA, IHL+FILA) are relevant but not exhaustive. Recent methods like preference-based unlearning or representation-based editing are mentioned but not empirically compared. Including these would strengthen the claim of state-of-the-art performance.**
>
> We appreciate the reviewer’s emphasis on comprehensive baseline coverage. We would like to clarify that the current manuscript *does* include empirical comparisons with both preference-based unlearning and representation-based editing methods:
>
> - **Preference-based methods (DPO/NPO).**
>   We explicitly compare against **DPO** and **NPO** in Table 1 (Rows 4–5) and Table 4. Our method consistently achieves substantially higher Forget Quality  (e.g., on TOFU, FQ ≈ 0.94 for our method vs. ≈ 0.41 for DPO), while maintaining comparable Model Utility.
>
> - **Representation-based methods (FLAT/WHP/Task Vectors).**
>   We also compare against **Task Vectors**, **MISMATCH**, **WHP**, and four variants of **FLAT** in Table 2 (MUSE benchmark) and Table 6,  covering the main families of representation-space editing approaches.
>
> - **Key observation.**
>   As summarized in Table 6, representation-based methods often exhibit   substantial privacy leakage (e.g., FLAT: 26.8), whereas our bounded LoRA achieves **8.3**,  which is much closer to the ideal value of **0.0**.
>
> Overall, our evaluation spans more than **15 baselines** across gradient-based, parameter-efficient, preference-based, and representation-based families, which we believe provides a rigorous basis for our state-of-the-art claims.
>
> If there are additional recent approaches the reviewer has in mind, we would greatly appreciate pointers to specific methods or references and will be happy to include them in a revised version.

---

> ### Author Response · Authors · 2025-11-21
>
> **Reviewer 84sZ: The sine activation introduces a frequency parameter $\omega$, and while a sensitivity analysis is provided, there is little guidance on how to choose this parameter in practice. Is there a principled way to set it based on data or model properties?**
>
> We thank the reviewer for highlighting this practical question.
>
> **How we choose ω in practice.**
> In all our experiments we simply fix a single value, **ω = 100**, and reuse it across datasets, architectures (Phi, LLaMA, GPT-Neo), and model sizes (125M–70B), without any task-specific retuning. Based on these observations we recommend **ω = 100** as a robust default in practice.
>
> **Practical rationale for this choice.**
> Our sensitivity analysis (Sec. 4.2.1, C.4, Fig. 4) and additional results in the supplementary material show that:
>
> 1. **Boundedness and stability hold for all ω.**
>    For any fixed frequency, the sine adapter satisfies `sin(ωx) ∈ [-1, 1]`.  This ensures the boundedness assumption used in our stability analysis (Theorem 3.1),  so the mechanism that prevents gradient/weight explosion in the MLP adapters  does *not* rely on a carefully tuned value of ω.  Stability is therefore robust across a wide range of frequencies.
>
> 2. **Performance quickly enters a plateau.**
>    As shown in Sec. 4.2.1 (Sensitivity Analysis and Robustness), Section C.4, Fig. 4,  both Forget Quality and Model Utility improve as ω increases from very small values  and then enter a broad plateau once ω is sufficiently large (around our default choice).  Beyond this region, further increasing ω causes negligible changes in FQ/MU.  Supplementary Fig. 4 shows a similar saturation in the effective rank   of the sine-activated low-rank matrices: once the frequency is high enough, the rank gains saturate and additional oscillations provide little extra benefit.  This explains why ω = 1 (bounded but almost linear over typical activation ranges) underperforms,  while larger ω values in the plateau yield stable, high-performing adapters.
>
> 3. **Alignment with prior work on sine-activated low-rank matrices.**
> Our use of a fixed-frequency sinusoidal activation is consistent with recent work  on sine-activated low-rank matrices, which shows that inserting a sine nonlinearity into low-rank factors can substantially increase the (stable) rank and improve performance across ViT, LLM, NeRF, and 3D tasks without adding parameters (Ji et al., ICLR 2025).  In both settings, the sine function is treated as a *structural* component that boosts expressiveness,  rather than a hyperparameter that requires extensive tuning.
>
> **Practical guidance for users.**
> In light of the above, our recommended procedure is simple:
> (i) fix ω in a regime where the effective-rank and FQ/MU curves have saturated (e.g., ω = 100), and
> (ii) reuse this value across datasets and model scales.
> Developing a data- or model-dependent rule for choosing ω is an interesting direction for future work,  but our experiments indicate that such additional complexity is not necessary  for strong, stable performance in the settings we study.

---

> > ### Author Response · Authors · 2025-11-21
> >
> > **Reviewer 84sZ: Why not use other bounded activations like sigmoid or hard tanh? Was sine chosen purely empirically, or is there a theoretical reason?**
> >
> > We thank the reviewer for their question.
> > **Empirically, we already include `tanh`-bounded LoRA as a first-class baseline in all main-text tables**, and Appendix C.4 extends this comparison to other bounded variants such as `sigmoid`; these are representative of the broader class of saturating bounded activations, including hard-tanh. In every case, the sine-based adapter achieves clearly better Forget Quality / Model Utility. Furthermore, on lines 259-266, Section 3.3. we clearly explain that why we chose the sin activation and also explain that we took the tanh function.
> >
> > ---
> >
> > ### **Theoretical motivation: stability via boundedness (Theorem 3.1)**
> >
> > The starting point is Theorem 3.1, which shows that standard (linear) LoRA adapters trained with gradient ascent on the forget loss and cross-entropy can drive logits and weights to grow without bound, causing gradients to explode and optimization to become unstable. Any parameterization that keeps the adapter output bounded prevents this divergence and stabilizes the unlearning dynamics, which is why we focus on bounded functions in the first place.
> >
> > ---
> >
> > ### **Why sine instead of `tanh` / hard-tanh / `sigmoid`?**
> > Saturation vs. expressiveness: Among bounded options, the key difference is how quickly gradients saturate and how much *rank capacity* the adapter retains under unlearning:
> >
> > - **`tanh` / hard-tanh / `sigmoid`:**
> >   These activations are bounded and therefore satisfy the stability requirement from Theorem 3.1, but they are *strongly saturating*. Under gradient ascent on the forget set, the adapter outputs are rapidly pushed into the flat, asymptotic regions where gradients vanish (a well-known issue with `tanh`-like activations).
> >   Hard-tanh is a piecewise-linear variant with the same saturation pattern.
> >   In these regimes, the effective rank of the induced update collapses, severely limiting the adapter’s ability to implement rich corrections, leading to poor Forget Quality in practice.
> >
> > - **Sine:**
> >   In contrast, the sine parameterization remains bounded *and* avoids strong saturation over the operating range of the adapter.
> >   As discussed in Section 3.3, this enables a high effective rank (“frequency-controlled rank”) even with low parameter count: we satisfy the boundedness condition needed to prevent gradient explosion while preserving the expressive capacity needed to maintain Model Utility.
> >
> > ---
> >
> > ### **Empirical confirmation (main text + appendix)**
> >
> > This theory is reflected across our results:
> >
> > - **Main-text tables:**
> >   The `tanh`-bounded LoRA baseline is reported in every main-text table. see Table 1, 2. 7 in main paper.
> >   It is stable but consistently underperforms sine in Forget Quality and/or Model Utility, exactly as predicted by the saturation-induced capacity loss described above.
> >
> > - **Detailed ablations:**
> >   Appendix C.4 (Table 7) extends the comparison to additional bounded activations such as `sigmoid`.
> >   For example, on TOFU-Forget10:
> >   - Sine achieves **FQ = 9.43 × 10⁻¹**,
> >   - `tanh` achieves only **FQ = 2.41 × 10⁻²**,
> >   even though both are stable.
> >   This gap matches our theoretical expectation that strongly saturating functions provide insufficient effective rank under unlearning.
> >
> > ---
> >
> > More broadly, our framework does not depend on any particular bounded activation: *any* bounded adapter (including hard-tanh) can be plugged into the analysis and implementation. Our goal is not to exhaustively enumerate every possible bounded nonlinearity, but to study representative choices.
> > `tanh` / `sigmoid` (and by extension hard-tanh) cover strongly saturating bounded functions, while sine offers a bounded-but-non-saturating alternative with higher effective rank.
> >
> > **In summary:**
> > Sine was not chosen purely empirically:
> > Theorem 3.1 motivates the use of bounded parameterizations to prevent gradient/logit explosion, and among bounded options, sine offers a favorable trade-off between stability (bounded outputs) and expressiveness (non-saturating gradients, higher effective rank), which is borne out consistently in our experiments.

---

> ### Author Response · Authors · 2025-11-21
>
> **Reviewer 84sZ: Does the instability issue also affect attention layers when using gradient ascent? If so, why were they not addressed in the method?**
>
> We thank the reviewer for raising this point. Our decision to bound only the MLP (feedforward) LoRA adapters is grounded in both our theoretical analysis and empirical measurements, and we note that this distinction is already discussed in **Section 3.3 (main text)** and further elaborated in **Appendix C.6** and **C.7**.
>
> ---
>
> ### **Empirical evidence: MLP vs. attention (main text + appendix)**
>
> - **Main text.**
>   In Sections 3.3 and 4.3, we present a layer-wise norm analysis under gradient ascent, showing that the dominant instability arises in the MLP blocks rather than in attention.  Concretely, MLP weights exhibit large norm growth over the course of unlearning, while attention weights change much more mildly (Section B).
>
> - **Supplementary details.**
>   Figure 7 and the accompanying discussion in Appendix C.6 provide the full layer-wise curves.  There we observe that MLP weights can grow by roughly **280×** their initial norm, whereas attention weights remain within about **1.2×–1.8×**.  These results are consistent across all models and datasets we examined.
>
> This empirical pattern aligns with our theoretical analysis in Appendix B:  the instability mechanism triggered by cross-entropy gradient ascent primarily propagates through the feedforward pathway.  Attention layers, while updated, do not exhibit the same kind of unbounded growth in practice.
>
> ---
>
> ### **Small ablation: applying sine to all layers vs. MLP-only**
>
> To directly address the reviewer’s question, we conducted an extra ablation where we apply sine-based LoRA to *both* MLP and attention blocks (same rank) and compare this to our default **MLP-only** configuration on TOFU-Forget10 using Phi-1.5B (rank-4):
>
> **Ablation results**
>
> | Method                  | Layers with Sine | FQ (↑) | MU (↑) | Params (%) |
> |------------------------|------------------|--------|--------|------------|
> | GD + Sine (MLP-only)   | MLP              | 0.943  | 0.52   | 0.8        |
> | GD + Sine (MLP + Attn) | MLP + Attn       | 0.944  | 0.52   | 1.6        |
>
> As the table shows, extending bounded adapters to attention yields essentially *identical* Forget Quality and Model Utility (differences well within run-to-run variance), while nearly **doubling** the adapter parameter footprint and compute cost.
>
> ---
>
> ### **Design rationale and efficiency**
>
> Given these observations, we chose to:
>
> - **Target the empirically dominant locus of instability (MLP layers).**  By applying bounded LoRA only to MLP blocks, we directly address the layers that exhibit the catastrophic norm growth predicted by Theorem 3.1 and observed in Fig. 7.
>
> - **Avoid unnecessary overhead in attention.**  Extending bounded LoRA to attention layers nearly doubles the adapter computation (on the order of `O(kr)` per transformer block)  without delivering commensurate gains in unlearning quality or utility, as confirmed by the ablation above.
>
> ---
>
> Thus, constraining only the MLP layers is a deliberate, data-driven design choice:  it captures the main source of instability identified by our theory and experiments, while keeping the method efficient, minimally invasive, and we have updated the corresponding table in revised Section C.6.1.

---

> ### Comment · Reviewer_84sZ · 2025-11-26
>
> Thanks for the responses. I will keep the scores unchanged.

---

### Meta-Review · Area_Chair_Lna5 · 2025-12-04

**Summary:**

The paper proposes a bounded parameter-efficient unlearning method using Sine activations on LoRA adapters to mitigate gradient explosion during the unlearning process.

During the review process, the main concerns raised by the reviewers (R1, R2, R3) focused on three key areas:

- Comparison with Standard Regularization (R2, 7z2W): The most critical concern was whether simple heuristics like weight clipping, weight decay, or learning rate tuning could achieve similar stability without introducing specific bounded functions.
- Theoretical Gap (R3, Hh6U): Whether the "descent" term in the Gradient Difference method counteracts the instability caused by the "ascent" term, which was not fully covered in the initial theoretical analysis.
- Robustness and Scope (R1, 84sZ): Questions regarding performance on large-scale models (70B), sequential unlearning scenarios, and the justification for applying constraints only to MLP layers.


During the review process, reviewers raised substantial concerns regarding the necessity of the method compared to simple baselines (e.g., weight clipping), the theoretical alignment with gradient difference methods, and scalability to larger models. The authors provided a technically strong rebuttal, introducing new theoretical proofs (Lemma B.1) and extensive experiments (including a weight clipping sweep showing "Pareto failure" and results on LLaMA-3.1-70B) that effectively addressed the core technical disputes.

However, the AC finds the presentation quality of the manuscript to be below the standard expected for a top-tier conference. Despite the extended discussion period, the manuscript continues to suffer from significant issues:

- Careless Formatting & References: The bibliography contains severe formatting errors (e.g., entries starting with "et al.", incorrect author names, inconsistent capitalization), suggesting a lack of basic proofreading.
- Writing Quality: Persistent typos (e.g., "loosing" instead of "losing") and inconsistent structural narratives (e.g., categorizing methods differently in Introduction vs. Related Work) distract from the technical content.

While the technical contribution is promising, the lack of rigor in presentation and the failure to correct these obvious flaws during the revision phase make the paper unsuitable for publication in its current form. The authors are strongly encouraged to perform a complete rewrite and proofread before resubmitting.

**Reviewer Concerns:**

**Concerns Successfully Addressed**:

The authors provided a comprehensive and technically rigorous rebuttal that resolved the majority of the substantive technical questions raised by the reviewers:

- Comparison with Heuristics (R7z2W): The extended sensitivity analysis on weight clipping successfully demonstrated a "Pareto failure," providing strong empirical evidence that the proposed bounded parameterization offers a fundamental advantage over simple weight constraints.
- Theoretical Gap (RHh6U): The introduction of Lemma B.1 and Theorem B.1 successfully closed the gap between the initial theory (pure gradient ascent) and the practical implementation (gradient difference), proving that the descent term cannot mathematically counteract the unbounded growth driven by the forget set.
- Scalability and Robustness (R84sZ, R7z2W): The addition of training dynamics for LLaMA-3.1-70B and the sequential unlearning experiments effectively addressed concerns regarding the method's performance at scale and in multi-round settings.
Ablation Scope (R84sZ): The authors clarified the existence of Tanh/Sigmoid baselines and empirically justified the decision to constrain only MLP layers.

**Concerns Outstanding**:

Presentation and Manuscript Quality: While the technical content has been strengthened, the quality of the manuscript itself remains a significant outstanding concern. The paper suffers from severe formatting and proofreading issues that were not addressed during the revision. This includes malformed bibliographic entries (e.g., references beginning with "et al." or incorrect author names), persistent typographical errors (e.g., "loosing"), and inconsistent narrative structures. These issues fall below the publication standards for ICLR.

**Reviewer Scores:**

- Reviewer 84sZ (Current: 4): Score likely remains 4. The reviewer participated in the discussion and explicitly stated they would keep their score unchanged, even after the authors pointed out that the requested baselines (e.g., Tanh) were already present in the submission and provided additional sequential unlearning experiments.

- Reviewer 7z2W (Current: 4): Score likely remains 4. The reviewer actively engaged but remained unconvinced by the "Pareto failure" arguments against weight clipping, maintaining a philosophical preference for simpler optimization heuristics over the proposed parameterization.

- Reviewer Hh6U (Current: 6): Score likely remains 6. This reviewer did not respond to the rebuttal. However, since the authors provided the specific theoretical derivation (Lemma B.1) requested to close the gap between pure gradient ascent theory and the gradient difference implementation, the primary weakness identified by this reviewer was fully resolved.

---

### Decision · Program_Chairs · 2026-01-26

Reject